# Connected correlations in partitioning protocols: a case study and beyond

S. Bocini

Université Paris-Saclay, CNRS, LPTMS, 91405, Orsay, France
saverio.bocini@universite-paris-saclay.fr

March 30, 2023

## Abstract

The assumption of local relaxation in inhomogeneous quantum quenches allows to compute asymptotically the expectation value of local observables via hydrodynamic arguments known as generalized hydrodynamics (GHD). In this work we address formally the question of when an observable is *local enough* to be described by GHD using the playground of partitioning protocols and non-interacting time evolution. We show that any state evolving under a quadratic Hamiltonian can be described via a set of decoupled dynamical fields such that one of those fields can be identified with a space-time-dependent generalisation of the root density. By studying the contribution to a connected spin correlation of each of those fields independently, we derive the locality conditions under which an observable can be described using the root density only. That shows both the regime of validity for hydrodynamic approaches that aim at describing the asymptotic value of observables in term of the root density only, such as GHD, and the locality conditions necessary for Gaussianification to occur.

# 1 Introduction

Generalized hydrodynamics (GHD) has been successfully applied to compute the expectation value of local observables in integrable systems prepared in several inhomogeneous setups, among which the archetypal example of partitioning protocols in quantum spin chains [1–31]. GHD relies on the assumption of locality and encodes all the information about the large-time behavior of a system in a classical field called *root density* (for generic models, more than one root density might be needed). The root density was originally introduced to characterise sta-

tionary macro-states in integrable systems described by thermodynamic Bethe Ansatz [32,33] and was later extended so as to capture also the large-time limit of the expectation value of local observables in inhomogeneous states [34, 35]. Remarkably, GHD has also been shown to describe experimental setups of cold atomic gases constrained to one dimension [36–38]. When applied to correlations, however, this powerful description shows its limitations: a correlation is characterised by multiple lengths and it is not evident in which limits a generalized hydrodynamic theory can be effective. We address the problem in the partitioning protocol in which two different states that are homogeneous in the bulk are joined together. The asymptotic GHD prediction for this setting is such that, at the Euler scale in which space and time scales are sent to infinity while their ratio is kept constant, the expectation values of local observables become functions of $x/t$, where $t$ is the time and $x$ accounts for the position of the observable with respect to the junction. The ratio $x/t$ is often referred to as *ray* and the ray-dependent state that one obtains at the Euler scale is called *locally quasi-stationary state* (LQSS), following Ref. [39]. When studying how connected correlations approach zero with the distance(s) of the operators, taking the time limit before the space limit so to apply GHD can become a purely abstract procedure, in the sense that it does not correctly account for numerical and experimental results, where time and distances are large but finite.

A related problem is the definition itself of the object(s) used to describe the asymptotic value of local observables, i.e. the space-time-dependent root density(s). In GHD such a definition relies on the assumption of separation of scales: we wait long enough that the system looks homogeneous at the spacial scale defined by our observable. This may simply not hold when we consider correlations. What is the typical size of the "cells in space" at the Euler scale in which correlations can be described by generalized hydrodynamics? Note that this problem goes beyond the correlation functions, since the definition of the root density at any time is problematic also if we restrict to local observables. The problem of defining the root density independently from the scale-separation hypothesis has already been tackled by Refs. [40,41] in the case of Gaussian states in the setup of non-interacting spin-chains and local observables. Here we take a step further and we consider any kind of partitioning protocol (not necessarily Gaussian) and we apply the result to the problem of connected correlation functions.

For simplicity, we focus on the two-point spin-spin connected correlation function and we consider only initial states in which the expectation value of a (finite) odd number of fermionic operators is zero, since those operators are non-local in the original spin theory. Moreover, we only discuss "bulk physics", meaning that we leave out exceptional behaviours; for example, we will not investigate observables at the edge of the lightcone, which in general are affected by different corrections [35,42–44], similarly to what happens to free fermions at the edge of a confining potential [45–47]. Under these assumptions, taking inspiration from the formulation of quantum mechanics in phase space [48–52], we describe the state at any time in terms of a set of classical dynamical fields that evolve in time under decoupled equations of motion. We will identify in one of those fields the (space-time-dependent) root density, which allows us to define such an object at any time and without assuming local relaxation. By comparing the contribution given by the root density with the ones from the other fields, we identify a condition on the scaling of distance between spins that is sufficient for the root density to give the leading contribution, settling the question of locality.

As a byproduct, we also discuss Gaussianification in partitioning protocols. Gaussianification is a phenomenon that happens in any quadratic system with a translational invariant, short-range Hamiltonian, and concerns the time-evolution of an initial state with finite corre-

lation length. Essentially, it consists in the fact that, in the large-time limit, the state is locally indistinguishable from a Gaussian state. Gaussianification has been proved by Refs. [53, 54] for any initial state that relaxes to a homogeneous state. That hypothesis excludes from the picture any partitioning protocol, for which the Lieb-Robinson bound [55] prevents global relaxation. We show Gaussianification for partitioning protocols, discussing also how local the observable should be for the phenomenon to happen. Because of the Lieb-Robinson bound, the system can not be described in terms of a single homogeneous Gaussian state, but whenever the leading order of local observables is accounted for by the root density alone, the state is locally Gaussian nonetheless.

The paper is organized as follows. In Section 3 we recall the well-known procedure to recast a free spin chain into a diagonal fermionic Hamiltonian; this allows us to set the notation. In Section 4 we present a convenient parametrization of any state evolving under a free Hamiltonian that allows us to study separately all the independent contribution to the large time limit of any observable; here we give a definition of root density (for non-interacting systems) that holds at any time and without any locality assumptions. In Section 5 we introduce the kind of partitioning protocols we specialize to. Sections 6, 7 and 8 contain the asymptotic analysis and our main results for the independent contributions to the spin-spin correlation function. In Section 9 we discuss how to include higher order correlations in our framework and in Section 10 we draw our final conclusions.

To conclude the introduction, we point out that the problem of how to compute correlations in a GHD framework was recently tackled in Refs. [56, 57], but with a different approach: in those works, fluctuations are quantized on top of the GHD solution in a similar fashion to Landau's theory of superfluidity [58], while our approach is somewhat closer to a microscopic theory, in the sense that we start from an exact description of the system and, from there, we derive leading order and corrections in a given scaling limit (accordingly, Refs. [40, 41] dubbed our kind of approach *higher-order* GHD). We also point out the collection of works reviewed in Ref. [59], in which the authors study correlation functions in quantum quenches. However, they focus on correlations between observables at different times, related to Drude weight and Onsager matrix, that are accounted for by a hydrodynamic description. In this work we consider correlations between observables evaluated at the same time, assuming that their distance diverges, which a priori goes beyond any hydrodynamic description. Finally, we point out that long-range correlations similar to the ones that we study here were recently considered in Ref. [60] in the case of interacting system, starting directly from a LQSS; here we consider a different setup, in which the (potential) relaxation to the LQSS still has to take place.

## 2   Preview of the results

Considering partitioning protocols, we investigate the problem of computing the large-time asymptotics of correlation functions in which the lengths scale with time. In particular, we focus on the example of the spin-spin connected correlation function $S_{m,n}^z(t) \equiv \langle \sigma_m^z(t) \sigma_n^z(t) \rangle - \langle \sigma_m^z(t) \rangle \langle \sigma_n^z(t) \rangle$, where $m \sim n \sim t$ and $m - n \sim t^\alpha$ with $\alpha \in [0, 1]$, for a variety of partitioning protocols evolving in the transverse-field Ising chain. We will see how to decompose the observable exactly at any time into independent contributions, each accounted for by a dynamical field. The various dynamical fields are decoupled, and we will identify one

of them with an inhomogeneous generalization of the root density in quadratic models.

In general, a *lightcone structure* forms, in which spins that are affected by the inhomogeneity (i.e. spins inside the lightcone) behave differently to spins that are not (i.e. outside the lightcone). The region of spins affected by the inhomogeneity grows linearly in time, consistently with Lieb-Robinson bounds [55].

The contributions to $S_{m,n}^z(t)$ are of three types: a contribution coming from the root density, a contribution coming from the so-called *auxiliary field*, that with the root density accounts for the Gaussian part of the state, and a contribution coming from the non-Gaussian part of the state. We can summarize the behavior of the three contributions as follows. The root density gives a contribution that scales as $t^{-2\alpha}$ inside the lightcone and does not alter the initial value of the correlation outside the lightcone. The auxiliary field gives a contribution $t^{-3+2\alpha}$ outside the lightcone and $t^{-\min(3-2\alpha,\,2)}$ inside the lightcone. The non-Gaussian part gives a contribution that decays as $t^{-1}$ both inside and outside the lightcone. The fact that the contributions are independent means, for example, that the root density gives the same contribution whether the state is Gaussian or not. More in general, if we know the initial value of the dynamical fields in the homogeneous state that form the partitioning protocol, we already know how the spin-spin connected correlation decays for large time for any value of $\alpha$, and we can also determine which one of the three contributions gives the leading order.

In conclusion, our results allow to determine the qualitative behavior of $S_{m,n}^z(t)$ for any scaling of $m-n$ in potentially *any* partitioning protocol evolving under the Ising Hamiltonian. Moreover, as we will discuss, the generalization of our results to other non-interacting models should not be complicated. Note how $\alpha = 1/2$ emerges as a special scaling for the correlation function: for $0 \leq \alpha < 1/2$ the leading order of the correlation function can be accounted for by considering solely the root density, while it is in general not the case for $1/2 \leq \alpha \leq 1$. This introduces a notion of locality: if the observable that we are interested in acts over a region of space that does not scale faster than $t^{1/2}$, the observable is *local enough* to be described by the root density only. A direct implication is that such an observable is also local enough to Gaussianize, in the sense that, for $\alpha < 1/2$, it exists a Gaussian state in which the expectation value of the observable has the same leading order as in the exact state of the system.

We also go beyond the qualitative behavior of $S_{m,n}^z(t)$ and we compute explicitly the full leading order, providing some new analytical results regarding connected correlations for partitioning protocols in the Ising model. All the analytical results are checked by comparing it with numerical simulations for different values of $\alpha$ and for different partitioning protocols.

## 3  The Hamiltonian

We consider spin-chain Hamiltonians that can be mapped to quadratic combinations of fermionic operators by a Jordan-Wigner transformation. While our discussion holds with minor modifications for any free translational-invariant spin chain with short-range interactions, we specialize to the transverse field Ising (infinite) chain (TFIC), whose Hamiltonian reads

$$H = -J\sum_{j\in\mathbb{Z}}\left(\sigma_j^x\sigma_{j+1}^x + h\sigma_j^z\right),\tag{1}$$

where $J > 0$ and we assume $|h| \neq 1$, in such a way that the model is not critical.

The Jordan-Wigner transformation from spin operators to Majorana fermions reads

$$\sigma_\ell^z = -\mathrm{i}a_{2\ell-1}a_{2\ell}, \qquad \sigma_\ell^x = \left(\prod_{j=-\infty}^{\ell-1}\sigma_j^z\right)a_{2\ell-1}, \qquad \sigma_\ell^y = \left(\prod_{j=-\infty}^{\ell-1}\sigma_j^z\right)a_{2\ell}, \qquad (2)$$

where the Majorana fermions $a_j$ satisfy the algebra $[a_i, a_j]_+ \equiv a_i a_j + a_j a_i = 2\delta_{i,j}$. By definition, the Hamiltonian of a non-interacting (or *free*, or *quadratic*) quantum spin chain in Majorana fermions (i.e. after a Jordan-Wigner transformation) reads

$$H = \frac{1}{4}\sum_{i,j\in\mathbb{Z}}a_i \mathcal{H}_{ij}a_j, \qquad (3)$$

where $\mathcal{H}$ is an infinite purely-imaginary Hermitian matrix.[1] Since we consider translational-invariant models, $\mathcal{H}_{2m+i,2n+j} = \mathcal{H}_{2(m-n)+i,j}$, for any $m, n \in \mathbb{Z}$ and $i, j \in \{1, 2\}$, i.e. $\mathcal{H}$ is the infinite-size limit of a $(2 \times 2)$-block-circulant matrix. For example,

$$H = \mathrm{i}J\sum_{\ell\in\mathbb{Z}}(a_{2\ell}a_{2\ell+1} + ha_{2\ell-1}a_{2\ell}) \qquad (4)$$

in the TFIC.

To deal with the quadratic form $\mathcal{H}$ representing the Hamiltonian of translational-invariant free models, it is a standard procedure to introduce its symbol $\hat{h}(p)$ as

$$(\hat{h}(p))_{i,j} := \sum_{m\in\mathbb{Z}}\mathrm{e}^{-\mathrm{i}pm}\mathcal{H}_{2m+i,j} \qquad \Leftrightarrow \qquad \mathcal{H}_{2m+i,2n+j} = \int_{-\pi}^{+\pi}\frac{\mathrm{d}p}{2\pi}\mathrm{e}^{\mathrm{i}(m-n)p}(\hat{h}(p))_{i,j}, \qquad (5)$$

for $i, j \in \{1, 2\}$ (see e.g. [61] for more details). For the TFIC we have

$$\hat{h}(p) = -2J[(h - \cos p)\sigma^y + (\sin p)\sigma^x]. \qquad (6)$$

The definition of the symbol allows for the introduction of a general method to diagonalize a translational-invariant free model. Introducing the Bogoliubov angle as the angle $\theta(p) \in [-\pi, \pi)$ such that

$$\frac{\hat{h}(p)}{\epsilon(p)} = \mathrm{e}^{-\mathrm{i}\frac{\theta(p)}{2}\sigma^z}\sigma^y\mathrm{e}^{\mathrm{i}\frac{\theta(p)}{2}\sigma^z}, \qquad (7)$$

we have that

$$v(p) := \frac{1}{\sqrt{2}}\begin{pmatrix}\mathrm{e}^{-\mathrm{i}\frac{\theta(p)}{2}} \\ \mathrm{i}\mathrm{e}^{\mathrm{i}\frac{\theta(p)}{2}}\end{pmatrix} \qquad (8)$$

is the eigenvector of $\hat{h}(p)$ with eigenvalue

$$\epsilon(p) \equiv 2J\sqrt{1 + h^2 - 2h\cos p}, \qquad (9)$$

where $\epsilon(p)$ is referred to as *dispersion relation*. Explicitly, it can be shown that

$$\mathrm{e}^{\mathrm{i}\theta(p)} = 2J\frac{\mathrm{e}^{\mathrm{i}p} - h}{\epsilon(p)}. \qquad (10)$$

---

[1] We stress out that, in our convention, the indices of the matrix $\mathcal{H}$ can be negative.

Finally, we introduce the infinite vectors

$$w_{2m+j}(p) := \mathrm{e}^{\mathrm{i}mp} v_j(p), \tag{11}$$

for $m \in \mathbb{Z}$ and $j \in \{1, 2\}$. It can be shown that

$$\sum_{\ell' \in \mathbb{Z}} \mathcal{H}_{\ell,\ell'} w_{\ell'}(p) = \epsilon(p) w_\ell(p), \qquad \sum_{\ell' \in \mathbb{Z}} \mathcal{H}_{\ell,\ell'} w_{\ell'}^*(p) = -\epsilon(p) w_\ell^*(p), \tag{12}$$

providing all the eigenvalues and the eigenvectors of $\mathcal{H}$. At this point we can finally introduce the Bogoliubov transformation between fermionic operators

$$\begin{cases} b^\dagger(p) = \frac{1}{\sqrt{2}} w(p) \cdot a \\ b(p) = \frac{1}{\sqrt{2}} w^*(p) \cdot a \end{cases} \iff a_j = \sqrt{2} \int_{-\pi}^{\pi} \frac{\mathrm{d}p}{2\pi} \left( w_j^*(p) b^\dagger(p) + w_j(p) b(p) \right). \tag{13}$$

The operators $b(k), b^\dagger(k)$ are referred to as *Bogoliubov fermions* and satisfy the algebra

$$[b(p), b^\dagger(q)]_+ = 2\pi \delta(p - q), \qquad [b^\dagger(p), b^\dagger(q)]_+ = 0 = [b(p), b(q)]_+. \tag{14}$$

After the Bogoliubov transformation, the Hamiltonian is in its diagonal form

$$H = \int_{-\pi}^{\pi} \frac{\mathrm{d}p}{2\pi} \epsilon(p) \left( b^\dagger(p) b(p) - \frac{1}{2} \right). \tag{15}$$

The advantage of using the Bogoliubov fermions over the initial spin operators or the Majorana operators is that the former have a trivial dynamics (in the Heisenberg picture), evolving simply by a phase: $b(p, t) = \mathrm{e}^{-\mathrm{i}t\epsilon(p)} b(p, 0)$. We point out that within this formalism one could compute the conserved charges of the model, as done in Refs. [61, 62].

# 4 The state in terms of dynamical fields

## 4.1 Gaussian states

Given a state, let us consider the $2n$-point connected correlation functions of Majorana operators. The first two of them read

$$\langle a_{\ell_1} a_{\ell_2} \rangle_c = \langle a_{\ell_1} a_{\ell_2} \rangle$$
$$\langle a_{\ell_1} a_{\ell_2} a_{\ell_3} a_{\ell_4} \rangle_c = \langle a_{\ell_1} a_{\ell_2} a_{\ell_3} a_{\ell_4} \rangle - \langle a_{\ell_1} a_{\ell_2} \rangle \langle a_{\ell_3} a_{\ell_4} \rangle + \langle a_{\ell_1} a_{\ell_3} \rangle \langle a_{\ell_2} a_{\ell_4} \rangle - \langle a_{\ell_1} a_{\ell_4} \rangle \langle a_{\ell_2} a_{\ell_3} \rangle, \tag{16}$$

and similar definitions for $n > 2$,[2] where the expectation values are computed over the state under consideration. By definition, if the $2n$-point connected correlation function is zero, the

---

[2]In general, the $N$-point connected correlation function can be written via a the generating function $Z(\{\eta\}) = \ln \left\langle \mathrm{e}^{\sum_i \eta_i a_i} \right\rangle$ as

$$\langle a_{\ell_1} ... a_{\ell_N} \rangle_c = \frac{\partial}{\partial \eta_{\ell_N}} ... \frac{\partial}{\partial \eta_{\ell_1}} Z(\{\eta\})|_{\eta_i = 0}, \tag{17}$$

where the $\eta_i$'s are Grassmann variables. Incidentally, we recall that the chain rule for Grassmann variables is $\frac{\partial}{\partial \eta} f(\theta(\eta)) = \frac{\partial \theta}{\partial \eta} \frac{\partial f(\theta)}{\partial \theta}$, where the order of the factors matters, and the Leibniz rule is $\frac{\partial}{\partial \eta_j}(f(\{\eta\}) g(\{\eta\})) = \frac{\partial f(\{\eta\})}{\partial \eta_j} g(\{\eta\}) + P(f(\{\eta\})) \frac{\partial g(\{\eta\})}{\partial \eta_j}$, where $P$ is an operator that changes the sign of each term with an odd number of variables in the polynomial expansion of the function.

$2n$-point correlation function can be expressed as a sum of products of correlation functions with maximum degree $2(n-1)$. We say that the state is *Gaussian* if all the $2n$-point connected correlation functions of Majorana operators are zero except, at most, the one with $n = 1$ (recall that we are assuming that the correlation functions of an odd number of fermions are zero).

For example, the eigenvectors of a quadratic Hamiltonians are in general Gaussian, as can be shown by applying Wick's theorem. More generally, a homogeneous Gaussian state is described by a density matrix

$$\hat{\rho} = \frac{1}{\text{Tr}[e^Q]} e^Q, \tag{18}$$

where $Q = \frac{1}{4} \sum_{i,j \in \mathbb{Z}} a_i \mathcal{Q}_{ij} a_j$ and $\mathcal{Q}$ is an infinite purely-imaginary Hermitian matrix such that $\mathcal{Q}_{2m+i,2n+j} = \mathcal{Q}_{2(m-n)+i,j}$. The Gaussianity of the state (18) can once again be proved by Wick's theorem [63]. Gaussian states are closely related to free systems since it can be proved that a Gaussian state stays Gaussian when it evolves under a quadratic Hamiltonian.

The rest of this section about Gaussian states is essentially a brief review of Ref. [41], although we make explicit some definitions that were left implicit. The goal is to give a phase-space formulation of quantum mechanics for what concerns Gaussian states. First of all, we can exploit the fact that Gaussian states are completely described by the 2-point correlation to encode all the information about the state in the *correlation matrix*, defined as

$$\Gamma_{ij} := \delta_{i,j} - \langle a_i a_j \rangle, \tag{19}$$

where the expectation value is evaluated with respect to the exact state of the system.

As done for the Hamiltonian, it is often convenient to describe the correlation matrix in terms of a symbol. Using the matrix notation

$$\left( \Gamma^{m-n}_{\frac{m+n}{2}} \right)_{i,j} \equiv \Gamma_{2m+i,2n+j}, \tag{20}$$

the *physical inhomogeneous symbol* of the correlation matrix is defined as

$$\hat{\Gamma}^{phys}_x(p) := \sum_{z \in 2(\mathbb{Z}+x)} e^{-izp} \Gamma^z_x, \qquad \text{where} \quad x \in \mathbb{Z}/2, \tag{21}$$

with inversion relation

$$\Gamma^z_x = \int_{-\pi}^{\pi} \frac{dp}{2\pi} e^{izp} \hat{\Gamma}^{phys}_x(p), \qquad \text{where} \quad z \in 2(\mathbb{Z}+x). \tag{22}$$

For both the correlation matrix and its symbol, when we specify the time dependence $t$, we refer to the state of the system at time $t$.

Importantly, the physical inhomogeneous symbol satisfies $\hat{\Gamma}^{phys}_x(p+\pi) = (-1)^{2x} \hat{\Gamma}^{phys}_x(p)$ and it does not reduce to the homogeneous one even if the matrix $\Gamma^z_x$ does not depend on $x$ (i.e. $\Gamma_{2m+i,2n+j} = \Gamma_{2(m-n)+i,j}$). Indeed, if we apply the definition of the physical inhomogeneous symbol to a homogeneous matrix, we get $\hat{\Gamma}^{phys}_x(p) = \frac{\hat{\Gamma}(p)+(-1)^{2x}\hat{\Gamma}(p+\pi)}{2}$, where $\hat{\Gamma}(p)$ is the homogeneous symbol of the correlation matrix, defined as in Eq. (5). Since it is inconvenient to have a symbol whose properties make a distinction between $x \in \mathbb{Z}$ and $x \in \mathbb{Z} + 1/2$, we want to define a new symbol that plays the role of $\hat{\Gamma}^{phys}_x(p)$ in describing physical quantities but does not distinguish between $x \in \mathbb{Z}$ and $x \in \mathbb{Z} + 1/2$. The fundamental observation is

that the inversion relation (22) holds for the more general symbol $\hat{\Gamma}_x^{phys}(p) + \hat{\Gamma}_x^{gauge}(p)$, where $\hat{\Gamma}_x^{gauge}(p+\pi) = -(-1)^{2x}\hat{\Gamma}_x^{gauge}(p)$, i.e.

$$\Gamma_x^z = \int_{-\pi}^{\pi} \frac{\mathrm{d}p}{2\pi} \mathrm{e}^{\mathrm{i}zp}(\hat{\Gamma}_x^{phys}(p) + \hat{\Gamma}_x^{gauge}(p)), \qquad \text{where} \quad z \in 2(\mathbb{Z}+x), \tag{23}$$

since the gauge part gives zero under integration. Moreover, note that the symbol is always evaluated in $x \in \mathbb{Z}/2$, so we can work with an extension to $\mathbb{R}$ of the physical symbol that assumes arbitrary values in $x \notin \mathbb{Z}/2$ and coincides with the physical symbol modulo a gauge part when $x$ is evaluated over the lattice sites. Since only the matrices are directly linked to physical observables, two symbols that differ in the gauge part and/or in $x \notin \mathbb{Z}/2$ still describe the same physics. Such a freedom can be used to define a new symbol, which we simply call *inhomogeneous symbol* $\hat{\Gamma}_x(p)$, that is an entire function in $x$ and that in the homogeneous limit (i.e. when $\Gamma_{2m+i,2n+j} = \Gamma_{2(m-n)+i,j}$), gives back exactly the homogeneous symbol; we also require the symbol to remain Hermitian and periodic in $p$. For example, a possible choice for the inhomogeneous symbol is

$$\hat{\Gamma}_x(p) = \sum_{y\in\mathbb{Z}/2} \frac{\sin(2\pi(x-y))}{2\pi(x-y)}(\hat{\Gamma}_y^{phys}(p) + \hat{\Gamma}_{y+1/2}^{phys}(p)), \tag{24}$$

which is an entire function of $x$, gives the matrix $\Gamma$ via the same inversion formula (22), i.e.

$$\Gamma_x^z = \int_{-\pi}^{\pi} \frac{\mathrm{d}p}{2\pi} \mathrm{e}^{\mathrm{i}zp}\hat{\Gamma}_x(p), \qquad \text{where} \quad z \in 2(\mathbb{Z}+x), \tag{25}$$

and coincides with the homogeneous symbol when the state is homogeneous and the symbol is evaluated in $x \in \mathbb{Z}/2$. In particular, we have $\hat{\Gamma}_x(p) = \hat{\Gamma}_x^{phys}(p) + \hat{\Gamma}_{x+1/2}^{phys}(p)$ for $x \in \mathbb{Z}/2$, which is in the form $\hat{\Gamma}_x^{phys}(p) + \hat{\Gamma}_x^{gauge}(p)$.

A decomposition of the physical inhomogeneous symbol for the correlation matrix that is fundamental for this work is the following:

$$\hat{\Gamma}_{x,t}^{phys}(p) =$$
$$= \frac{\mathrm{i}}{2}\begin{pmatrix} 0 & \mathrm{e}^{-\mathrm{i}\theta(p)} \\ -\mathrm{e}^{\mathrm{i}\theta(p)} & 0 \end{pmatrix} + (-1)^{2x}\frac{\mathrm{i}}{2}\begin{pmatrix} 0 & \mathrm{e}^{-\mathrm{i}\theta(p+\pi)} \\ -\mathrm{e}^{\mathrm{i}\theta(p+\pi)} & 0 \end{pmatrix} + \sum_{y\in\mathbb{Z}/2}\int_{-\pi}^{\pi}\frac{\mathrm{d}q}{2\pi}\mathrm{e}^{2\mathrm{i}q(x-y)} \times$$

$$\begin{pmatrix} \mathrm{e}^{\mathrm{i}\frac{\theta(p-q)-\theta(p+q)}{2}}(4\pi\rho_{y,t;o}^{phys}(p) + \mathrm{Re}\,\psi_{y,t}^{phys}(p)) & \mathrm{e}^{-\mathrm{i}\frac{\theta(p-q)+\theta(p+q)}{2}}(-4\pi\mathrm{i}\rho_{y,t;e}^{phys}(p) - \mathrm{Im}\,\psi_{y,t}^{phys}(p)) \\ \mathrm{e}^{\mathrm{i}\frac{\theta(p-q)+\theta(p+q)}{2}}(4\pi\mathrm{i}\rho_{y,t;e}^{phys}(p) - \mathrm{Im}\,\psi_{y,t}^{phys}(p)) & \mathrm{e}^{-\mathrm{i}\frac{\theta(p-q)-\theta(p+q)}{2}}(4\pi\rho_{y,t;o}^{phys}(p) - \mathrm{Re}\,\psi_{y,t}^{phys}(p)) \end{pmatrix}, \tag{26}$$

where the fields $\rho_{x,t}^{phys}(p)$ and $\psi_{x,t}^{phys}(p)$ are defined as

$$\rho_{x,t}^{phys}(k) := \int_{-\pi}^{+\pi} \frac{\mathrm{d}p}{(2\pi)^2} \mathrm{e}^{2\mathrm{i}xp} \langle b^\dagger(k-p)b(k+p)\rangle_t,$$
$$\psi_{x,t}^{phys}(k) := -2\int_{-\pi}^{+\pi} \frac{\mathrm{d}p}{2\pi} \mathrm{e}^{2\mathrm{i}xp} \langle b(p+k)b(p-k)\rangle_t. \tag{27}$$

Here the expectation values $\langle ...\rangle_t$ are computed with respect to the state of the system at time $t$ and the indices $e$ and $o$ refer respectively to the even and odd part of the function in the variable $p$:

$$f_e(p) = \frac{f(p)+f(-p)}{2}, \qquad f_o(p) = \frac{f(p)-f(-p)}{2}. \tag{28}$$

Note that $\rho_{x,t}^{phys}(p)$ is a real function and $\psi_{x,t}^{phys}(p)$ is a complex function obeying $\psi_{x,t}^{phys}(p) = -\psi_{x,t}^{phys}(-p)$. Moreover, both the fields are even (resp. odd) under $k \to k+\pi$ when $x$ is integer (resp. half-odd-integer), as a consequence of the property $\hat{\Gamma}_x^{phys}(p+\pi) = (-1)^{2x}\hat{\Gamma}_x^{phys}(p)$.

The decomposition (27) is derived starting from the definition of the symbol (21), plugging in the expression of the correlation matrix in term of the Majorana fermions (19) and expressing the Majorana fermions in terms of Bogoliubov fermions using the Bogoliubov transformation (13). After performing all the possible simplifications, we can express time-dependence simply as a phase using $b(p,t) = e^{-it\epsilon(p)}b(p,0)$ and we can regroup together all the terms with the same time-dependence: those are the independent subspaces. Finally, we rotate the symbol by the Bogoliubov angle with respect to the $z$-direction in such a way to treat different non-interacting models on the same ground (the only thing that varies from one model to the other is the Bogoliubov angle). Once the invariant-subspaces have been identified, we define a dynamical field to describe each of them.

The discussion about the symbol's gauge freedom applies to the dynamical fields as well: we can introduce the dynamical fields $\rho_{x,t}(p)$ and $\psi_{x,t}(p)$ that parametrize the (general) inhomogeneous symbol and coincide with $\rho_{x,t}^{phys}(p)$ and $\psi_{x,t}^{phys}(p)$ for $x \in \mathbb{Z}/2$, modulo a gauge transformation, which in this case consists in adding a function $g_{x,t}(p)$ such that $g_{x,t}(p+\pi) = -(-1)^{2x}g_{x,t}(p)$. We always choose a gauge such that $\rho_{x,t}(p)$ remains a real function and $\psi_{x,t}(p)$ remains a complex function obeying $\psi_{x,t}(p) = -\psi_{x,t}(-p)$. The parametrization of the inhomogeneous symbol in terms of the dynamical fields reads

$$
\hat{\Gamma}_{x,t}(p) = e^{-i\frac{\theta(p)}{2}\sigma^z} \star \bigg( (4\pi\rho_{x,t;e}(p) - 1)\,\sigma^y + 4\pi\rho_{x,t;o}(p)\mathbb{I}_2 +
$$
$$
+ \operatorname{Re}\psi_{x,t}(p)\sigma^z - \operatorname{Im}\psi_{x,t}(p)\sigma^x \bigg) \star e^{i\frac{\theta(p)}{2}\sigma^z}, \quad (29)
$$

where the *star* (or *Moyal*) product is defined as

$$
(f \star g)(p,x) := f(p,x)e^{i\frac{\overleftarrow{\partial}_x \overrightarrow{\partial}_p - \overleftarrow{\partial}_p \overrightarrow{\partial}_x}{2}}g(p,x) \equiv \left[ e^{i\frac{\partial_x\partial_q - \partial_p\partial_y}{2}}f(p,x)g(q,y) \right]_{(y,q)\to(x,p)}. \quad (30)
$$

Note that Moyal product allows also for an integral representation that involves only the symbol(s) evaluated in semi-integer positions. In particular, it is worth writing down the following property:

$$
f(p) \star \hat{\Gamma}_x(p) \star g(p) = \int_{-\pi}^{\pi} \frac{dk}{2\pi} \sum_{y\in\mathbb{Z}/2} f(p+k)\hat{\Gamma}_y(p)g(p-k)e^{2i(x-y)k} \quad (31)
$$

that do not require the symbol $\hat{\Gamma}_x(p)$ to be entire in $x$.

Importantly, if now we compute $\rho_{x,t}(p)$ on a homogeneous state, it reduces to the density of quasi-particle excitations, giving the thermodynamic limit of $\frac{\langle b_k^\dagger b_k \rangle}{2\pi}$, where the operators $b_k^\dagger, b_k$ are the finite-size version of the Bogoliubov fermions (see e.g. Ref. [34] for the definition of the density of excitations for finite size and its thermodynamic limit). Note that the space-time dependence has disappeared, as a consequence of the homogeneity assumption. So, in the homogeneous limit, $\rho_{x,t}(p)$ is nothing but the root density of non-interacting systems.

The time-dependence of our model is completely accounted for by the time evolution of the dynamical fields, which, being defined in terms of Bogoliubov fermions, evolve in a simple

way:

$$\rho_{x,t}(p) = \sum_{y \in \mathbb{Z}/2} \int_{-\pi}^{+\pi} \frac{\mathrm{d}q}{2\pi} \mathrm{e}^{2iq(x-y)} \mathrm{e}^{-it[\epsilon(p+q)-\epsilon(p-q)]} \rho_{y,0}(p),$$

$$\psi_{x,t}(p) = \sum_{y \in \mathbb{Z}/2} \int_{-\pi}^{+\pi} \frac{\mathrm{d}q}{2\pi} \mathrm{e}^{2iq(x-y)} \mathrm{e}^{-it[\epsilon(p+q)+\epsilon(-p+q)]} \psi_{y,0}(p).$$

(32)

The dynamical equations can be written as

$$i\frac{\partial}{\partial t}\rho_{x,t}(p) = \epsilon(p) \star \rho_x(p,t) - \rho_{x,t}(p) \star \epsilon(p) \equiv \{\{\epsilon(p), \rho_{x,t}(p)\}\},$$

$$i\frac{\partial}{\partial t}\psi_{x,t}(p) = \epsilon(p) \star \psi_{x,t}(p) + \psi_{x,t}(p) \star \epsilon(p) \equiv \{\{\epsilon(p), \psi_{x,t}(p)\}\}_+.$$

(33)

Note that the two fields are decoupled. Importantly, the first order of the homogeneous expansion of the equation of motion for $\rho_{x,t}(p)$, representing the local relaxation hypothesis, is

$$\frac{\partial}{\partial t}\rho_{x,t}(p) + \epsilon'(p)\frac{\partial}{\partial x}\rho_{x,t}(p) = 0,$$

(34)

which is precisely the equation of GHD [35]. This property, combined with the homogeneous limit of $\rho_{x,t}(p)$ that we discussed above, allows us to treat $\rho_{x,t}(p)$ as the space-time dependent generalization of the root density, and in the following we will refer to it as such. What we gained with respect to the GHD approach is that we have its definition for any time, without any need for local-relaxation hypothesis. This enables many interesting computations, such as the investigation of the condition for which the (space-time dependent) root density gives the leading contribution. The field $\psi_{x,t}(p)$, instead, will be referred to as *auxiliary field*, to highlight the fact that $\rho_{x,t}(p)$ and $\psi_{x,t}(p)$ are the only two fields needed to describe exactly any Gaussian state.

We point out that the parametrization (29) of the correlation matrix's symbol was already given in Ref. [41], but the definition of the fields was left implicit.

## 4.2 A step beyond Gaussian states

Our goal is to extend the phase-space description of quantum mechanics formulated above for Gaussian states to any kind of state. To go beyond Gaussian states, we have to consider nonzero $2n$-point connected correlation functions also for $n > 1$. We start by discussing the 4-point connected correlation. Let us define the object that corresponds to the correlation matrix in this new case:

$$C_{m,n,r,s} := \langle a_m a_n a_r a_s \rangle - \langle a_m a_n \rangle \langle a_r a_s \rangle + \langle a_m a_r \rangle \langle a_n a_s \rangle - \langle a_m a_s \rangle \langle a_n a_r \rangle .$$

(35)

As with the correlation matrix, when we specify the time dependence $t$ we refer to the state of the system at time $t$. We remark that $C_{m,n,r,s} = 0$ for a Gaussian state as a trivial consequence of Wick theorem. The dynamical fields that we introduced in the previous section are obviously not enough to describe the 4-point connected correlation of Majorana operators and we need to introduce new fields to complete its description. First of all, we

define a new kind of physical inhomogeneous symbol, adapted to the present situation:

$$\left(\hat{C}^{phys}_{x_1,x_2}(p_1,p_2)\right)_{j_1,j_2,j_3,j_4} :=$$

$$\sum_{z_1\in 2(\mathbb{Z}+x_1)} \sum_{z_2\in 2(\mathbb{Z}+x_2)} e^{-ip_1z_1}e^{-ip_2z_2} C_{2(x_1+\frac{z_1}{2})+j_1,2(x_1-\frac{z_1}{2})+j_2,2(x_2+\frac{z_2}{2})+j_3,2(x_2-\frac{z_2}{2})+j_4}, \quad (36)$$

which is a real and anti-symmetric $2\times 2\times 2\times 2$ tensor. We also define its matrix version $\tilde{C}^{phys}_{x_1,x_2}(p_1,p_2)$ as

$$\left(\tilde{C}^{phys}_{x_1,x_2}(p_1,p_2)\right)_{2(j_1-1)+j_3,2(j_2-1)+j_4} = \left(\hat{C}^{phys}_{x_1,x_2}(p_1,p_2)\right)_{j_1,j_2,j_3,j_4}, \quad (37)$$

which is a $4\times 4$ matrix. The inversion relation is

$$C_{2(x_1+\frac{z_1}{2})+j_1,2(x_1-\frac{z_1}{2})+j_2,2(x_2+\frac{z_2}{2})+j_3,2(x_2-\frac{z_2}{2})+j_4} =$$

$$= \int_{-\pi}^{\pi} \frac{d^2p}{(2\pi)^2} e^{iz_1p_1}e^{iz_2p_2} \left(\tilde{C}^{phys}_{x_1,x_2}(p_1,p_2)\right)_{2(j_1-1)+j_3,2(j_2-1)+j_4}, \quad (38)$$

with $z_i \in 2(\mathbb{Z}+x_i)$. In order to parametrize the new symbol with a set of new dynamical fields, we decompose it in a similar way to what we did for $\hat{\Gamma}_x(p)$. In particular, we rewrite the symbol in terms of Bogoliubov fermions, which allows to easily identify the components with the same time dependence. Then we parametrize each independent component with a field. Once again, the Bogoliubov angle is used to treat different free models on the same ground. In the end we get

$$\left(e^{i\frac{\theta(p_1)}{2}\sigma^z}\otimes e^{i\frac{\theta(p_2)}{2}\sigma^z}\right)\star \tilde{C}^{phys}_{x_1,x_2,t}(p_1,p_2)\star\left(e^{-i\frac{\theta(p_1)}{2}\sigma^z}\otimes e^{-i\frac{\theta(p_2)}{2}\sigma^z}\right) =$$

$$4\xi^{phys}_{x_1,x_2,t;oo}(p_1,p_2)e_{00} + 4\xi^{phys}_{x_1,x_2,t;eo}(p_1,p_2)e_{20} + 4\xi^{phys}_{x_1,x_2,t;oe}(p_1,p_2)e_{02} + 4\xi^{phys}_{x_1,x_2,t;ee}(p_1,p_2)e_{22}+$$

$$+ 2\operatorname{Re}(\Omega^{phys}_{x_1,x_2,t}(p_1,p_2) + \omega^{phys}_{x_1x_2}(p_1,p_2))e_{11} - 2\operatorname{Re}(\Omega^{phys}_{x_1,x_2,t}(p_1,p_2) - \omega^{phys}_{x_1x_2}(p_1,p_2))e_{33}+$$

$$+ 2\operatorname{Im}(\Omega^{phys}_{x_1,x_2,t}(p_1,p_2) - \omega^{phys}_{x_1x_2}(p_1,p_2))e_{13} + 2\operatorname{Im}(\Omega^{phys}_{x_1,x_2,t}(p_1,p_2) + \omega^{phys}_{x_1x_2}(p_1,p_2))e_{31}+$$

$$+ 4\operatorname{Im}\Upsilon^{phys}_{x_1,x_2,t;o}(p_1,p_2)e_{01} + 4\operatorname{Im}\Upsilon^{phys}_{x_1,x_2,t;e}(p_1,p_2)e_{21} - 4\operatorname{Re}\Upsilon^{phys}_{x_1,x_2,t;o}(p_1,p_2)e_{03}+$$

$$- 4\operatorname{Re}\Upsilon^{phys}_{x_1,x_2,t;e}(p_1,p_2)e_{23} + 4\operatorname{Im}\Upsilon^{phys}_{x_2,x_1,t;o}(p_2,p_1)e_{10} + 4\operatorname{Im}\Upsilon^{phys}_{x_2,x_1,t;e}(p_2,p_1)e_{12}+$$

$$- 4\operatorname{Re}\Upsilon^{phys}_{x_2,x_1,t;o}(p_2,p_1)e_{30} - 4\operatorname{Re}\Upsilon^{phys}_{x_2,x_1,t;e}(p_2,p_1)e_{32}, \quad (39)$$

where $e_{\alpha\beta} \equiv \sigma^\alpha \otimes \sigma^\beta$ and the indices $e$ and $o$ of $\xi$ refer respectively to the symmetrization and the antisymmetrization in the corresponding momentum, e.g.

$$f_{eo}(p_1,p_2) = \frac{f(p_1,p_2) + f(-p_1,p_2) - f(p_1,-p_2) - f(-p_1,-p_2)}{4}, \quad (40)$$

and the indices $e$ and $o$ of $\Upsilon$ refer respectively to a symmetrization and an antisymmetrization in the *first* momentum. The fields are defined as

$$\xi^{phys}_{x_1,x_2,t}(p_1,p_2) := \int_{-\pi}^{+\pi} \frac{d^2q}{(2\pi)^2} e^{2ix_1q_1}e^{2ix_2q_2} \langle b^\dagger(p_1-q_1)b(p_1+q_1)b^\dagger(p_2-q_2)b(p_2+q_2)\rangle_{c,t},$$

$$\Omega^{phys}_{x_1,x_2,t}(p_1,p_2) := -\int_{-\pi}^{+\pi} \frac{d^2q}{(2\pi)^2} e^{2ix_1q_1}e^{2ix_2q_2} \langle b(q_1+p_1)b(q_1-p_1)b(q_2+p_2)b(q_2-p_2)\rangle_{c,t},$$

$$\Upsilon^{phys}_{x_1,x_2,t}(p_1,p_2) := \int_{-\pi}^{+\pi} \frac{d^2q}{(2\pi)^2} e^{2ix_1q_1}e^{2ix_2q_2} \langle b^\dagger(-q_1+p_1)b(q_1+p_1)b(q_2+p_2)b(q_2-p_2)\rangle_{c,t}.$$

$$(41)$$

We did not include the field $\omega$ in the definitions above because it is actually obtained from $\xi$, using

$$
\begin{aligned}
\omega^{phys}_{x_1,x_2,t}(p_1,p_2) &= \int_{-\pi}^{+\pi} \frac{\mathrm{d}^2 q}{(2\pi)^2} \mathrm{e}^{2\mathrm{i}x_1 q_1} \mathrm{e}^{2\mathrm{i}x_2 q_2} \langle b(q_1+p_1)b(q_1-p_1)b^\dagger(-q_2-p_2)b^\dagger(-q_2+p_2)\rangle_{c,t} \\
&= \frac{1}{2} \sum_{y\in\mathbb{Z}/2} \int_{-\pi}^{+\pi} \frac{\mathrm{d}q}{2\pi} \mathrm{e}^{2\mathrm{i}q(x_1-x_2)} \mathrm{e}^{2\mathrm{i}y(p_1-p_2)} \xi^{phys}_{x_1+y,x_2-y,t}(q-p_1,q+p_2)+ \\
&\quad + \frac{(-1)^{2x_2}}{2} \sum_{y\in\mathbb{Z}/2} \int_{-\pi}^{+\pi} \frac{\mathrm{d}q}{2\pi} \mathrm{e}^{2\mathrm{i}q(x_1-x_2)} \mathrm{e}^{2\mathrm{i}y(p_1-p_2+\pi)} \xi^{phys}_{x_1+y,x_2-y,t}(q-p_1,q+p_2+\pi),
\end{aligned} \quad (42)
$$

which comes from the symmetry properties of the original correlation function. From the symmetries of the symbol, one can derive several properties of the dynamical fields, such as the fact that $\xi$ is real.

As in the Gaussian case, we have a gauge freedom linked to the symmetry

$$
\hat{C}^{phys}_{x_1,x_2}(p_1,p_2) = (-1)^{2x_1} \hat{C}^{phys}_{x_1,x_2}(p_1+\pi,p_2) = (-1)^{2x_2} \hat{C}^{phys}_{x_1,x_2}(p_1,p_2+\pi). \quad (43)
$$

In particular, if we replace $\hat{C}^{phys}_{x_1,x_2}(p_1,p_2)$ with $\hat{C}^{phys}_{x_1,x_2}(p_1,p_2) + \hat{C}^{gauge}_{x_1,x_2}(p_1,p_2)$, where

$$
\hat{C}^{gauge}_{x_1,x_2}(p_1,p_2) = -(-1)^{2x_1} \hat{C}^{gauge}_{x_1,x_2}(p_1+\pi,p_2) \;\vee\; \hat{C}^{gauge}_{x_1,x_2}(p_1,p_2) = -(-1)^{2x_2} \hat{C}^{gauge}_{x_1,x_2}(p_1,p_2+\pi), \quad (44)
$$

the inversion relation still holds. We thus define the homogeneous symbol $\hat{C}_{x_1,x_2}(p_1,p_2)$ as an entire function of $x_1, x_2 \in \mathbb{R}$ that coincides with $\hat{C}^{phys}_{x_1,x_2}(p_1,p_2)$ for $x_1, x_2 \in \mathbb{Z}/2$, modulo a gauge term, and choose the gauge term in such a way that all the properties of the physical fields are conserved. $\tilde{C}_{x_1,x_2,t}(p_1,p_2)$ is parametrised in an analogous way to $\tilde{C}^{phys}_{x_1,x_2,t}(p_1,p_2)$ in Eq. (39), with the only difference that the physical fields are substituted by generic ones, that differ only by a gauge part in $x_1, x_2 \in \mathbb{Z}/2$ and are continuous in $x_1$ and $x_2$. The dynamical equations satisfied by the new fields are

$$
\begin{aligned}
\mathrm{i}\partial_t \xi_{x_1,x_2,t}(p_1,p_2) &= \{\{\epsilon(p_1),\xi_{x_1,x_2,t}(p_1,p_2)\}\} + \{\{\epsilon(p_2),\xi_{x_1,x_2,t}(p_1,p_2)\}\}, \\
\mathrm{i}\partial_t \Omega_{x_1,x_2,t}(p_1,p_2) &= \{\{\epsilon(p_1),\Omega_{x_1,x_2,t}(p_1,p_2)\}\}_+ + \{\{\epsilon(p_2),\Omega_{x_1,x_2,t}(p_1,p_2)\}\}_+, \\
\mathrm{i}\partial_t \Upsilon_{x_1,x_2,t}(p_1,p_2) &= \{\{\epsilon(p_1),\Upsilon_{x_1,x_2,t}(p_1,p_2)\}\} + \{\{\epsilon(p_2),\Upsilon_{x_1,x_2,t}(p_1,p_2)\}\}_+,
\end{aligned} \quad (45)
$$

where in the terms with $\epsilon(p_1)$ (resp. $\epsilon(p_2)$) the conjugated variables with repsect to the Moyal products are $p_1$ and $x_1$ (resp. $p_2$ and $x_2$), and are solved by

$$
\begin{aligned}
\xi_{x_1,x_2,t}(p_1,p_2) &= \sum_{y_1,y_2\in\mathbb{Z}/2} \int_{-\pi}^{+\pi} \frac{\mathrm{d}^2 k}{(2\pi)^2} \mathrm{e}^{2\mathrm{i}k_1(x_1-y_1)} \mathrm{e}^{2\mathrm{i}k_2(x_2-y_2)} \times \\
&\quad \times \mathrm{e}^{-\mathrm{i}t[\epsilon(p_1+k_1)-\epsilon(p_1-k_1)]} \mathrm{e}^{-\mathrm{i}t[\epsilon(p_2+k_2)-\epsilon(p_2-k_2)]} \xi_{y_1,y_2,0}(p_1,p_2), \\
\Omega_{x_1,x_2,t}(p_1,p_2) &= \sum_{y_1,y_2\in\mathbb{Z}/2} \int_{-\pi}^{+\pi} \frac{\mathrm{d}^2 k}{(2\pi)^2} \mathrm{e}^{2\mathrm{i}k_1(x_1-y_1)} \mathrm{e}^{2\mathrm{i}k_2(x_2-y_2)} \times \\
&\quad \times \mathrm{e}^{-\mathrm{i}t[\epsilon(k_1+p_1)+\epsilon(k_1-p_1)]} \mathrm{e}^{-\mathrm{i}t[\epsilon(k_2+p_2)+\epsilon(k_2-p_2)]} \Omega_{y_1,y_2,0}(p_1,p_2) \\
\Upsilon_{x_1,x_2,t}(p_1,p_2) &= \sum_{y_1,y_2\in\mathbb{Z}/2} \int_{-\pi}^{+\pi} \frac{\mathrm{d}^2 k}{(2\pi)^2} \mathrm{e}^{2\mathrm{i}k_1(x_1-y_1)} \mathrm{e}^{2\mathrm{i}k_2(x_2-y_2)} \times \\
&\quad \times \mathrm{e}^{\mathrm{i}t[\epsilon(-k_1+p_1)-\epsilon(k_1+p_1)]} \mathrm{e}^{-\mathrm{i}t[\epsilon(k_2+p_2)+\epsilon(k_2-p_2)]} \Upsilon_{y_1,y_2,0}(p_1,p_2).
\end{aligned} \quad (46)
$$

To summarize, we encoded the information about the 2- and 4-point connected correlations in a set of five independent dynamical fields, among which the root density.

### 4.3   General non-Gaussian state

The inclusion of higher order non-zero connected correlation functions is straightforward: to account for the $2n$-point connected correlation function we shall introduce a $2^n \times 2^n$ inhomogeneous symbol, which can be described in terms of a finite set of dynamical fields. In order to count the fields corresponding to such a symbol, we have to think of all the possible correlation functions that can be defined using the Bogoliubov fermions $b$ and $b^\dagger$, following three rules: the argument of the operators is irrelevant; correlations that are obtained as a permutation of the fermionic operators one from the other count as one; a correlation and its complex conjugate count as one. So the number of independent fields introduced by the $2n$-point connected correlation function is $n + 1$. The way in which they can be defined is a generalization of what we did above, as are the equations of motion that one obtains. In the main part of this work, we consider observables involving at most 4-point connected correlations, leaving some general considerations for the final discussion.

## 5   Partitioning protocols

### 5.1   Definition of our setup

As the archetypal example of an initial inhomogeneous state, we consider partitioning protocols. We refer as *partitioning protocol* to a setup in which the system is prepared in an inhomogeneous initial state in which connected correlations between spins decay to zero exponentially with distance and such that the state is indistinguishable from a given homogeneous state when considered far to the left of the origin and it is indistinguishable from a second homogeneous state when considered far to the right of the origin. We warn the reader that often in the literature *partitioning protocol* actually refers to the special case in which the correlations between any spin on the left half and any spin on the right half are exactly zero – see e.g. Refs. [39, 44, 64–68] for some specific examples. Often (but not always), the two halves are prepared in stationary states, in such a way that the non-trivial time-evolution takes place only in proximity of the junction. We consider also partitioning protocols in which the junction between the two homogeneous states is not sharp and we do not limit ourselves to stationary homogeneous states.

Given a partitioning protocol, we are interested in the asymptotic study of observables that are local both in spins and in fermions. The simplest local observable that involves also some non-Gaussian fields is the spin-spin connected correlation function

$$S_{mn}^z(t) := \langle \sigma_m^z \sigma_n^z \rangle_t - \langle \sigma_m^z \rangle_t \langle \sigma_n^z \rangle_t, \tag{47}$$

where we let the lattice indices $m$ and $n$ and the distance between spins scale with time as $m \sim n \sim t$ and $m - n \sim t^\alpha$, with $\alpha \in [0, 1]$. This is the observable that we will consider in all our examples.

A distinctive feature of partitioning protocols is the so called *lightcone*. Due to the bounds on propagation of information known as Lieb-Robinson bound [55], for large times, only a

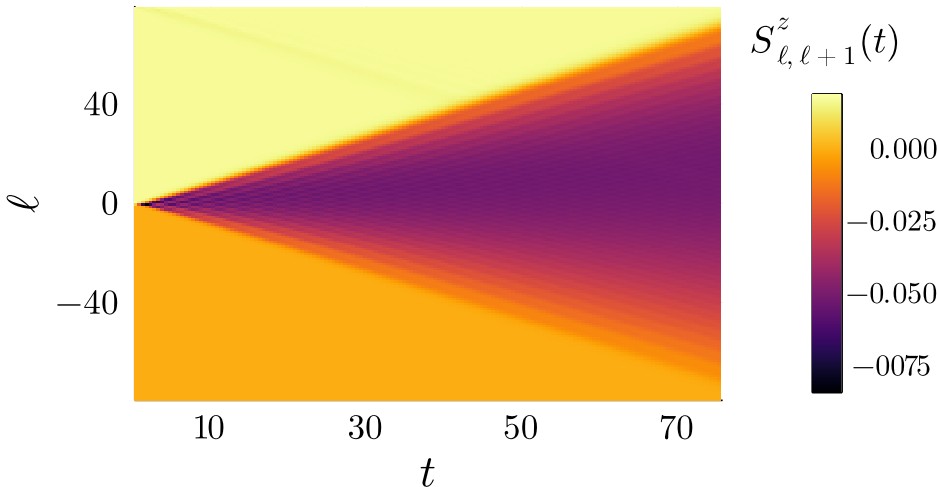

Figure 1: Example of the lightcone structure in a partitioning protocol. We report the expectation value for the nearest-neighbor spin-spin connected correlation $S^z_{\ell,\ell+1}(t)$ in function of position and time for the partitioning protocol detailed in Section 5.2.1.

region that grows linearly around the junction, called lightcone, is affected by the inhomogeneity of the state, while the effects of the inhomogeneity on local observables outside this region are exponentially suppressed – see Fig. 1. In non-interacting systems, the velocity of propagation of the lightcone's front is the maximal group velocity of elementary excitations, i.e. $v_{max} \equiv \max_p \epsilon'(p)$. Note that the propagation's speed in the TFIC is symmetric with respect to the origin since we have $\epsilon(-p) = \epsilon(p)$. We will not delve into more details here, since a lightcone structure will clearly emerge from our computations. For the moment, the only important thing is the definition: we say that a given lattice site $\ell$ is inside the lightcone if $|\ell| < v_{max}t$, where $t$ is the time; analogously, a *ray* $\zeta \equiv \ell/t$ is inside the lightcone if $|\zeta| < v_{max}$.

In the following, we want to see how each field contributes to the spin-spin connected correlation. To do so, it is convenient to rewrite our observable as

$$
\begin{aligned}
S^z_{mn}(t) = & - \langle a_{2m-1} a_{2m} a_{2n-1} a_{2n} \rangle_t + \langle a_{2m-1} a_{2m} \rangle_t \langle a_{2n-1} a_{2n} \rangle_t \\
= & -C_{2(m-1)+1,2(m-1)+2,2(n-1)+1,2(n-1)+2}(t) + \Gamma_{2(m-1)+1,2(n-1)+1}(t) \Gamma_{2(m-1)+2,2(n-1)+2}(t) \\
& + \Gamma_{2(m-1)+1,2(n-1)+2}(t) \Gamma_{2(m-1)+2,2(n-1)+1}(t) \quad (48)
\end{aligned}
$$

and split the correlation matrix in the two parts accounted for by $\rho_{x,t}(p)$ and $\psi_{x,t}(p)$ independently as $\Gamma = \Gamma^\rho + \Gamma^\psi$.[3] The idea is to study independently the behavior of $C$, $\Gamma^\rho$ and $\Gamma^\psi$ in the scaling limit above. In this way, by knowing the value of the fields in the initial state, we also know how the spin-spin connected correlation behaves and where the leading contribution comes from. In practice, we engineer three different partitioning protocols, each presenting just one among the contributions $C$, $\Gamma^\rho$ and $\Gamma^\psi$. To construct such examples, we proceed by introducing a homogeneous state for each of the cases we want to study and combine it

---

[3]The decomposition $\Gamma = \Gamma^\rho + \Gamma^\psi$ can be achieved starting from Eq. (29) and defining $\hat{\Gamma}^\rho_{x,t}(p) = e^{-i \frac{\theta(p)}{2} \sigma^z} \star ((4\pi \rho_{x,t;e}(p) - 1) \sigma^y + 4\pi \rho_{x,t;o}(p) \mathbb{I}_2) \star e^{i \frac{\theta(p)}{2} \sigma^z}$ and $\hat{\Gamma}^\psi_{x,t}(p) = e^{-i \frac{\theta(p)}{2} \sigma^z} \star (\mathrm{Re}\, \psi_{x,t}(p)\sigma^z - \mathrm{Im}\, \psi_{x,t}(p)\sigma^x) \star e^{i \frac{\theta(p)}{2} \sigma^z}$. Then $\Gamma^\rho$ and $\Gamma^\psi$ are the correlation matrices obtained from those inhomogeneous symbols respectively. We will come back to this point later on.

with an infinite-temperature thermal state to create a partitioning protocol; we choose the infinite-temperature state to be on the left. The peculiarity of the infinite-temperature state which makes it convenient to consider is that all the $2n$-point connected correlation functions are zero, which implies that all the dynamical field in this state are zero, except the root density, that equals $\frac{1}{4\pi}$.

## 5.2 Examples of initial states

### 5.2.1 Pure-root-density state

The first example that we consider is a partitioning protocol that is described solely by the root density:

$$\rho_{x,0}(p) = \frac{1 - \Theta_x}{4\pi} + \Theta_x \rho^R(p), \qquad \psi_{x,0}(p) = 0, \tag{49}$$

where $\rho^R(p) = \frac{1}{2\pi} \frac{1}{1+e^{\beta\epsilon(p)}}$. Here $\Theta_x$ is the discrete step-function defined for $x \in \mathbb{Z}/2$, whose integral representation is given by

$$\Theta_x = \int_{-\pi}^{+\pi} \frac{d\tau}{2\pi} \frac{e^{2ix\tau}}{1 - e^{-i(\tau-i0)}} = \int_{\mathcal{C}} \frac{dz}{2\pi i} \frac{z^{2x}}{z - 1}, \tag{50}$$

where $\mathcal{C}$ is a closed path winding once around the origin and enclosing the unitary circle (note that $\Theta_0 = 1$). This is a partitioning protocol between a thermal state with zero inverse temperature to the left and a state with inverse temperature $\beta$ to the right. Note that it is not sharp, in the sense that there are spin to the left of the origin whose correlation with some spin to the right of the origin is not exactly zero. Note also that, since the homogeneous states that form the partitioning protocols are defined in terms of the root density only and the root density of homogeneous states does not evolve in time, the non-trivial dynamics happens only in the proximity of the junction – see Fig. 1.

### 5.2.2 Pure-auxiliary-field state

We set here

$$\rho_{x,0}(p) = \frac{1}{4\pi}, \qquad \psi_{x,0}(p) = \Theta_x \psi^R(p), \tag{51}$$

where $\psi^R(p) = -\frac{i}{2} \sin(p) \sqrt{1 + h^2 - 2h\cos(p)}$. Choosing $\rho_{x,0}(p) = \frac{1}{4\pi}$ is equivalent to impose that the contribution to the correlation matrix coming from the root density is the same that one gets for the infinite-temperature state, i.e. zero. That is why we refer to this state as *pure-auxiliary-field state*. Note that the correlation matrix of the homogeneous state defined by $\rho_{x,0}(p) = \frac{1}{4\pi}$ and $\psi_{x,t}(p) = \psi^R(p)$ has elements

$$\Gamma^0 = \sigma^y/4, \qquad \Gamma^{\pm 1} = \mp i\sigma^x/4, \qquad \Gamma^{\pm 2} = \pm i\sigma^{\pm}/4, \qquad \Gamma^z = 0 \text{ if } |z| > 2, \tag{52}$$

which describe an initial state with local correlations (for the sake of our discussion, the only important thing is that the correlations decay at least exponentially with distance). Unlike the previous example, this state evolves also far from the junction. Indeed, whenever there is a non-zero component in any field except $\rho_{x,t}(p)$, the homogeneous part evolves too, so we effectively have a global quench far to the right of the junction, while around the junction a lightcone structure still emerges, as we will discuss in more details later on.

### 5.2.3   Non-Gaussian state

The last partitioning protocol that we introduce is a partitioning protocol in which the state is described only in terms of the non-Gaussian fields. We define it via the density matrix

$$\hat{\rho} = Z^{-1} e^{\beta_R \sum_{\ell=0}^{+\infty} \sigma_\ell^z \sigma_{\ell+1}^z}, \tag{53}$$

where $Z$ is the normalization constant such that $\mathrm{Tr}(\hat{\rho}) = 1$. This state has already been studied in Ref. [69] and we denote it as *rotated-Ising thermal state* (RITS), since it is the thermal state of an Ising model with the axis of the nearest-neighbor interaction that is rotated with respect to the one in our Hamiltonian (1).

What is convenient about the RITS is that the computation of the expectation value of any string of $\sigma^z$ is essentially classic, since everything is diagonal in the $z$ basis. In particular, $\langle \sigma_\ell^z \rangle_{t=0} = 0 \; \forall \ell$, and

$$\langle \sigma_m^z \sigma_n^z \rangle_{t=0} = (\tanh\beta)^{|m-n|} \Theta_m \Theta_n. \tag{54}$$

From here, one can compute the 2-point and 4-point Majorana correlation functions in the operators $a$, and then, writing the Bogoliubov fermions in terms of the Majorana fermions, arrive at the 4-point connected correlation functions in Bogoliubov fermions, which allow us to compute the value of the non-Gaussian dynamical fields at time zero (see Appendix C). It can be shown that the Gaussian part of the state, i.e. the correlation matrix, is zero throughout the whole chain, or, equivalently, $\rho_{x,t}(p) = \frac{1}{4\pi}$ and $\psi_{x,t}(p) = 0$, but the connected four point function is non-zero. Note that, differently from the Gaussian partitioning protocols above, the partitioning protocol that we have just defined is sharp, in the sense that the correlations between spins belonging to different halves of the chain are zero. The downside is that the state excites all the non-Gaussian fields (so it is not described by one field only), but this is not a problem, since we are mainly interested in comparing the non-Gaussian contribution with the one from the root density.

## 6   Root-density partitioning protocol

### 6.1   Integral representation of the correlation matrix

In this section we perform the asymptotic analysis of the spin-spin connected correlation function (47) in the case of the partitioning protocol (49) defined in terms of the root density only, evolving under the TFIC Hamiltonian (1). To do so, we first focus on the asymptotics of the correlation matrix in this partitioning protocol.

To extract the contribution coming from the root density we start with the inhomogeneous symbol of the correlation matrix (29) and we consider only the part that involves the root density:

$$\hat{\Gamma}_{x,t}^\rho(p) := e^{-i\frac{\theta(p)}{2}\sigma^z} \star \left( (4\pi\rho_{x,t;e}(p) - 1)\,\sigma^y + 4\pi\rho_{x,t;o}(p)\mathbb{I}_2 \right) \star e^{i\frac{\theta(p)}{2}\sigma^z}. \tag{55}$$

Using the representation (31) of the Moyal product, we can rewrite it as

$$\hat{\Gamma}^\rho_{x,t}(p) = i \begin{pmatrix} 0 & e^{-i\theta(p)} \\ -e^{i\theta(p)} & 0 \end{pmatrix} + 2 \sum_{y\in\mathbb{Z}/2} \int_{-\pi}^{\pi} dq \ e^{2iq(x-y)} \times$$

$$\times \begin{pmatrix} e^{i\frac{\theta(p-q)-\theta(p+q)}{2}}\rho_{y,t;o}(p) & e^{-i\frac{\theta(p-q)+\theta(p+q)}{2}}i\rho_{y,t;e}(p) \\ e^{i\frac{\theta(p-q)+\theta(p+q)}{2}}i\rho_{y,t;e}(p) & e^{-i\frac{\theta(p-q)-\theta(p+q)}{2}}\rho_{y,t;o}(p) \end{pmatrix}. \quad (56)$$

The contribution of the root density to the correlation matrix is simply the correlation matrix that one gets using the inversion relation (25) on the inhomogeneous symbol $\hat{\Gamma}^\rho_{x,t}(p)$, which gives

$$(\Gamma^{z;\rho}_x(t))_{1+\frac{1\mp1}{2},1+\frac{1\mp1}{2}} =$$

$$= 4\pi i \, \mathrm{Re} \int_{-\pi}^{\pi} \frac{dpdq}{(2\pi)^2} \sin(pz) e^{2iqx\pm i\frac{\theta(p-q)-\theta(p+q)}{2}-it\epsilon(p+q)+it\epsilon(p-q)} \left( \sum_{y\in\mathbb{Z}/2} e^{-2iqy}\rho_y(p,0) \right),$$

$$(\Gamma^{z;\rho}_x(t))_{21} = -i \int_{-\pi}^{\pi} \frac{dp}{2\pi} \cos(pz + \theta(p)) +$$
$$\quad (57)$$

$$+ 4\pi i \int_{-\pi}^{\pi} \frac{dpdq}{(2\pi)^2} \cos\left( pz + \frac{\theta(p-q)+\theta(p+q)}{2} \right) e^{2iqx-it\epsilon(p+q)+it\epsilon(p-q)} \left( \sum_{y\in\mathbb{Z}/2} e^{-2iqy}\rho_y(p,0) \right).$$

Finally, plugging in the expression (49) for $\rho_{x,t}(p)$, we obtain

$$(\Gamma^{z;\rho}_x(t))_{1+\frac{1\mp1}{2},1+\frac{1\mp1}{2}} =$$

$$= 4\pi i \, \mathrm{Re} \int_{-\pi}^{\pi} \frac{dpdq}{(2\pi)^2} \frac{e^{2i(p-q)x-it\epsilon(2p)+it\epsilon(2q)}}{1-e^{-i(p-q-i0)}} \sin((p+q)z)e^{\pm i\frac{\theta(2q)-\theta(2p)}{2}} \left( \rho^R(p+q) - \frac{1}{4\pi} \right),$$

$$(\Gamma^{z;\rho}_x(t))_{21} = -\left( \Gamma^{-z;\rho}_x(t) \right)_{12} =$$

$$= 4\pi i \int_{-\pi}^{\pi} \frac{dpdq}{(2\pi)^2} \frac{e^{2i(p-q)x-it\epsilon(2p)+it\epsilon(2q)}}{1-e^{-i(p-q-i0)}} \cos\left( (p+q)z + \frac{\theta(2q)+\theta(2p)}{2} \right) \left( \rho^R(p+q) - \frac{1}{4\pi} \right),$$
$$\quad (58)$$

where we used

$$\sum_{y\in\mathbb{Z}/2} e^{-2iqy}\rho_y(p,0) = \frac{1}{2}\delta(q) + \sum_{y\in\mathbb{Z}/2} \int_{-\pi}^{+\pi} \frac{d\tau}{2\pi} e^{-2iqy} \left( \rho^R(p) - \frac{1}{4\pi} \right) \frac{e^{2iy\tau}}{1-e^{-i(\tau-i0)}} =$$

$$= \frac{1}{2}\delta(q) + \frac{\rho^R(p) - \frac{1}{4\pi}}{1-e^{-i(q-i0)}} \quad (59)$$

and the property $\int_{-\pi}^{\pi} dpdq \ f(p,q) = \int_{-\pi}^{\pi} dpdq \ f(p+q, p-q)$, that holds for functions such that $f(p,q) = f(p+\pi, q) = f(p, q+\pi)$.

To study the asymptotic behavior of this double integral, it is convenient to change variables as $w_1 = e^{ip}, w_2 = e^{i(q+i0)}$ and switch to the complex plane. We obtain

$$(\Gamma^{z;\rho}_x(t))_{1+\frac{1\mp1}{2},1+\frac{1\mp1}{2}} = -2\pi i \, \mathrm{Im} \int_{\mathcal{C}_1} \frac{dw_1}{2\pi} \int_{\mathcal{C}_2} \frac{dw_2}{2\pi w_2} \left( \rho^R(-i\ln(w_1 w_2)) - \frac{1}{4\pi} \right) \times$$

$$\times e^{\pm i\frac{\theta(-2i\ln w_2)-\theta(-2i\ln w_1)}{2}} \frac{e^{itE(w_2^2)-itE(w_1^2)}}{w_1 - w_2} \left( w_1^{2x+z}w_2^{-2x+z} - w_1^{2x-z}w_2^{-2x-z} \right), \quad (60)$$

$$(\Gamma_x^{z;\rho}(t))_{21} = -\left(\Gamma_x^{-z;\rho}(t)\right)_{12} = -2\pi i \int_{\mathcal{C}_1} \frac{\mathrm{d}w_1}{2\pi} \int_{\mathcal{C}_2} \frac{\mathrm{d}w_2}{2\pi w_2} \left(\rho^R(-i\ln(w_1 w_2)) - \frac{1}{4\pi}\right)$$

$$\frac{e^{itE(w_2^2) - itE(w_1^2)}}{w_1 - w_2} \left(w_1^{2x+z} w_2^{-2x+z} e^{i\frac{\theta(-2i\ln w_2) + \theta(-2i\ln w_1)}{2}} + w_1^{2x-z} w_2^{-2x-z} e^{-i\frac{\theta(-2i\ln w_2) + \theta(-2i\ln w_1)}{2}}\right),$$

$$(61)$$

where

$$E(w) := 2J\sqrt{(1-hw)(1-h/w)} \tag{62}$$

is the dispersion relation in complex variables, and $\mathcal{C}_1$ and $\mathcal{C}_2$ are closed curve with winding number 1 around the origin, in an annulus around the unit circle, and such that $\mathcal{C}_2$ is inside the region delimited by $\mathcal{C}_1$. For the function $E(w)$, we choose the branch cut to be on the real axis in the interval $[0, \min\{1/h, h\}] \cup [\max\{1/h, h\}, +\infty)$,[4] where we recall that we assumed that the model is gapped, and hence $h \neq 1$; in this way the function is analytic in an annulus including the unit circle. Note that $2x \pm z \in 2\mathbb{Z}$, so that the functions $w^{s_1 2x + s_2 z}$ do not have any branch cut $\forall s_1, s_2 \in \{\pm 1\}$. As for $\rho^R(-i\ln w)$ and $e^{i\theta(-i\ln w)}$, they are also analytic in an annulus including the unit circle, since they are analytic function of $\epsilon(p)$ in the annulus. That is in general true for any noncritical state satisfying clustering. We also mention that the computation could be carried out also without introducing complex variables and sticking to the momentum space, in a similar fashion to what was done in Ref. [70] for a similar asymptotics.

We can recast the expressions above in a suitable form for the application of the saddle-point method:

$$(\Gamma_x^{z;\rho})_{1+\frac{1\mp 1}{2}, 1+\frac{1\mp 1}{2}} = -2\pi i \operatorname{Im} \int_{\mathcal{C}_1} \frac{\mathrm{d}w_1}{2\pi} \int_{\mathcal{C}_2} \frac{\mathrm{d}w_2}{2\pi w_2(w_1 - w_2)} \left(\rho^R(-i\ln(w_1 w_2)) - \frac{1}{4\pi}\right) \times$$

$$\times e^{\pm i \frac{\theta(-2i\ln w_2) - \theta(-2i\ln w_1)}{2}} \left(e^{tS_{(x+\frac{z}{2})/t}(w_1) - tS_{(x-\frac{z}{2})/t}(w_2)} - e^{tS_{(x-\frac{z}{2})/t}(w_1) - tS_{(x+\frac{z}{2})/t}(w_2)}\right)$$

$$(\Gamma_x^{z;\rho})_{21} = -\left(\Gamma_x^{-z;\rho}\right)_{12} = -2\pi i \int_{\mathcal{C}_1} \frac{\mathrm{d}w_1}{2\pi} \int_{\mathcal{C}_2} \frac{\mathrm{d}w_2}{2\pi w_2(w_1 - w_2)} \left(\rho^R(-i\ln(w_1 w_2)) - \frac{1}{4\pi}\right) \times \quad (63)$$

$$\times \left(e^{tS_{(x+\frac{z}{2})/t}(w_1) - tS_{(x-\frac{z}{2})/t}(w_2)} e^{i\frac{\theta(-2i\ln w_2) + \theta(-2i\ln w_1)}{2}} + \right.$$

$$\left. + e^{tS_{(x-\frac{z}{2})/t}(w_1) - tS_{(x+\frac{z}{2})/t}(w_2)} e^{-i\frac{\theta(-2i\ln w_2) + \theta(-2i\ln w_1)}{2}}\right),$$

where

$$S_\zeta(w) := 2\zeta \ln(w) - iE(w^2) \tag{64}$$

and $E(w)$ is the complex dispersion relation defined in (62). In the following we discuss how to compute the leading contribution to the integral in the different regimes defined by $\alpha$.

## 6.2  Asymptotic analysis

Here we compute the asymptotics of the integrals (63) in the scaling limit $t \to +\infty$, with $x = \zeta t, z = ct^\alpha > 0, \alpha \in [0,1], \zeta \neq 0$. For any value of $\alpha$ the strategy is the same: we deform

---

[4]To choose the branch cuts, here it is convenient to isolate the singular point, i.e. the origin, so we write $E(w) = 2J\sqrt{\frac{(1-hw)(w-h)}{w}}$. We can choose the branch cut of the numerator to be on the real axis in the interval $[\min\{1/h, h\}, \max\{1/h, h\}]$ and the branch cut of the denominator to be in $[0, +\infty)$.

the integration contours $\mathcal{C}_1$ to $\mathcal{C}'_1$ and $\mathcal{C}_2$ to $\mathcal{C}'_2$, trying to make the function under integration exponentially small for almost all $w_1 \in \mathcal{C}'_1$ and $w_2 \in \mathcal{C}'_2$. To do so we need to study the function at the exponent, multiplying the large parameter.

Since there are two possible options for this function, i.e. $S_{(x\pm\frac{z}{2})/t}(w_1) - S_{(x\mp\frac{z}{2})/t}(w_2)$, we found it convenient to split our analysis in two: we define

$$\mathcal{I}_1^\rho[f(w_1, w_2)](x, z, t) := \int_{\mathcal{C}_1} \frac{\mathrm{d}w_1}{2\pi} \int_{\mathcal{C}_2} \frac{\mathrm{d}w_2}{2\pi} \frac{\mathrm{e}^{tS_{(x+\frac{z}{2})/t}(w_1) - tS_{(x-\frac{z}{2})/t}(w_2)}}{w_1 - w_2} f(w_1, w_2), \qquad (65)$$

$$\mathcal{I}_2^\rho[f(w_1, w_2)](x, z, t) := \int_{\mathcal{C}_1} \frac{\mathrm{d}w_1}{2\pi} \int_{\mathcal{C}_2} \frac{\mathrm{d}w_2}{2\pi} \frac{\mathrm{e}^{tS_{(x-\frac{z}{2})/t}(w_1) - tS_{(x+\frac{z}{2})/t}(w_2)}}{w_1 - w_2} f(w_1, w_2), \qquad (66)$$

where $f(w_1, w_2)$ is an analytic function in an annulus around the unit circle and $\mathcal{C}_1$ and $\mathcal{C}_2$ are defined as above. In this way

$$(\Gamma_x^{z;\rho})_{1+\frac{1\mp1}{2},1+\frac{1\mp1}{2}} =$$
$$= 2\pi\mathrm{i}\operatorname{Im}\mathcal{I}_1^\rho\left[\frac{1}{w_2}\left(\frac{1}{4\pi} - \rho^R(-\mathrm{i}\ln(w_1 w_2))\right)\mathrm{e}^{\pm\mathrm{i}\frac{\theta(-2\mathrm{i}\ln w_2)-\theta(-2\mathrm{i}\ln w_1)}{2}}\right](x, z, t) +$$
$$- 2\pi\mathrm{i}\operatorname{Im}\mathcal{I}_2^\rho\left[\frac{1}{w_2}\left(\frac{1}{4\pi} - \rho^R(-\mathrm{i}\ln(w_1 w_2))\right)\mathrm{e}^{\pm\mathrm{i}\frac{\theta(-2\mathrm{i}\ln w_2)-\theta(-2\mathrm{i}\ln w_1)}{2}}\right](x, z, t), \quad (67)$$

$$(\Gamma_x^{z;\rho})_{21} = -\left(\Gamma_x^{-z;\rho}\right)_{12} =$$
$$= 2\pi\mathrm{i}\mathcal{I}_1^\rho\left[\frac{1}{w_2}\left(\frac{1}{4\pi} - \rho^R(-\mathrm{i}\ln(w_1 w_2))\right)\mathrm{e}^{\mathrm{i}\frac{\theta(-2\mathrm{i}\ln w_2)+\theta(-2\mathrm{i}\ln w_1)}{2}}\right](x, z, t) +$$
$$- 2\pi\mathrm{i}\mathcal{I}_2^\rho\left[\frac{1}{w_2}\left(\frac{1}{4\pi} - \rho^R(-\mathrm{i}\ln(w_1 w_2))\right)\mathrm{e}^{-\mathrm{i}\frac{\theta(-2\mathrm{i}\ln w_2)+\theta(-2\mathrm{i}\ln w_1)}{2}}\right](x, z, t).$$

In the following we consider $\mathcal{I}_1^\rho$ and $\mathcal{I}_2^\rho$ separately.

### 6.2.1 Linear scaling of the distance between spins

We start by considering $\mathcal{I}_1^\rho[f(w_1, w_2)](x, z, t)$, defined in Eq. (65), assuming $\alpha = 1$. The idea is the following. We identify a path $\mathcal{C}'_1$ such that the real part of $S_{(x+\frac{z}{2})/t}(w_1)$ is negative almost everywhere $\forall w_1 \in \mathcal{C}'_1$, in such a way that $\mathrm{e}^{tS_{(x+\frac{z}{2})/t}(w_1)}$ is exponentially suppressed for $t \to +\infty$. Similarly, we identify a path $\mathcal{C}'_2$ such that the real part of $S_{(x-\frac{z}{2})/t}(w_2)$ is positive almost everywhere $\forall w_2 \in \mathcal{C}'_2$, in such a way that $\mathrm{e}^{-tS_{(x-\frac{z}{2})/t}(w_2)}$ is also exponentially suppressed. Then we deform $\mathcal{C}_1$ to $\mathcal{C}'_1$ and $\mathcal{C}_2$ to $\mathcal{C}'_2$. In the deformation process, the contours may exchange, in which case we get a residue contribution from the pole $w_1 = w_2$. For concreteness, let us go through some specific examples.

We start by looking at the case $-v_{max} < x/t < v_{max}$ and $z/t > 2v_{max}$, as in the leftmost plot of panel (a) of Figure 2. In the figure we show that the two curves $\mathcal{C}'_1$ (in black) and $\mathcal{C}'_2$ (in red) as described above exist. In this case, $\mathcal{C}'_2$ is contained in the region enclosed by $\mathcal{C}'_1$. Since that is true also for the corresponding initial paths, i.e. $\mathcal{C}_2$ is contained in the region enclosed by $\mathcal{C}_1$, the deformation of $\mathcal{C}_1$ to $\mathcal{C}'_1$ and $\mathcal{C}_2$ to $\mathcal{C}'_2$ can be done keeping the former always outside the second, so that the singularity $w_1 = w_2$ is avoided. Therefore there is not any residue

contribution in this case. We end up with a double integral over the paths $\mathcal{C}'_1$ and $\mathcal{C}'_2$ of an exponentially suppressed function which leads to an exponentially suppressed contribution.

Let us consider now $\zeta \pm \frac{z}{2t} > v_{max}$, as in the rightmost plot of panel (b) in Figure 2. Again the figure shows that the two curves $\mathcal{C}'_1$ (in black) and $\mathcal{C}'_2$ (in red) as described above exist. However, this time the external curve is $\mathcal{C}'_2$ and thus, in the deformation process of $\mathcal{C}_1$ to $\mathcal{C}'_1$ and $\mathcal{C}_2$ to $\mathcal{C}'_2$, the two contours are fully exchanged, hitting the singularity $w_1 = w_2$. A convenient way to compute the residue contribution is the following. First, we deform $\mathcal{C}_1$ to a path $\mathcal{C}''_1$ that contains both $\mathcal{C}_2$ and $\mathcal{C}'_2$. Second, we deform $\mathcal{C}_2$ to $\mathcal{C}'_2$; this can be done without hitting the singularity $w_1 = w_2$ because of how $\mathcal{C}''_1$ is defined. Finally, we deform $\mathcal{C}''_1$ to $\mathcal{C}'_1$; in this deformation we necessarily hit the singularity $w_1 = w_2$. To compute the contribution coming from the singularity we can consider $w_2 \in \mathcal{C}'_2$ as fixed and use

$$
\int_{\mathcal{C}''_1} \mathrm{d}w_1 \frac{F(w_1, w_2)}{w_1 - w_2} = \int_{\mathcal{C}'_1} \mathrm{d}w_1 \frac{F(w_1, w_2)}{w_1 - w_2} + 2\pi\mathrm{i}\, \mathrm{Res}_{w_2} \frac{F(w_1, w_2)}{w_1 - w_2} =
$$
$$
= \int_{\mathcal{C}'_1} \mathrm{d}w_1 \frac{F(w_1, w_2)}{w_1 - w_2} + 2\pi\mathrm{i} F(w_2, w_2), \quad (68)
$$

for any function $F(w_1, w_2)$ that is analytic for $w_1$ and $w_2$ in a connected region containing both $\mathcal{C}''_1$ and $\mathcal{C}'_1$. Here we used that $w_2$ is inside the region delimited by $\mathcal{C}''_1$ and outside the one delimited by $\mathcal{C}'_1$. Applying the result above to our case we get

$$
\mathcal{I}^\rho_1[f(w_1, w_2)](x, z, t) = \int_{\mathcal{C}'_1} \frac{\mathrm{d}w_1}{2\pi} \int_{\mathcal{C}'_2} \frac{\mathrm{d}w_2}{2\pi} \frac{\mathrm{e}^{tS_{(x+\frac{z}{2})/t}(w_1) - tS_{(x-\frac{z}{2})/t}(w_2)}}{w_1 - w_2} f(w_1, w_2) +
$$
$$
+ \mathrm{i} \int_{\mathcal{C}'_2} \frac{\mathrm{d}w}{2\pi} \mathrm{e}^{tS_{(x+\frac{z}{2})/t}(w) - tS_{(x-\frac{z}{2})/t}(w)} f(w, w). \quad (69)
$$

The first term is exponentially suppressed, since the function under integration is exponentially suppressed by definition of the integration contours. Let us then focus on the residue contribution. First of all, note that $S_{(x+\frac{z}{2})/t}(w) - S_{(x-\frac{z}{2})/t}(w) = 2\frac{z}{t} \ln w$, whose real is negative inside the unit circle (recall we are assuming $z > 0$), where $\mathcal{C}'_2$ is defined. This means that also the residue contribution is exponentially suppressed in the limit $t \to +\infty$. In conclusion, in this case the whole $\mathcal{I}^\rho_1[f(w_1, w_2)](x, z, t)$ is exponentially suppressed.

The last special case that we are going to analyze is $-v_{max} < \zeta \pm \frac{z}{2t} < v_{max}$, as in the panel (c) of Fig. 2. As in the previous cases, the two paths $\mathcal{C}'_1$ and $\mathcal{C}'_2$ as described above exist; they are reported in panel (c) of Fig. 2 in black and red respectively. What distinguishes this case from the previous ones is that the two paths are only partially exchanged. We still use the method described above and Eq. (68) to describe the residue contribution, but this time we do not get a residue contribution $\forall w_2 \in \mathcal{C}'_2$. Indeed, we get the residue contribution from Eq. (68) only for those points $w_2 \in \mathcal{C}'_2$ that are inside $\mathcal{C}''_1$ but not $\mathcal{C}'_1$. Eventually, this leads to

$$
\mathcal{I}^\rho_1[f(w_1, w_2)](x, z, t) = \int_{\mathcal{C}'_1} \frac{\mathrm{d}w_1}{2\pi} \int_{\mathcal{C}'_2} \frac{\mathrm{d}w_2}{2\pi} \frac{\mathrm{e}^{tS_{(x+\frac{z}{2})/t}(w_1) - tS_{(x-\frac{z}{2})/t}(w_2)}}{w_1 - w_2} f(w_1, w_2) +
$$
$$
+ \mathrm{i} \int_{\gamma} \frac{\mathrm{d}w}{2\pi} \mathrm{e}^{tS_{(x+\frac{z}{2})/t}(w) - tS_{(x-\frac{z}{2})/t}(w)} f(w, w), \quad (70)
$$

where $\gamma$ is homotopy-equivalent to the part of $\mathcal{C}'_2$ that is external to $\mathcal{C}'_1$. As in the previous case, since $\gamma$ can be defined inside the unit circle, where $\mathrm{Re}(S_{(x+\frac{z}{2})/t}(w_1) - S_{(x-\frac{z}{2})/t}(w_2)) < 0$,

the residue contribution is exponentially suppressed. Therefore, we are left with the double integral over $\mathcal{C}_1'$ and $\mathcal{C}_2'$, where the function under integration is exponentially suppressed everywhere except in four saddle points for each contour. The leading contribution from those points can be evaluated with a standard application of the saddle-point method, leading to $\mathcal{I}_1^\rho[f(w_1, w_2)](x, z, t) \sim t^{-1}$.

One can go through all the other cases and show that a deformation of $\mathcal{C}_1$ to $\mathcal{C}_1'$ and $\mathcal{C}_2$ to $\mathcal{C}_2'$ as the one envisaged above always exists. Some other special cases are reported in Fig. 2. In the deformation process, the initial contours may exchange, and in such cases we get a residue contribution from the pole $w_1 = w_2$, to be integrated along a curve that has the intersections between $\mathcal{C}_1'$ and $\mathcal{C}_2'$ as extrema (if there is no intersection, because the contours fully exchange, then the curve is a closed curve around the origin). However, such a residue contribution is negligible. As for the remaining double integral over $\mathcal{C}_1'$ and $\mathcal{C}_2'$, if both the rays $(x \pm z/2)/t$ are inside the lightcone, there are four saddle points to be considered for each of the two integrals, so that the precise computation of the leading order is lengthy but straightforward; we will not report it here, but it can be concluded that $\mathcal{I}_1^\rho[f(w_1, w_2)](x, z, t) \sim t^{-1}$ for general values of the parameters. If instead at least one of the two rays is outside the lightcone, at least one of the two integrals in the double integral has no saddle point and the integration's domain can be deformed to a region where the integral is exponentially suppressed, consistently with the Lieb-Robinson bound. Note incidentally that this is the same reason for which, in the case $\alpha > 1$, we have an exponentially small contribution for any value of the rays.

An analogous result holds for the term $\mathcal{I}_2[f(w_1, w_2)](x, z, t)$, as reported in Appendix A: if the rays $(x \pm \frac{z}{2})/t$ are inside the lightcone, $\mathcal{I}_2[f(w_1, w_2)](x, z, t) \sim t^{-1}$, otherwise it is exponentially suppressed. The two contributions are eventually combined as in Eq. (67) to get the final asymptotics.

### 6.2.2 Sub-linear scaling of the distance between spins

Let us now consider the case $0 < \alpha < 1$, under the assumption that the two rays $(x \pm z/2)/t$ are inside the lightcone, since if at least one ray is outside the lightcone the argument we discussed for $\alpha = 1$ still holds and we get an exponentially small contribution. Again, we start with $\mathcal{I}_1^\rho$.

The considerations made above for the case $\alpha = 1$ for the contour deformations still hold, the difference is that the extrema of the curve for the residue contribution asymptotically collapse to the unit circle (one can see that by solving the saddle-point equation for $z/t = 0$). That means that the residue contribution can not be dropped as before. Moreover, the two sets of saddle points corresponding to the integrals over $\mathcal{C}_1'$ and $\mathcal{C}_2'$ are asymptotically the same, so, expanding around those, we may end up with a denominator that goes to zero, so that the contribution that we get from $\mathcal{I}_1^{\rho'}$ may also be different.

We denote $e^{i\bar{p}_j}$, for $j \in \{1, 2, 3, 4\}$, the four saddle points of $S_{x/t}(w)$. The saddle points of $S_{(x+\frac{z}{2})/t}(w)$ are a correction of order $t^{\alpha-1}$ to $e^{i\bar{p}_j}$ and they are still on the unit circle for any value $z/t$ as long as the ray $(x + \frac{z}{2})/t$ is inside the lightcone. We denote them $e^{i\bar{p}_j + i\delta p_j}$, with $\delta p_j \sim t^{\alpha-1}$. As a consequence of linearity of the saddle points in small $z/t$, we have that the saddle points of $S_{(x-\frac{z}{2})/t}(w)$ are $e^{i\bar{p}_j - i\delta p_j}$.

We start by discussing the residue contribution. Resuming from Eq. (70), we define the residue contribution as

$$\mathcal{R}_1^\rho[f(w_1, w_2)](z, t) \equiv i \int_\gamma \frac{dw}{2\pi} e^{2z \ln w} f(w, w), \tag{71}$$

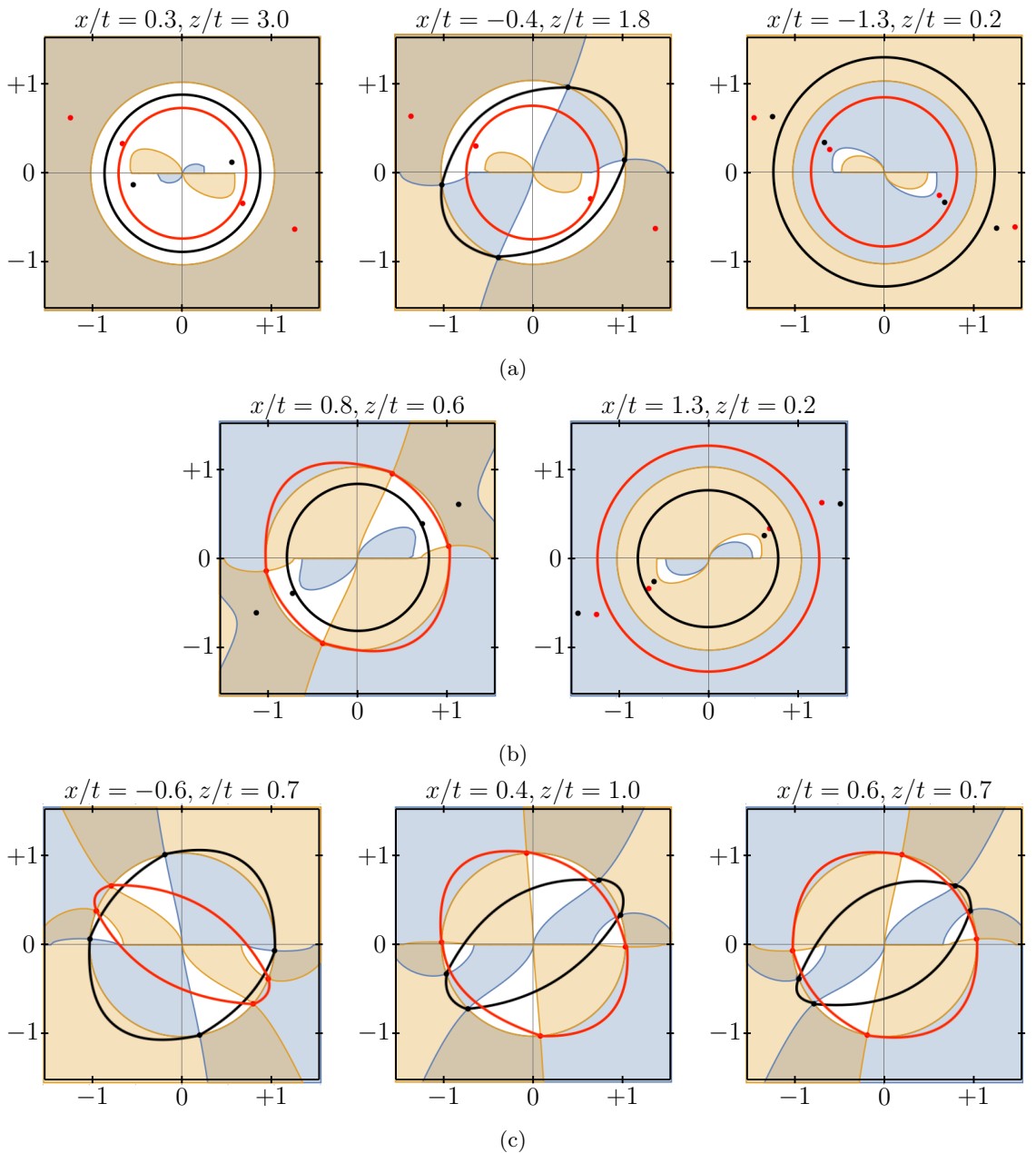

Figure 2: Regions in which $\mathrm{Re}(S_{(x+\frac{z}{2})/t}(a+\mathrm{i}b)) > 0$ (blue region) and $\mathrm{Re}(S_{(x-\frac{z}{2})/t}(a+\mathrm{i}b)) < 0$ (orange region) in function of $a$ (horizontal axis) and $b$ (vertical axis). We consider a TFIC with $J = 1/2$, so that $v_{max} = 1$, and $h = 2$. The black and red lines represent respectively the curves $\mathcal{C}_1'$ and $\mathcal{C}_2'$ obtained by deforming $\mathcal{C}_1$ and $\mathcal{C}_2$ as detailed in the text. (a) Case in which the ray $(x - \frac{z}{2})/t, z > 0$, is on the left of the lightcone. (b) Case in which the ray $(x + \frac{z}{2})/t, z > 0$, is on the right of the lightcone and $(x - \frac{z}{2})/t$ is either inside or to the right of it. (c) Case in which the two rays $(x \pm \frac{z}{2})/t$ are both inside the lightcone.

where $\gamma$ is homotopy-equivalent to the part of $\mathcal{C}_2'$ that is external to $\mathcal{C}_1'$. Note that there is an implicit dependence on $x/t$ in the definition of $\gamma$. Here we can already make a simplification. As can be inferred from the the picture, in general, the residue contribution comes in the form of two integrals along two curves that are related by $w \leftrightarrow -w$. Since the function under integration is odd and the two curves are traveled in opposite direction, the two contributions are the same and we can actually consider just one of them, so that

$$\mathcal{R}_1^\rho[f(w_1, w_2)](z, t) = 2\mathrm{i} \int_\gamma \frac{\mathrm{d}w}{2\pi} \mathrm{e}^{2z \ln w} f(w, w), \tag{72}$$

where now $\gamma$ is homotopy-equivalent to just one of the two continuous curves identified by the part of $\mathcal{C}_2'$ that is external to $\mathcal{C}_1'$.

Since the distance between the saddle points in the integral over $\mathcal{C}_1'$ and the corresponding saddle points in the integral over $\mathcal{C}_2'$ goes to zero as $t^{\alpha-1}$, also the distance from the unit circle of the points where $\mathcal{C}_1'$ and $\mathcal{C}_2'$ scales as $t^{\alpha-1}$, in the inward direction. Then the curve $\gamma$ for the residue contribution can be deformed in such a way that the real part of the exponent takes finite negative values almost everywhere, except around its extrema, where it goes to zero. So the only non-exponentially-suppressed contribution can come from those neighborhoods. Therefore we linearize the residue contribution around its extrema. We denote $\mathrm{e}^{\mathrm{i}\bar{p}_j} - \delta w_j \mathrm{e}^{\mathrm{i}\bar{p}_j}$, with $\delta w \sim t^{\alpha-1}$, the extrema of $\gamma$, and we parametrize $\gamma$ in the neighborhood of its extrema as $\mathrm{e}^{\mathrm{i}\bar{p}_j} - \delta w_j \mathrm{e}^{\mathrm{i}\bar{p}_j} - s\mathrm{e}^{\mathrm{i}\bar{p}_j}$, with $s \in [-\Delta, \Delta]$, $\Delta \sim t^{-\omega}$ with $\omega \in (\alpha/2, \alpha)$. In this way we have

$$\mathcal{R}_1^\rho[f(w_1, w_2)](z, t) \sim 2\mathrm{i} \sum_{j=1}^2 (-1)^j \int_0^\Delta \frac{\mathrm{d}s}{2\pi} \mathrm{e}^{\mathrm{i}\bar{p}_j} \mathrm{e}^{2z \ln(\mathrm{e}^{\mathrm{i}\bar{p}_j} - \delta w_j \mathrm{e}^{\mathrm{i}\bar{p}_j} - s\mathrm{e}^{\mathrm{i}\bar{p}_j})} f(\mathrm{e}^{\mathrm{i}\bar{p}_j}, \mathrm{e}^{\mathrm{i}\bar{p}_j}) =$$

$$= 2\mathrm{i} \sum_{j=1}^2 (-1)^j f(\mathrm{e}^{\mathrm{i}\bar{p}_j}, \mathrm{e}^{\mathrm{i}\bar{p}_j}) \mathrm{e}^{\mathrm{i}\bar{p}_j} \int_0^\Delta \frac{\mathrm{d}s}{2\pi} (\mathrm{e}^{\mathrm{i}\bar{p}_j} - \delta w_j \mathrm{e}^{\mathrm{i}\bar{p}_j} - s\mathrm{e}^{\mathrm{i}\bar{p}_j})^{2z} =$$

$$= 2\mathrm{i} \sum_{j=1}^2 (-1)^{j-1} f(\mathrm{e}^{\mathrm{i}\bar{p}_j}, \mathrm{e}^{\mathrm{i}\bar{p}_j}) \frac{\mathrm{e}^{\mathrm{i}(2z+1)\bar{p}_j}}{2\pi(2z+1)} (1 - \delta w_j - s)^{2z+1}|_0^\Delta. \tag{73}$$

At this point we use that, for $0 < \alpha < 1/2$, $(1 - \delta w_j - \Delta)^{2z} \sim (1 - \Delta)^{2z} \sim \mathrm{e}^{-2z\Delta}$ to conclude that the term containing $\Delta$ goes to zero exponentially. We also use that in this regime $(1 - \delta w_j)^{2z} \sim 1$ to finally arrive at

$$\mathcal{R}_1^\rho[f(w_1, w_2)](z, t) \sim \mathrm{i} \sum_{j=1}^2 (-1)^j \frac{1}{2\pi z} \mathrm{e}^{\mathrm{i}\bar{p}_j} \mathrm{e}^{\mathrm{i}2z\bar{p}_j} f(\mathrm{e}^{\mathrm{i}\bar{p}_j}, \mathrm{e}^{\mathrm{i}\bar{p}_j}) \tag{74}$$

for $0 < \alpha < 1/2$. We can use a similar argument to conclude that $\mathcal{R}_1^\rho$ is exponentially suppressed for $1/2 < \alpha < 1$. We put this result aside for a little while: we will now compute the double-integral contribution and then compare the two.

Let us turn to the double-integral contribution of Eq. (70), that we denote

$$\mathcal{I}_1^{\rho'}[f(w_1, w_2)](x, z, t) \equiv \int_{\mathcal{C}_1'} \frac{\mathrm{d}w_1}{2\pi} \int_{\mathcal{C}_2'} \frac{\mathrm{d}w_2}{2\pi} \frac{\mathrm{e}^{tS_{(x+\frac{z}{2})/t}(w_1) - tS_{(x-\frac{z}{2})/t}(w_2)}}{w_1 - w_2} f(w_1, w_2). \tag{75}$$

Here the leading contribution is obtained from the neighborhoods of the saddle-points. For $\alpha \in (0, 1)$, it is not a standard application of the saddle point method as it was in the case

$\alpha = 1$ because the saddle points of the two integrals tend to overlap, and the denominator of the function under integration tends to zero. As we have done to compute the residue contribution, we denote $e^{i\bar{p}_j \pm i\delta p_j}$ the saddle points of $S_{(x\pm\frac{z}{2})/t}$ and we parametrize the curves in the neighborhood of the saddle points as $e^{i\bar{p}_j \pm i\delta p_j} + s^\pm e^{i\phi_j^\pm}$, where $s^\pm \in [-\Delta, \Delta]$, $\Delta \sim t^\omega$ with $\omega \in (-1/2, -1/3)$, and $\phi_j^\pm$ to be determined in such a way to pass through the steepest-descent path. Then we have

$$\mathcal{I}_1^{\rho'}[f(w_1, w_2)](x, z, t) \sim \sum_{j=1}^4 \int_{-\Delta}^{\Delta} \frac{\mathrm{d}s^+ \mathrm{d}s^-}{(2\pi)^2} e^{i\phi_j^+} e^{i\phi_j^-} f(e^{i\bar{p}_j}, e^{i\bar{p}_j})$$

$$\frac{e^{tS_{(x+\frac{z}{2})/t}(e^{i\bar{p}_j + i\delta p_j}) - tS_{(x-\frac{z}{2})/t}(e^{i\bar{p}_j - i\delta p_j}) - \frac{t}{2}|S_{x/t}''(e^{i\bar{p}_j + i\delta p_j})|(s^+)^2 - \frac{t}{2}|S_{x/t}''(e^{i\bar{p}_j - i\delta p_j})|(s^-)^2}}{2i e^{i\bar{p}_j} \sin(\delta p_j) + s^+ e^{i\phi_j^+} - s^- e^{i\phi_j^-}}$$

$$\sim \sum_{j=1}^4 e^{tS_{(x+\frac{z}{2})/t}(e^{i\bar{p}_j + i\delta p_j}) - tS_{(x-\frac{z}{2})/t}(e^{i\bar{p}_j - i\delta p_j})} e^{i\phi_j^+} e^{i\phi_j^-} f(e^{i\bar{p}_j}, e^{i\bar{p}_j}) \times$$

$$\times \int_{-\Delta\sqrt{t}}^{\Delta\sqrt{t}} \frac{\mathrm{d}s^+ \mathrm{d}s^-}{(2\pi)^2} \frac{e^{-\frac{1}{2}|S_{x/t}''(e^{i\bar{p}_j + i\delta p_j})|(s^+)^2 - \frac{1}{2}|S_{x/t}''(e^{i\bar{p}_j - i\delta p_j})|(s^-)^2}}{2i e^{i\bar{p}_j} t\delta p_j + \sqrt{t}s^+ e^{i\phi_j^+} - \sqrt{t}s^- e^{i\phi_j^-}}, \quad (76)$$

where the phases $\phi_j^\pm$ satisfy

$$e^{i \arg(S_{(x\pm z/2)/t}''(e^{i\bar{p}_j \pm i\delta p_j})) + 2i\phi_j^\pm} \sim e^{i \arg(S_{x/t}''(e^{i\bar{p}_j})) + 2i\phi_j^\pm} = \mp 1, \quad (77)$$

and are defined in such a way that the direction is consistent with the initial contour. Introducing the real function

$$s_\zeta(p) := 2\zeta p - \epsilon(2p), \quad (78)$$

we have $S_\zeta(e^{ip}) = i s_\zeta(p)$ and we can compute

$$S_\zeta''(w) = i\partial_w^2 s_\zeta(-i \ln w) = \partial_w \left( s_\zeta'(-i \ln w)\frac{1}{w} \right) = s_\zeta''(-i \ln w)\frac{-i}{w^2} - s_\zeta'(-i \ln w)\frac{1}{w^2}, \quad (79)$$

from which

$$S_\zeta''(e^{i\bar{p}_j}) \sim -i e^{2i\bar{p}_j} s_\zeta''(\bar{p}_j) \quad \Rightarrow \quad e^{i \arg(S_\zeta''(e^{i\bar{p}_j}))} \sim e^{i(2\bar{p}_j - \pi/2 + \pi\Theta(-s_\zeta(\bar{p}_j)))}, \quad (80)$$

where we have used $S_\zeta'(e^{ip}) \sim 0 \Rightarrow s_\zeta'(p) \sim 0$. So, finally,

$$\phi_j^\pm = \bar{p}_j + \frac{\pi}{2} \pm \frac{\pi}{4} + \kappa_j^\pm \pi - \frac{\pi}{2}\Theta(-s_\zeta(\bar{p}_j)), \quad (81)$$

where $\kappa_j^\pm \in \mathbb{Z}$ are chosen in such a way to respect the direction of the integration contour. It can be shown that $s_{x/t}''(\bar{p}_1) > 0$, $s_{x/t}''(\bar{p}_2) < 0$ and $\kappa_1^\pm = \kappa_2^+ = 0$, $\kappa_2^- = 1$.

Now, for $0 < \alpha < 1/2$, we can change variables and write $s^+ \to s^+ - 2i e^{i(\bar{p}_j - \phi_j^+)}\delta p_j \sqrt{t}$. Since $\delta p\sqrt{t} \sim t^{\alpha - 1/2}$ goes to zero, we have

$$\mathcal{I}_1^{\rho'}[f(w_1, w_2)](x, z, t) \sim \sum_{j=1}^4 e^{tS_{(x+\frac{z}{2})/t}(e^{i\bar{p}_j + i\delta p_j}) - tS_{(x-\frac{z}{2})/t}(e^{i\bar{p}_j - i\delta p_j})} e^{i\phi_j^+} e^{i\phi_j^-} f(e^{i\bar{p}_j}, e^{i\bar{p}_j}) \times$$

$$\times \int_{-\Delta\sqrt{t}}^{\Delta\sqrt{t}} \frac{\mathrm{d}s^+ \mathrm{d}s^-}{(2\pi)^2} \frac{e^{-\frac{1}{2}|S_{x/t}''(e^{i\bar{p}_j + i\delta p_j})|(s^+)^2 - \frac{1}{2}|S_{x/t}''(e^{i\bar{p}_j - i\delta p_j})|(s^-)^2}}{\sqrt{t}s^+ e^{i\phi_j^+} - \sqrt{t}s^- e^{i\phi_j^-}}. \quad (82)$$

Note that there is no singularity in this integral because $e^{i(\phi_j^+ - \phi_j^-)} = \pm i$, and thus $\mathcal{I}_1^{\rho'}$ decays as $t^{-1/2}$.

For $1/2 < \alpha < 1$, instead, we can take $-1/2 < \omega < \min\{\alpha - 1, -1/3\}$, so that the leading order is

$$\mathcal{I}_1^{\rho'}[f(w_1, w_2)](x, z, t) \sim \sum_{j=1}^4 \int_{-\Delta\sqrt{t}}^{\Delta\sqrt{t}} \frac{\mathrm{d}s^+ \mathrm{d}s^-}{(2\pi)^2} e^{i\phi_j^+} e^{i\phi_j^-} f(e^{i\bar{p}_j}, e^{i\bar{p}_j})$$

$$\frac{e^{tS_{(x+\frac{z}{2})/t}(e^{i\bar{p}_j + i\delta p_j}) - tS_{(x-\frac{z}{2})/t}(e^{i\bar{p}_j - i\delta p_j}) - \frac{1}{2}|S''_{x/t}(e^{i\bar{p}_j + i\delta p_j})|(s^+)^2 - \frac{1}{2}|S''_{x/t}(e^{i\bar{p}_j - i\delta p_j})|(s^-)^2}}{2i e^{i\bar{p}_j} \delta p_j t}$$

$$\sim \sum_{j=1}^4 \frac{e^{tS_{(x+\frac{z}{2})/t}(e^{i\bar{p}_j + i\delta p_j}) - tS_{(x-\frac{z}{2})/t}(e^{i\bar{p}_j - i\delta p_j})}}{4\pi i t |S''_{x/t}(e^{i\bar{p}_j})| e^{i\bar{p}_j} \delta p_j} e^{i\phi_j^{(1)}} e^{i\phi_j^{(2)}} f(e^{i\bar{p}_j}, e^{i\bar{p}_j}), \quad (83)$$

that goes to zero as $\frac{1}{t\delta p_j} \sim t^{-\alpha}$. This expression can be further simplified noting that in the stationary points $|S''_\zeta(e^{i\bar{p}_j})| = 4|\epsilon''(2\bar{p}_j)|$.

The computation for the second contribution $\mathcal{I}_2[f(w_1, w_2)](x, z, t)$ goes along the same lines of the previous one and it is discussed in Appendix A. The two contributions are eventually combined as in Eq. (67) to get the final asymptotics.

### 6.2.3 Standard GHD

Although in this work we are mainly interested in the case $\alpha > 0$, let us briefly discuss $\alpha = 0$, when the observable has a support that does not scale with time and it is thus strictly local, to show how standard GHD emerges from our formalism. All the steps explained above still hold, but we shall drop the $z$-dependence from the function multiplying the large parameter in the saddle-point-method representation of the integral, since $z$ is not anymore a divergent parameter.

For what concerns the double-integral contributions (after the deformation of the contours), the conclusions do not change: it still behaves at most as $t^{-1}$. As for the residue contribution, the important difference is that it loses any explicit dependence on large parameters (there is still an implicit $x/t$ dependence in the definition of the saddle-points), so it does not go to zero anymore. We can parametrize $\gamma$ such that it consists in two arcs of circumference that go from the first to the second saddle point and from the third to the fourth one. In particular, here $\gamma$ is composed by the two pieces of the unit-circle such that if we move infinitesimally outside the unit-circle the real part of $S_{x/t}(z)$ is positive: expanding $S_{x/t}(e^{i(p+i\delta)})$ for real $p$ and small real $\delta$ we find that $\gamma$ contains the points such that $s'_{x/t}(p) > 0$, i.e. $x/t > \epsilon'(2p)$. Then we have

$$(\Gamma_x^{z;\rho}(t))_{11} \sim (\Gamma_x^{z;\rho}(t))_{22} \sim 2i \int_\pi^\pi \mathrm{d}p \left(\rho^R(2p) - \frac{1}{4\pi}\right) \sin(2zp)\Theta\left(\frac{x}{t} > \epsilon'(2p)\right)$$

$$= 4\pi i \int_\pi^\pi \frac{\mathrm{d}p}{2\pi} \left(\rho^R(p) - \frac{1}{4\pi}\right) \sin(zp)\Theta\left(\frac{x}{t} - \epsilon'(p)\right), \quad (84)$$

$$(\Gamma_x^{z;\rho}(t))_{21} = -\left(\Gamma_x^{-z;\rho}(t)\right)_{12} \sim 2\int_\gamma \frac{\mathrm{d}w}{w}\left(\rho^R(-2\mathrm{i}\ln(w)) - \frac{1}{4\pi}\right)\cos\left(-2z\mathrm{i}\ln w + \theta(-2\mathrm{i}\ln w)\right)$$

$$= 4\pi\mathrm{i}\int_{-\pi}^{\pi}\frac{\mathrm{d}p}{2\pi}\left(\rho^R(p) - \frac{1}{4\pi}\right)\cos\left(zp + \theta(p)\right)\Theta\left(\frac{x}{t} - \epsilon'(p)\right), \quad (85)$$

consistently with the GHD prediction. This has a physical interpretation in terms of quasi-particles that propagate with velocity $\epsilon'(p)$: only the excitations with group velocity $\epsilon'(p)$ manage to arrive at time $t$ at the position $x$, while the other give no contribution to the leading order.

## 6.3 Result of the asymptotic analysis

Summarizing, we considered the partition protocol defined in (49) and we gave an integral representation in (63) of the correlation-matrix elements $\Gamma_{2m+i,2n+j} \equiv (\Gamma_{\frac{m-n}{2}}^{m-n})_{ij}$, that we denoted $\Gamma^\rho$ in this partitioning protocol. We computed the leading order of those integrals in the scaling limit $t \to +\infty$, with $x = \zeta t$ and $z = ct^\alpha > 0$, for $\alpha \in [0,1]$. Let us go through all the different regimes.

If at least one ray is outside the lightcone, meaning that $|\zeta + \frac{z}{t}| > v_{max}$ or $|\zeta - \frac{z}{t}| > v_{max}$, the integrals (63) are exponentially suppressed in time for any value of $\alpha$. If both rays are inside the lightcone we have different regimes, depending on $\alpha$.

For $\alpha = 1$, the leading order can be reduced to a standard application of the saddle-point method. We do not report the explicit result because it is not particularly insightful, but it can be shown that it decays as $t^{-1}$.

For $0 < \alpha < 1/2$, we obtain

$$(\Gamma_x^{z;\rho})_{11} \sim (\Gamma_x^{z;\rho})_{22} \sim -\mathrm{i}\sum_{j=1}^2 (-1)^j \frac{2}{z}\cos(2z\bar{p}_j)\left(\rho^R(2\bar{p}_j) - \frac{1}{4\pi}\right),$$

$$(\Gamma_x^{z;\rho})_{21} = \left(\Gamma_x^{-z;\rho}\right)_{12} \sim -\mathrm{i}\sum_{j=1}^2 (-1)^j \frac{2}{z}\sin(2z\bar{p}_j + \theta(2\bar{p}_j))\left(\rho^R(2\bar{p}_j) - \frac{1}{4\pi}\right), \quad (86)$$

where $\epsilon(p)$ and $\theta(p)$ are the dispersion relation (9) and the Bogoliubov angle (10), $\rho^R(p)$ is defined by the partitioning protocol (49), and $\bar{p}_1$ and $\bar{p}_2$ are the stationary points of the function $s_{x/t}(p) = 2\frac{x}{t}p - \epsilon(2p)$ that go respectively to $\pi/2$ and $\pi$ when $x/t$ is continuously sent to zero. Therefore the elements of the correlation matrix scale as $\sim z^{-1} \sim t^{-\alpha}$. Note that there is an implicit dependence on $x$, since $\bar{p}_j$ depends on $x/t$.

For $1/2 < \alpha < 1$, we have

$$(\Gamma_x^{z;\rho})_{11} \sim (\Gamma_x^{z;\rho})_{22} \sim$$

$$\sim -\mathrm{i}\sum_{j=1}^2\left(\rho^R(2\bar{p}_j) - \frac{1}{4\pi}\right)\frac{\cos\left(ts_{(x+\frac{z}{2})/t}(\bar{p}_j + \delta p_j) - ts_{(x-\frac{z}{2})/t}(\bar{p}_j - \delta p_j)\right)}{2t|\epsilon''_{x/t}(\mathrm{e}^{\mathrm{i}\bar{p}_j})|\delta p_j}, \quad (87)$$

$$\left(\Gamma_x^{z;\rho}\right)_{21} = -\left(\Gamma_x^{-z;\rho}\right)_{12} \sim$$

$$\sim i \sum_{j=1}^{2} \left(\rho^R(2\bar{p}_j) - \frac{1}{4\pi}\right) \frac{\sin\left(ts_{(x+\frac{z}{2})/t}(\bar{p}_j + \delta p_j) - ts_{(x-\frac{z}{2})/t}(\bar{p}_j - \delta p_j) + \theta(2\bar{p}_j)\right)}{2t|\epsilon''_{x/t}(e^{i\bar{p}_j})|\delta p_j} , \quad (88)$$

where $\bar{p}_j + \delta p_j$ are the saddle points of $s_{(x+\frac{z}{2})/t}(p)$, with $\bar{p}_1$ and $\bar{p}_2$ defined as above. Importantly, $\delta p_j \sim t^{\alpha-1}$, so that the elements of the correlation matrix scale as $(t\delta p_j)^{-1} \sim t^{-\alpha}$.

For $\alpha = 0$, we showed how to recover the standard GHD result. In that case the contribution of the root density to the correlation matrix's elements is finite. As we will see, this is the only regime in which one field can give a non-zero contribution in the infinite-time limit.

We can summarize everything saying that the contribution to the correlation matrix accounted for by the root density behaves as $\Gamma^\rho \sim t^{-\alpha}$, for any $\alpha \in [0, 1]$. From this result and using Eq. (48) we get directly a prediction for the spin-spin connected correlation $S_{m,n}^z(t)$, with $x \equiv \frac{m+n}{2} \equiv \zeta t$ and $z \equiv m - n \equiv ct^\alpha$, for $\alpha \in [0, 1]$, obtaining an explicit result that decays as $t^{-2\alpha}$. A comparison with numerical simulation is reported in Fig. 3 (see Appendix D for more details about the numerical simulation). We point out that the spin-spin correlation in the case $\alpha = 0$ was already discussed in Ref. [44].

# 7 Auxiliary-field partitioning protocol

## 7.1 Integral representation of the correlation matrix

In this section we perform the asymptotic analysis of the spin-spin connected correlation function (47) in the case of the partitioning protocol (51) defined in terms of the auxiliary field only. To do so, we first focus on the asymptotics of the correlation matrix in this partitioning protocol. Following exactly the same steps used for the root density, we rewrite the contribution of the auxiliary field to the correlation matrix as

$$\left(\Gamma_x^{z;\psi}\right)_{11} = -\left(\Gamma_x^{z;\psi}\right)_{22} = i\,\mathrm{Re}\sum_{y\in\mathbb{Z}/2}\int_{-\pi}^{\pi}\frac{\mathrm{d}p\mathrm{d}q}{(2\pi)^2}\times$$

$$\times \sin\left(pz\right)\cos\left(2q(x-y) + \frac{\theta(p-q) - \theta(p+q)}{2}\right)e^{-it(\epsilon(p+q)+\epsilon(p-q))}\psi_y(p,0), \quad (89)$$

$$\left(\Gamma_x^{z;\psi}\right)_{21} = -\left(\Gamma_x^{-z;\psi}\right)_{12} = -i\,\mathrm{Im}\sum_{y\in\mathbb{Z}/2}\int_{-\pi}^{\pi}\frac{\mathrm{d}p\mathrm{d}q}{(2\pi)^2}\times$$

$$\times \sin\left(pz + \frac{\theta(p-q) + \theta(p+q)}{2}\right)\cos\left(2q(x-y)\right)e^{-it(\epsilon(p+q)+\epsilon(p-q))}\psi_y(p,0). \quad (90)$$

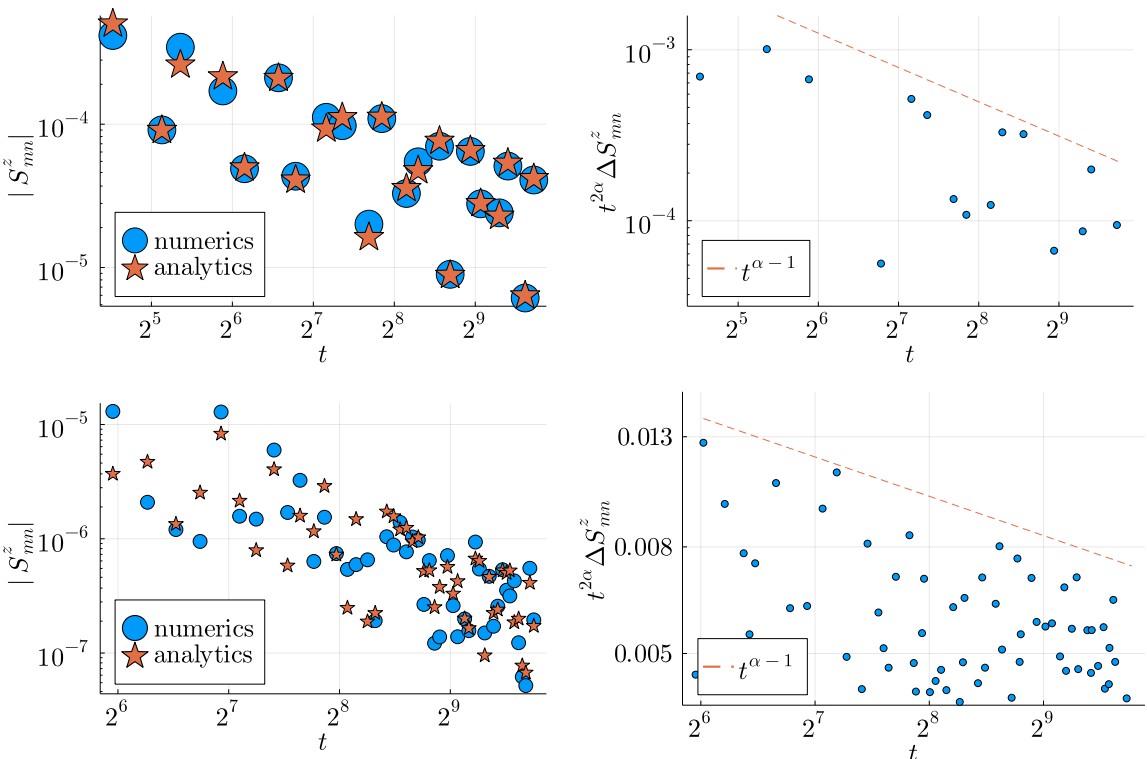

Figure 3: Spin-spin connected correlation matrix for the partitioning protocol between two thermal states, involving only the root density. The inverse temperatures are zero on the left and $\beta_R = 0.9$ on the right. Here $J = 1/2$ and $h = 2$. On the left we report both the numerical and the analytical prediction. On the right we report the absolute value of the difference between the two $\Delta S_{mn}^z(t) = |analytical\ S_{mn}^z(t) - numerical\ S_{mn}^z(t)|$, rescaled by the behaviour of the leading order. A reference curve is plotted to help the reader identify how fast it goes to zero. The positions $(m, n)$ that we use are $(\zeta t + ct^\alpha, \zeta t - ct^\alpha)$, rounded to the closest integer. Top: $\zeta = 1/3$, $c = 3$, $\alpha = 1/3$. Bottom: $\zeta = -1/3$, $c = 2$, $\alpha = 3/4$.

Using that for any odd function $f(q)$

$$\sum_{y \in \mathbb{Z}/2} \int_{-\pi}^{\pi} \frac{\mathrm{d}q}{2\pi} \cos(2q(x - y) + f(q))\psi_y(p, 0)$$

$$= \sum_{y \in \mathbb{Z}/2} \int_{-\pi}^{+\pi} \frac{\mathrm{d}\tau \mathrm{d}q}{(2\pi)^2} \frac{\mathrm{e}^{-2\mathrm{i}qy}\mathrm{e}^{2\mathrm{i}qx+\mathrm{i}f(q)} + \mathrm{e}^{2\mathrm{i}qy}\mathrm{e}^{-2\mathrm{i}qx-\mathrm{i}f(q)}}{2}\psi^R(p)\frac{\mathrm{e}^{2\mathrm{i}y\tau}}{1 - \mathrm{e}^{-\mathrm{i}(\tau-\mathrm{i}0)}}$$

$$= \int_{-\pi}^{+\pi} \frac{\mathrm{d}\tau}{2\pi} \frac{\mathrm{e}^{2\mathrm{i}\tau x+\mathrm{i}f(\tau)}}{1 - \mathrm{e}^{-\mathrm{i}(\tau-\mathrm{i}0)}}\psi^R(p), \quad (91)$$

we can recast the expression above in the form

$$\left(\Gamma_x^{z;\psi}\right)_{11} = -\left(\Gamma_x^{z;\psi}\right)_{22} = \mathrm{i}\,\mathrm{Re}\int_{-\pi}^{\pi} \frac{\mathrm{d}p\mathrm{d}q}{(2\pi)^2} \times$$

$$\times \sin(pz)\frac{\mathrm{e}^{2\mathrm{i}qx}}{1 - \mathrm{e}^{-\mathrm{i}(q-\mathrm{i}0)}}\mathrm{e}^{\mathrm{i}\frac{\theta(p-q)-\theta(p+q)}{2}}\mathrm{e}^{-\mathrm{i}t(\epsilon(p+q)+\epsilon(p-q))}\psi^R(p), \quad (92)$$

$$\left(\Gamma_x^{z;\psi}\right)_{21} = -\left(\Gamma_x^{-z;\psi}\right)_{12} = -i\,\mathrm{Im}\int_{-\pi}^{\pi}\frac{\mathrm{d}p\mathrm{d}q}{(2\pi)^2}\times$$

$$\times\sin\left(pz + \frac{\theta(p-q)+\theta(p+q)}{2}\right)\frac{e^{2iqx}}{1-e^{-i(q-i0)}}e^{-it(\epsilon(p+q)+\epsilon(p-q))}\psi^R(p). \quad (93)$$

Using the property $\int_{-\pi}^{\pi}\mathrm{d}p\mathrm{d}q\;f(p,q) = \int_{-\pi}^{\pi}\mathrm{d}p\mathrm{d}q\;f(p+q,p-q)$ of periodic functions, we can rewrite it as

$$\left(\Gamma_x^{z;\psi}\right)_{11} = -\left(\Gamma_x^{z;\psi}\right)_{22} = i\,\mathrm{Re}\int_{-\pi}^{\pi}\frac{\mathrm{d}p\mathrm{d}q}{(2\pi)^2}\times$$

$$\times\sin\left((p+q)z\right)\frac{e^{2i(p-q)x}}{1-e^{-i(p-q-i0)}}e^{i\frac{\theta(2q)-\theta(2p)}{2}}e^{-it(\epsilon(2p)+\epsilon(2q))}\psi^R(p+q), \quad (94)$$

$$\left(\Gamma_x^{z;\psi}\right)_{21} = -\left(\Gamma_x^{-z;\psi}\right)_{12} = -i\,\mathrm{Im}\int_{-\pi}^{\pi}\frac{\mathrm{d}p\mathrm{d}q}{(2\pi)^2}\times$$

$$\times\sin\left((p+q)z + \frac{\theta(2q)+\theta(2p)}{2}\right)\frac{e^{2i(p-q)x}}{1-e^{-i(p-q-i0)}}e^{-it(\epsilon(2p)+\epsilon(2q))}\psi^R(p+q). \quad (95)$$

At this point we change variables as $w_1 = e^{ip}, w_2 = e^{i(q+i0)}$ and we switch to the complex plane:

$$\left(\Gamma_x^{z;\psi}\right)_{11} = -\left(\Gamma_x^{z;\psi}\right)_{22} = \frac{i}{2}\,\mathrm{Re}\,i\int_{\mathcal{C}_1}\frac{\mathrm{d}w_1}{2\pi}\int_{\mathcal{C}_2}\frac{\mathrm{d}w_2}{2\pi w_2}\psi^R(-i\ln(w_1 w_2))\times$$

$$\times e^{i\frac{\theta(-2i\ln w_2)-\theta(-2i\ln w_2)}{2}}\frac{e^{tS_{(x+\frac{z}{2})/t}(w_1)+tS_{(-x+\frac{z}{2})/t}(w_2)} - e^{tS_{(x-\frac{z}{2})/t}(w_1)+tS_{(-x-\frac{z}{2})/t}(w_2)}}{w_1 - w_2}$$

$$\left(\Gamma_x^{z;\psi}\right)_{21} = -\left(\Gamma_x^{-z;\psi}\right)_{12} = -\frac{i}{2}\,\mathrm{Im}\,i\int_{\mathcal{C}_1}\frac{\mathrm{d}w_1}{2\pi}\int_{\mathcal{C}_2}\frac{\mathrm{d}w_2}{2\pi w_2}\frac{\psi^R(-i\ln(w_1 w_2))}{w_1 - w_2}\times \quad (96)$$

$$\times\left(e^{tS_{(x+\frac{z}{2})/t}(w_1)+tS_{(-x+\frac{z}{2})/t}(w_2)}e^{i\frac{\theta(-2i\ln w_1)+\theta(-2i\ln w_2)}{2}} + \right.$$

$$\left. - e^{tS_{(x-\frac{z}{2})/t}(w_1)+tS_{(-x-\frac{z}{2})/t}(w_2)}e^{-i\frac{\theta(-2i\ln w_1)+\theta(-2i\ln w_2)}{2}}\right),$$

where the curves $\mathcal{C}_1$ and $\mathcal{C}_2$ and the function $S_\zeta(w)$ are defined as in Eq. (63). The considerations made in the previous section about the singularities of the function under integration still hold.

## 7.2   Result of the asymptotic analysis

Eq. (96) provides an integral representation of the correlation-matrix element $\Gamma_{2m+i,2n+j} \equiv (\Gamma_{\frac{m+n}{2}}^{m-n})_{ij}$ in the partitioning protocol (51), that we denoted $\Gamma^\psi$. We are interested in the scaling limit $t \to +\infty$, with $x \equiv \frac{m+n}{2} = \zeta t$ and $z \equiv m - n = ct^\alpha > 0$, for $\alpha \in [0,1]$. Such an

asymptotic study can be done using the same arguments that we used in the root-density case. The application of those argument to the case under consideration is reported in Appendix B. Here we just report the result, going through all the different regimes.

Consistently with the Lieb-Robinson bound, $\Gamma^{m-n;\psi}_{\frac{m+n}{2}}$ is exponentially suppressed if the distance $m - n$ is larger than $2v_{max}t$, i.e. when the matrix element refers to components of the system that are causally disconnected. Same thing when at least one of the rays $m/t$ and $n/t$ is to the left of the lightcone, where the state is locally indistinguishable from an infinite-temperature thermal state (up to exponentially small corrections). Therefore, in the following we assume $m - n < 2v_{max}t$ and both $m/t$ and $n/t$ larger than $-v_{max}$.

For $-v_{max} < \zeta < 0$ or for $-v_{max} < \zeta < v_{max} \wedge \alpha < 1/2$, the leading order scales as $t^{-1}$ and it is given by a standard application of the saddle-point method to a double integral. Essentially, one can deform the integration contours of the integrals (96) neglecting the pole $w_1 - w_2$. Since the application of the saddle-point method is standard, we do not report here the explicit result.

For $0 < \zeta < v_{max} \wedge \alpha > 1/2$ or for $\zeta > v_{max}$ we have

$$
\begin{aligned}
\left(\Gamma^{z;\psi}_x\right)_{11} &= -\left(\Gamma^{z;\psi}_x\right)_{22} \sim -\frac{i}{2\sqrt{\pi t}} \operatorname{Re} \sum_{j=1}^{2} e^{i\frac{\pi}{4}(5-2j)} \frac{e^{2its\frac{z}{2t}(\bar{p}_j)} \psi^R(2\bar{p}_j)}{\sqrt{|\epsilon''(2\bar{p}_j)|}}, \\
\left(\Gamma^{z;\psi}_x\right)_{21} &= -\left(\Gamma^{-z;\psi}_x\right)_{12} \sim \frac{i}{2\sqrt{\pi t}} \operatorname{Im} \sum_{j=1}^{2} e^{i\frac{\pi}{4}(5-2j)} \frac{e^{2its\frac{z}{2t}(\bar{p}_j)} e^{i\theta(2\bar{p}_j)} \psi^R(2\bar{p}_j)}{\sqrt{|\epsilon''(2\bar{p}_j)|}},
\end{aligned}
\tag{97}
$$

where $\psi^R(p)$ enters the definition of the partitioning protocol (96), $\epsilon(p)$ and $\theta(p)$ are the dispersion relation (9) and the Bogoliubov angle (10), $s_\zeta(p) = 2\zeta p - \epsilon(2p)$, and $\bar{p}_1$ and $\bar{p}_2$ are the stationary points of $s_{\frac{z}{2t}}(p)$ that continuously go to $\pi/2$ and $\pi$ when $z \to 0$. Note that these matrix elements are not exponentially suppressed even outside (to the right) of the lightcone, consistently with the fact that a homogeneous state for which at least one field different from $\rho_{x,t}(p)$ is different from zero undergoes a global quench. Importantly, because $\psi^R(p)$ is periodic, smooth and odd, it is always zero when evaluated in $0$ and $\pi$. This leads to $\psi^R(2\bar{p}_j) \sim t^{\alpha-1}$, which gives an overall scaling $t^{\alpha-3/2}$. Note that for generic dispersion relations, the saddle points of $s_{\frac{z}{2t}}(p)$ do not collapse to $\pi/2$ and $\pi$, which means that $\psi^R(2\bar{p}_j) \sim const$ in the generic case and the overall scaling is simply $t^{-1/2}$ instead of $t^{\alpha-3/2}$ of the Ising model.

We can plug these results in Eq. (48) to get the prediction for the spin-spin connected correlation, which is of order $t^{-2}$ for $-v_{max} < \zeta < 0$, $t^{-\min(3-2\alpha, 2)}$ for $0 < \zeta < v_{max}$ and $t^{-3+2\alpha}$ for $\zeta > v_{max}$. See Fig. 4 for a comparison with a numerical simulation. Note that the lightcone can be inferred by looking at the qualitative behavior of "on-ray" observables, despite the global quench on the right: the connected correlation between nearest neighbors decays as $t^{-2}$ inside the lightcone and as $t^{-3}$ to its right. As a byproduct, we have recovered the special case considered in Ref. [71], where it is shown that the relaxation to a stationary state in a global quench from a generic Gaussian state in the TFIC is attained with corrections $t^{-3/2}$ to the local elements of the correlation matrix.

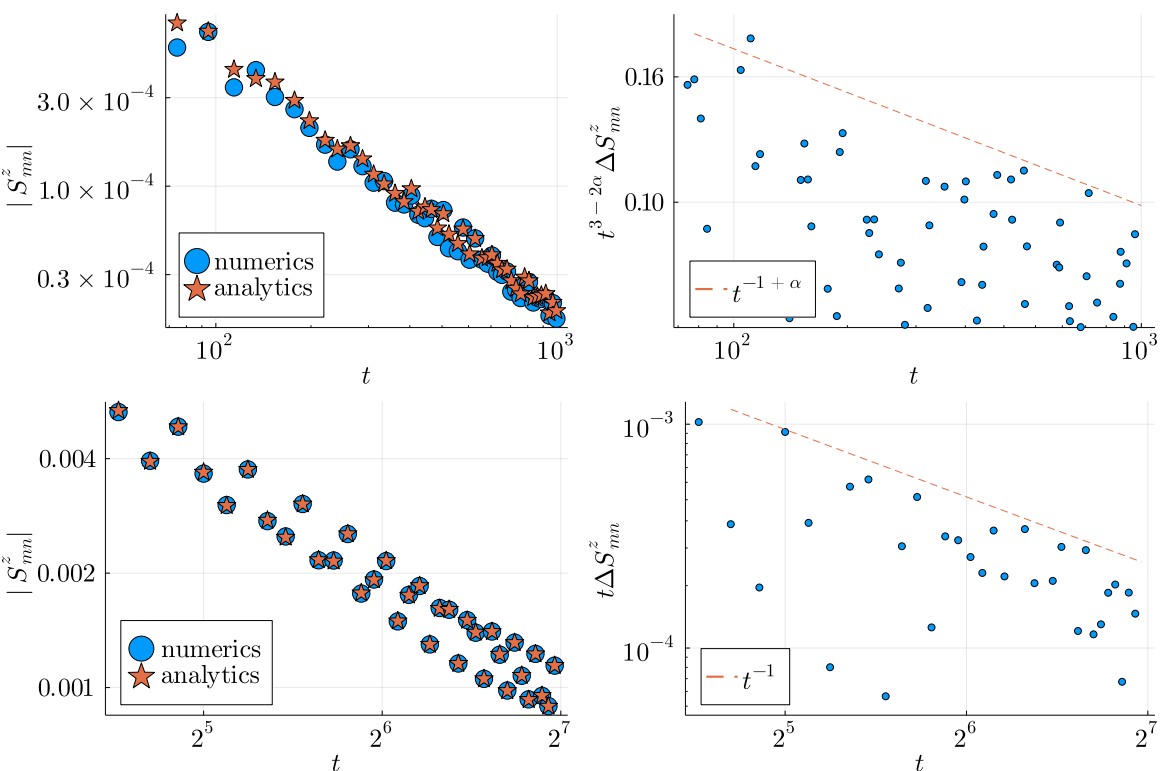

Figure 4: Spin-spin connected correlation matrix for the partitioning protocol defined in the main text in terms of the auxiliary field only. Here $J = 1/2$ and $h = 2$. On the left we report both the numerical and the analytical prediction. On the right report the absolute value of the difference between the two, rescaled by the behaviour of the leading order. A reference curve is plotted to help the reader identify how fast it goes to zero. The positions $(m, n)$ that we used are $(\zeta t + \frac{ct^\alpha}{2}, \zeta t - \frac{ct^\alpha}{2})$, rounded to the closest integer. Top: $\zeta = 1/3$, $c = 2$, $\alpha = 3/4$. Bottom: $\zeta = 7/3$, $c = 1$, $\alpha = 1$.

# 8 Non-Gaussian partitioning protocol

In this section we perform the asymptotic analysis of the spin-spin connected correlation function (47) in the case of the partitioning protocol (53) defined in terms of the non-Gaussian fields only.

The contribution from each field can be computed as illustrated in Appendix $C$. In short, the simplicity of the initial state can be exploited to compute all the connected 4-point functions in Bogoliubov fermions, which are then plugged in the definitions of the fields, and, finally, they are connected to the observable via the parametrization (39). For example, the contribution of the field $\xi_{x_1, x_2, t}(p_1, p_2)$ reads

$$S_{mn}^{z,\xi}(t) = \int_{-\pi}^{\pi} \frac{d^2p\, d^2q\, dk}{(2\pi)^5} \frac{e^{2iq_1(m-n)} e^{imk} e^{2inq_2} e^{-it(\epsilon(p_1+q_1+k)-\epsilon(p_1-q_1)+\epsilon(p_2-q_1+2q_2)-\epsilon(p_2+q_1))}}{(1 - e^{-i(k-i0)})(1 - e^{-i(2q_2-i0)})} \times$$
$$\times e^{-ik} e^{-2iq_2} \cos\left(\frac{\theta(p_1-q_1)+\theta(p_1+q_1+k)}{2}\right) \cos\left(\frac{\theta(p_2+q_1)+\theta(p_2-q_1+2q_2)}{2}\right) G_\beta(p_1, p_2, q_1, 2q_2, k), \quad (98)$$

where

$$G_\beta(p_1, p_2, q_1, q_2, k) := T_\beta(k + p_1 + p_2) \sin\left(\tfrac{\theta(p_1-q_1)-\theta(p_2+q_1)}{2}\right) \sin\left(\tfrac{\theta(p_1+q_1+k)-\theta(p_2-q_1+q_2)}{2}\right) +$$
$$+ T_\beta(2q_1) \cos\left(\tfrac{\theta(p_1-q_1)+\theta(p_1+q_1+k)}{2}\right) \cos\left(\tfrac{\theta(p_2+q_1)+\theta(p_2-q_1+q_2)}{2}\right) +$$
$$- T_\beta(k + p_1 - p_2 - q_2) \cos\left(\tfrac{\theta(p_1-q_1)+\theta(p_2-q_1+q_2)}{2}\right) \cos\left(\tfrac{\theta(p_1+q_1+k)+\theta(p_2+q_1)}{2}\right), \quad (99)$$

and

$$T_\beta(k) := \frac{1}{\cosh^2(\beta)} \frac{1}{\tanh^2(\beta) - 2\tanh(\beta)\cos(k) + 1}. \quad (100)$$

Note that $G_\beta(p_1, p_2, q_1, q_2, k)$ is periodic in all its entries with period $2\pi$. Similar computations can be carried out also for the other non-Gaussian fields $\Upsilon$ and $\Omega$. We will not report them here, since, as discussed below, the leading order from the non-Gaussian part comes from the field $\xi_{x_1,x_2,t}(p_1, p_2)$.

## 8.1 Asymptotic analysis

Here we compute the asymptotics of the integrals (98) in the scaling limit $t \to +\infty$, with $x = \zeta t, z = ct^\alpha, \alpha \in [0,1], \zeta \neq 0, c > 0$. The leading contribution comes from the neighborhood of those points in which the function under integration singular and its divergent phase is stationary. We start by expanding (98) around the singular points, then we study the phase that multiplies the large parameters. There are two of those points: $(q_2, k) = (0,0)$ and $(q_2, k) = (\pi, 0)$. Since the function that we are integrating is periodic in $q_2$ with period $\pi$, the two contributions give the same result, so that

$$S_{mn}^{z,\xi}(t) \sim \int_{-\pi}^{\pi} \frac{\mathrm{d}^2p\,\mathrm{d}^2q\,\mathrm{d}k}{(2\pi)^5} \mathrm{e}^{2iq_1(m-n)} \mathrm{e}^{-it(\epsilon(p_1+q_1)-\epsilon(p_1-q_1)+\epsilon(p_2-q_1)-\epsilon(p_2+q_1))} G_\beta(p_1, p_2, q_1, 0, 0)$$
$$\cos\left(\tfrac{\theta(p_1-q_1)+\theta(p_1+q_1)}{2}\right) \cos\left(\tfrac{\theta(p_2+q_1)+\theta(p_2-q_1)}{2}\right) \frac{\mathrm{e}^{imk} \mathrm{e}^{2inq_2} \mathrm{e}^{-it(\epsilon'(p_1+q_1)k+\epsilon'(p_2-q_1)2q_2)}}{i(k-i0)i(q_2-i0)}$$
$$\sim \int_{-\pi}^{\pi} \frac{\mathrm{d}^2p\,\mathrm{d}q}{(2\pi)^3} \mathrm{e}^{2iq(m-n)} \mathrm{e}^{-it(\epsilon(p_1+q)-\epsilon(p_1-q)+\epsilon(p_2-q)-\epsilon(p_2+q))} G_\beta(p_1, p_2, q, 0, 0)$$
$$\cos\left(\tfrac{\theta(p_1-q)+\theta(p_1+q)}{2}\right) \cos\left(\tfrac{\theta(p_2+q)+\theta(p_2-q)}{2}\right) \Theta\left(\tfrac{m}{t} - \epsilon'(p_1+q)\right) \Theta\left(\tfrac{n}{t} - \epsilon'(p_2-q)\right). \quad (101)$$

Similar expressions represent the leading contribution from the other fields and can be computed as shown in Appendix C. Note that we can see how the lightcone structure of the partitioning protocol emerges from this integral: if one of the two rays $m/t$ or $n/t$ is smaller than $-v_{max}$, the integral is zero because of the step functions, and we would have to expand to the next order; once both rays are larger than $v_{max}$, the only remaining dependence on position is $m - n$, so we get the homogeneous result (recall that the state is not stationary outside the lightcone, since at least one field beside the root density is different from zero in the initial state).

Let us consider first the scaling limit in which $m - n$ is constant in time, i.e. $\alpha = 0$. To extract the leading order from the expression above, we need to perform an analysis of the phase that multiplies time. In particular, we look for its stationary points. The

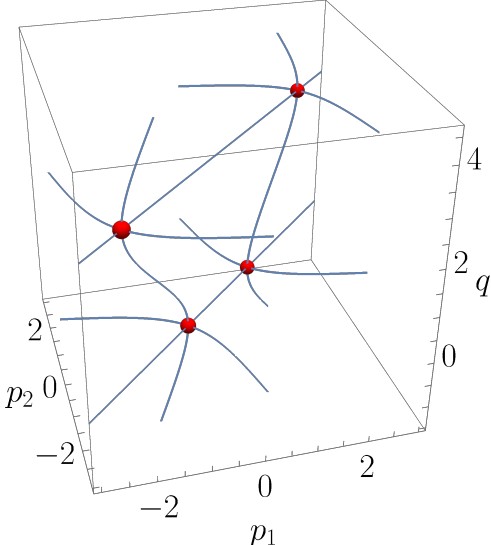

Figure 5: Stationary points of the function $\epsilon(p_1 + q) - \epsilon(p_1 - q) + \epsilon(p_2 - q) - \epsilon(p_2 + q)$ in the domain $p_1, p_2 \in [-\pi, \pi], q \in [-\frac{1}{2}\pi, \frac{3}{2}\pi]$. The blue lines are curves of stationary points, the red points are special stationary points in which also the Hessian of the phase is zero.

stationary-points for the integral (101) are actually stationary curves, represented in Fig. 5 and parametrized in the $(p_1, p_2, q)$ coordinates as

$$
\begin{aligned}
(s, s, 0), & \quad s \in [-\pi, \pi), \\
(s, s, \pi), & \quad s \in [-\pi, \pi), \\
(s, -\eta(s), 0), & \quad s \in [-\pi, 0), \\
(s, \eta(s), 0), & \quad s \in [0, \pi), \\
(s, -\eta(s), \pi), & \quad s \in [-\pi, 0), \\
(s, \eta(s), \pi), & \quad s \in [0, \pi), \\
\tfrac{1}{2}(s + \eta(s), s + \eta(s), s - \eta(s)), & \quad s \in [0, \pi), \\
\tfrac{1}{2}(s + \eta(s) - 2\pi, s + \eta(s) - 2\pi, s - \eta(s) + 2\pi), & \quad s \in [0, \pi), \\
\tfrac{1}{2}(s - \eta(s), s - \eta(s), s + \eta(s)), & \quad s \in [-\pi, 0) \cup [\pi, 2\pi),
\end{aligned}
\tag{102}
$$

where

$$
\eta(s) := \arccos\left( \frac{2h - (1 + h^2)\cos s}{1 + h^2 - 2h\cos s} \right). \tag{103}
$$

Instead, it can be shown that the dynamical fields $\Upsilon$ and $\Omega$ only have isolated stationary points. So we can already conclude that the leading order to our observable from the non-Gaussian part comes from the field $\xi$. Such a property applies in general, and not only for our specific example, since it only relies on the presence of the singularities (which are a general characteristic of partitioning protocols) and the phase (which comes directly from the equation of motion).

Note that the integral above is only part of the contribution of the field $\xi$ and in general one would need to compute also the part related to the field $\omega$ (that can be written in function of $\xi$). That contribution has the same structure of the one already discussed: it is asymptotically equivalent to a triple integral with a phase that has curves of stationary points. However, for our specific observable and initial state, the function under integration is zero along those curves, so that such a contribution is sub-leading, although in general one would have to consider it (it does not change the qualitative behavior of the result, since, as we said, it has the same structure of the term that we are discussing).

To apply the stationary-phase approximation in presence of stationary curves, one wants to expand around those curves. To do so, we divide the domain in various pieces that contain a curve each (note that there are a few intersection points, but it can be shown that counting them twice introduces sub-leading errors, since the function under integration is zero in those points), then we make the change of variables $(p_1, p_2, q) \to (s, a, b)$, where $s$ is the parameter that parametrizes the stationary curves, such that, in the new variables, they are written as $(s, 0, 0)$. For example, for the third and seventh curves we can take

$$
\begin{cases} p_1 = s + (a+b)\eta'(s) \\ p_2 = -\eta(s) + a + b \\ q = a - b \end{cases} \quad \text{and} \quad \begin{cases} p_1 = \frac{1}{2}(s + 2a - (a+b)\eta'(s) + \eta(s)) \\ p_2 = \frac{1}{2}(s + 2b - (a+b)\eta'(s) + \eta(s)) \\ q = \frac{1}{2}(s - a - b - (a+b)\eta'(s) - \eta(s)) \end{cases} \tag{104}
$$

respectively. We can finally expand for small $a$ and $b$, obtaining two Gaussian integrals that can be computed and give a factor $\sim t^{-1/2}$ each. The remaining integral over $s$ can be evaluated numerically and, since it does not contain any large parameter anymore, it simply gives a constant. Note that, whenever at least one ray is inside the lightcone, only parts of the stationary curves give a contribution, since some of the stationary points are excluded because of the step functions. We stress out that the only thing that changes is the domain over which we integrate $s$, hence the overall constant, but the behavior is the same.

Let us turn to study what happens when we let the distance between the spins to scale with time: $m - n \sim t^\alpha$. For $\alpha = 1$ the stationary curves get modified, but, importantly, they still exist, so that the qualitative behavior of the integral does not change. They can be computed and one can obtain the leading term with the same method used for $\alpha = 0$.

Since, unlike in the Gaussian case, there are not any emergent/disappearing properties when we go from $\alpha = 0$ to $\alpha = 1$, we conclude that the case $\alpha \in (0, 1)$ shows no qualitative difference from the limiting cases $\alpha = 0$ and $\alpha = 1$. In the end, the leading order of the non-Gaussian part of the state goes to zero as $t^{-1}$ for any $\alpha \in [0, 1]$.

## 8.2 Result of the asymptotic analysis

Summarizing, we computed the leading order of spin-spin connected correlation $S_{m,n}^z(t)$ in the scaling limit $t \to +\infty$, with $\frac{m+n}{2} \sim t$ and $m - n \sim t^\alpha$, for $\alpha \in [0, 1]$. We have shown that, in the case under consideration, $S_{m,n}^z(t)$ decays as $t^{-1}$, for any ray inside or to the right of the lightcone. We have not reported here the explicit result since it is lengthy and it would not add much to our discussion, but we show a comparison between analytic predictions obtained with the method described above and numerical simulations in Fig. 6. The simulation for the case $\alpha = 1$ was done considering a partitioning protocol between a RITS on the right and a thermal state with finite (and not infinite as usual) temperature on the left to show that the leading order does not change. As a matter of fact a thermal state is accounted for by the root density and we have shown previously that the contribution from the root density to

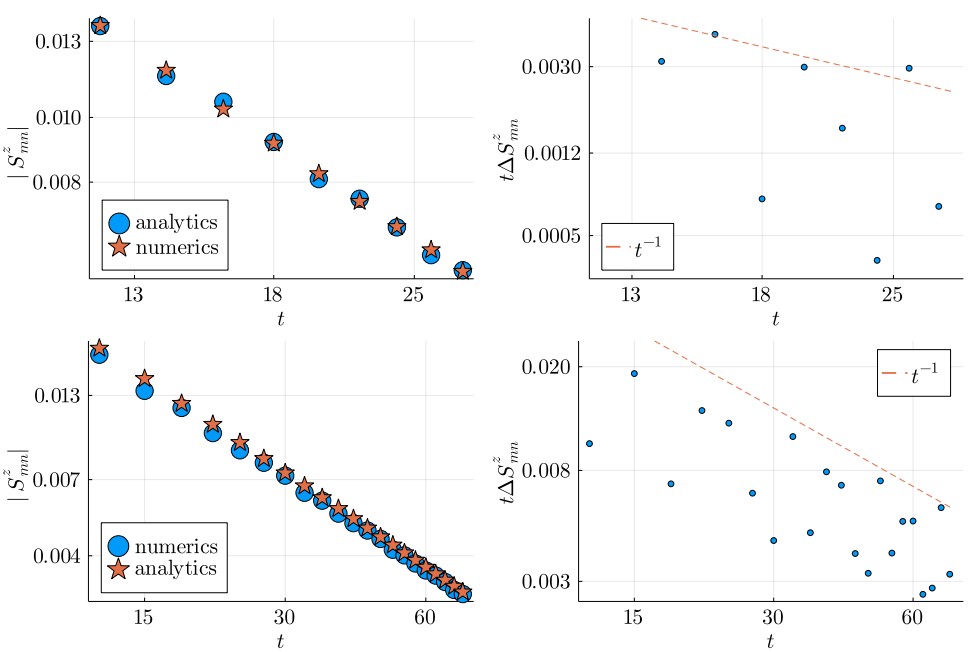

Figure 6: Spin-spin connected correlation matrix for the partitioning protocol between a thermal state with inverse temperature $\beta_L$ on the left and a RITS with $\beta^R$ on the the right. Here $J = 1/2$, $h = 2$, $\beta^R = 0.5$ and $\beta^L$ equals 0 for the top line and 0.5 for the bottom line. On the left we report both the numerical and the analytical prediction. On the right report the absolute value of the difference between the two, rescaled by the behaviour of the leading order. A reference curve is plotted to help the reader identify how fast it goes to zero. The positions $(m, n)$ that we used are $(\zeta t + \frac{ct^\alpha}{2}, \zeta t - \frac{ct^\alpha}{2})$, rounded to the closest integer. Top: $\zeta = 1/2$, $c = 1$, $\alpha = 0$. Bottom: $\zeta = 1/2$, $c = 1/6$, $\alpha = 1$.

the spin-spin connected correlation in this regime goes to zero as $t^{-2}$, so it is subleading with respect to the non-Gaussian contribution.

# 9  Beyond the 2-point spin correlation

In the last three sections, our discussion has been restricted to the spin-spin connected correlation (47). For any initial state, the only fields that give a contribution to such an observable (and to any observable that is written using connected correlations of Majorana fermions of order not higher than 4) are the ones we discussed. So, given any state, we know how the correlation $\langle\sigma_m^z\sigma_n^z\rangle_{c,t}$, with $m - n \sim t^\alpha$, decays for any $\alpha$, depending on which fields are non-zero in the homogeneous states that form the partitioning protocol. However, if we want to consider higher order correlation functions, we have to consider the higher-order fields discussed in Section 4.3. In particular, to study $n$-point spin connected correlations, we generally need to introduce $n + 1$ new fields with respect to the $(n - 1)$-point function. In this section we sketch how one would tackle the asymptotics in the generic case.

Let us first consider the homogeneous case assuming that the relative distance between any two spins does not scale in time, which is the generalization of the regime $\alpha = 0$ in the spin-spin correlation. A qualitative study of the integral that defines each field suggests that, among the new fields that are introduced, the dominant one for large time is the one corresponding

to the Bogoliubov correlation function where there are as many dagger operators as normal ones, which we denote $\xi^{(n)}$:

$$\xi^{(n)phys}_{x_1,\ldots,x_n,t}(p_1,\ldots,p_n) =$$
$$= \int_{-\pi}^{\pi} \frac{\mathrm{d}^n q}{(2\pi)^n} \mathrm{e}^{2\mathrm{i}x_1 q_1} \ldots \mathrm{e}^{2\mathrm{i}x_n q_n} \langle b^\dagger(p_1 - q_1)b(p_1 + q_1) \ldots b^\dagger(p_n - q_n)b(p_n + q_n)\rangle_{c,t}. \quad (105)$$

So far we have discussed the case $n = 1$ and $n = 2$, where $\xi^{(1)} = \rho$ dominates over the auxiliary field and $\xi^{(2)} = \xi$ dominates over the other fields of the case $n = 2$, namely $\Omega$ and $\Upsilon$. To give physical prediction, the field $\xi^{(n)}$ has to be linked to an actual observable. Such an operation involves additional phases that depend on the Bogoliubov angle and the coordinates $z_i$ that generalize the ones we have seen so far, and an integration over all momenta $p_i$. However, the assumption that the relative distance between spins is fixed implies that $x_i - x_n$ and $z_i$ do not scale with time $\forall i \in \{1, ..., n\}$. So the time dependence of the resulting integral comes only from $b(p, t) = \mathrm{e}^{-it\epsilon(p)}b(p, 0)$. Then we can conclude that the stationary points satisfy the system of $2n - 1$ equations

$$\begin{cases} \epsilon'(p_i + q_i) = \epsilon'(p_i - q_i), & \forall i \in \{1, ..., n-1\} \\ \epsilon'(p_n + \sum_{i=1}^{n-1} q_i) = \epsilon'(p_n - \sum_{i=1}^{n-1} q_i), \\ \epsilon'(p_i + q_i) + \epsilon'(p_i - q_i) = \epsilon'(p_n + \sum_{i=1}^{n-1} q_i) + \epsilon'(p_n - \sum_{i=1}^{n-1} q_i) & \forall i \in \{1, ..., n-1\}, \end{cases}$$
$$(106)$$

where we have used that the homogeneity of the state enforces $\sum_{i=1}^{n} q_i = 0$. The system has to be solved for the $2n - 1$ variables $\{p_i\}_{i=1}^{n}$, $\{q_i\}_{i=1}^{n-1}$ and it has stationary curves for solution. For instance, a possible solution is $q_i = 0 \, \forall i \in \{1, ..., n-1\}$, $p_i = p_j \, \forall i, j \in \{1, ..., n\}$. Intuition based on the stationary phase approximation tells that the corresponding multiple integral is expected to decay as $t^{-(n-1)}$, that is a power $t^{-1/2}$ for each of the $2n - 1$ integrals except one, that is compensated by having stationary curves. This allows us to identify the part of the state that gives the leading order for a given connected correlation. For example, $\xi^{(n)}$ should give in general a contribution $t^{-(n-1)}$ to the $2n$-point fermionic connected correlation and $t^{-2(n-1)}$ to the $4n$-point one.

Let us also comment what we expect in the inhomogeneous case, still under the assumption that the relative distance between spins is fixed. We expect something similar to what we saw for $n = 2$: the stationary curves are the same of the homogeneous case but only part of them gives a contribution. This implies that the qualitative behavior of the inhomogeneous case is the same as in the homogeneous one. Note that, according to this argument, the root density is the only field that can ever give a finite contribution to local observables.

Finally, to study what happens to the regime $\alpha > 0$, one should first of all look at what happens to those stationary curves. We will not go beyond this heuristic qualitative argument, but we stress out that the asymptotic analysis would be a direct generalization of what we have seen in the previous sections.

## 10    Discussion of the results

In the main part of the paper we focused on the spin-spin connected correlation function $S^z_{m,n}(t) \equiv \langle \sigma^z_m(t)\sigma^z_n(t)\rangle - \langle \sigma^z_m(t)\rangle \langle \sigma^z_n(t)\rangle$, with $m - n \sim t^\alpha$ and $\alpha \in [0, 1]$. We studied three partitioning protocols that were engineered in such a way to isolate the independent

contributions to $S_{m,n}^z$ and we argued that the contributions should be summed independently for a generic partitioning protocol in which all the dynamical fields have non-zero value over at least one of the homogeneous states involved in the partitioning protocol. We saw that the leading order for large time is obtained by considering the root density only, and neglecting all the other fields, not only for $\alpha = 0$, when the support of the observable is fixed and finite (i.e. the GHD's regime of validity), but also for $\alpha < 1/2$. We also gave an argument according to which the crossover scaling $\alpha = 1/2$ does not depend on the special observable we are looking at. Indeed, when considering observables with larger (but finite) support in the fermionic representation, the higher-order fields needed to describe it give a contribution that decays faster to zero than the one coming from the root density.

We point out that, for $\alpha = 0$, the leading order from the fields different from the root density is in general of the same order $(t^{-1})$ as the late-time corrections from higher-order GHD, implying that, unless the homogeneous states forming the partitioning protocol are stationary, a higher-order GHD that relies on the root density only can not in general give the right predictions. Note the important role of non-Gaussian correlations in the initial state on the relaxation process. For instance, in global homogeneous quenches from a Gaussian state, the GGE value of the spin-spin correlation function in the Ising model is attained with corrections decaying as $t^{-3}$. Non-Gaussian correlations slow this process down to $t^{-1}$.

The results above also show how Gaussianification takes place in partitioning protocols. We argued that the expectation value of local observables at the Euler scale can be described using a Gaussian state that depends on the position of the observable. We also answered to the question *how local is "local enough"* for the spin-spin connected correlation to Gaussianize. The answer is that the distance between the spins should not grow faster than $t^{1/2}$, since otherwise the non-Gaussian fields become dominant.

As for the generality of our arguments, the generalization to any quadratic model is quite straightforward. As a matter of fact one needs to change the dispersion relation, the Bogoliubov angle and, at most, introduce other angles to diagonalize the Hamiltonian (see e.g. [41]), but the definition of the dynamical fields would stay the same. Although we did not carry out the computation in the general case, we argued that the qualitative behaviors of the independent contributions to the spin-spin connected correlation would not change, except at most, the one linked to the auxiliary field, which may decay as $t^{-1}$ instead of $t^{-3+2\alpha}$, as it happens in the XY quantum spin chain when the external magnetic field is such that the dispersion relation has more than two stationary points. Note that this implies that a Gaussian partitioning protocol in the Ising chain attains its locally quasi-stationary state faster than in those other spin chains.

The generalization to other observables with 4-sites support in fermions is also straightforward: the precise expression of the observable would change, but, since our arguments rely only on the structure of time evolution and the singularities that define the partitioning protocol, the conclusions would be the same. We also discussed how observables that require a finite number of higher order connected correlations can be included in the picture: $n + 1$ new independent dynamical fields should be introduced to account for the $2n$-point fermionic connected correlation. We also gave a qualitative argument on how to obtain their scaling.

It remains the question of what happens to local observables the are not local in fermions, such as $\langle \sigma_\ell^x \rangle$. We can not apply our arguments to those observables because infinitely-many (in the thermodynamic limit) dynamical fields are involved. The situation for those observables is generally more complicated and it even turns out that there are situations in which the usual GGE/GHD is not enough to describe the late-time behavior of those observable, despite

the model being free [72, 73]. We leave the inclusion of those observables in our description as a prospect for a future work.

# Acknowledgements

I am grateful to Maurizio Fagotti for giving me the idea behind this work. I thank also Lenart Zadnik and Vanja Marić for useful discussions. This work was supported by the European Research Council under the Starting Grant No. 805252 LoCoMacro.

# A   Complement to the root-density partitioning protocol

In the main text – Eqs. (65) and (66) – we identified two contributions to the correlation matrix coming from the field $\rho_{x,t}(p)$ and we showed how to compute one of them. In this section we use the same ideas to compute the second contribution. Fig. 7 shows how one should deform the contours in this case for the same rays used in Fig. 2. As can be inferred from the pictures, the contour for the residue contribution can always be chosen to be outside the unit circle. For both rays inside the lightcone, it comes in the form of two curves that are symmetric with reflection with respect to the origin. Calling $\gamma$ one of the two pieces, we have that the residue contribution is

$$\mathcal{R}_2^\rho[f(w_1, w_2)](z, t) := 2i \int_\gamma \frac{dw}{2\pi} e^{-2z \ln w} f(w, w). \tag{107}$$

For $\alpha = 1$, the real part of $-2z \ln w$ is negative outside the unit circle (recall $z > 0$), so that such contribution is always exponentially suppressed. As for the previous case, in this regime we can just deform the two initial contours as if the pole was not there. In the end, the leading contribution is given by a standard application of the saddle-point method to the remaining double integral over $\mathcal{C}_1'$ and $\mathcal{C}_2'$. When both rays are inside the lightcone, there are four saddle points to be considered for each of those integrals, yielding a leading order of $\mathcal{I}_2^\rho[f(w_1, w_2)](z, t) \sim t^{-1}$. Whenever at least one of the two rays is outside the lightcone, at least one integral of the double integral has no saddle points and the domain can be deformed in such a way that the function under integration is exponentially suppressed.

In the case of $0 < \alpha < 1/2$, as we have already discussed, the residue contribution is not negligible anymore, but gives instead what turns out to be the leading contribution to the initial double-integral. As we have done for the previous contribution, we linearize it around its extrema. Calling $e^{i\bar{p}_j} + \delta w_j e^{i\bar{p}_j}$, with $\delta w \sim t^{\alpha-1}$ and $j \in \{1, 2\}$, the extrema of the first piece of $\gamma$:

$$\mathcal{R}_2^\rho[f(w_1, w_2)](z, t) \sim 2i \sum_{j=1}^2 (-1)^{j-1} \int_0^\Delta \frac{ds}{2\pi} e^{i\bar{p}_j} e^{-2z \ln(e^{i\bar{p}_j} + \delta w_j e^{i\bar{p}_j} + s e^{i\bar{p}_j})} f(e^{i\bar{p}_j}, e^{i\bar{p}_j}), \tag{108}$$

where $\Delta \sim t^{-\omega}$, $\omega \in (\alpha/2, \alpha)$. Computing the integral gives

$$\mathcal{R}_2^\rho[f(w_1, w_2)](z, t) \sim -i \sum_{j=1}^2 (-1)^j \frac{1}{2\pi z} e^{i\bar{p}_j} e^{-i2z\bar{p}_j} f(e^{i\bar{p}_j}, e^{i\bar{p}_j}), \tag{109}$$

where we have used $(1 + \delta w + \Delta)^{-2z} \sim e^{-2z\delta w}$, to conclude that the term containing $\Delta$ goes exponentially to zero, and $(1 + \delta w)^{-2z} \sim 1$. This is the leading contribution to the integral. Note that there is an implicit dependence on $x$, since $\bar{p}_j$ depends on $x/t$.

For $1/2 < \alpha < 1$ instead, the residue contribution is exponentially suppressed for the same reasons that we discussed in the main text and the leading contribution comes from the remaining double integral. Using the same notations introduced for the previous contribution,

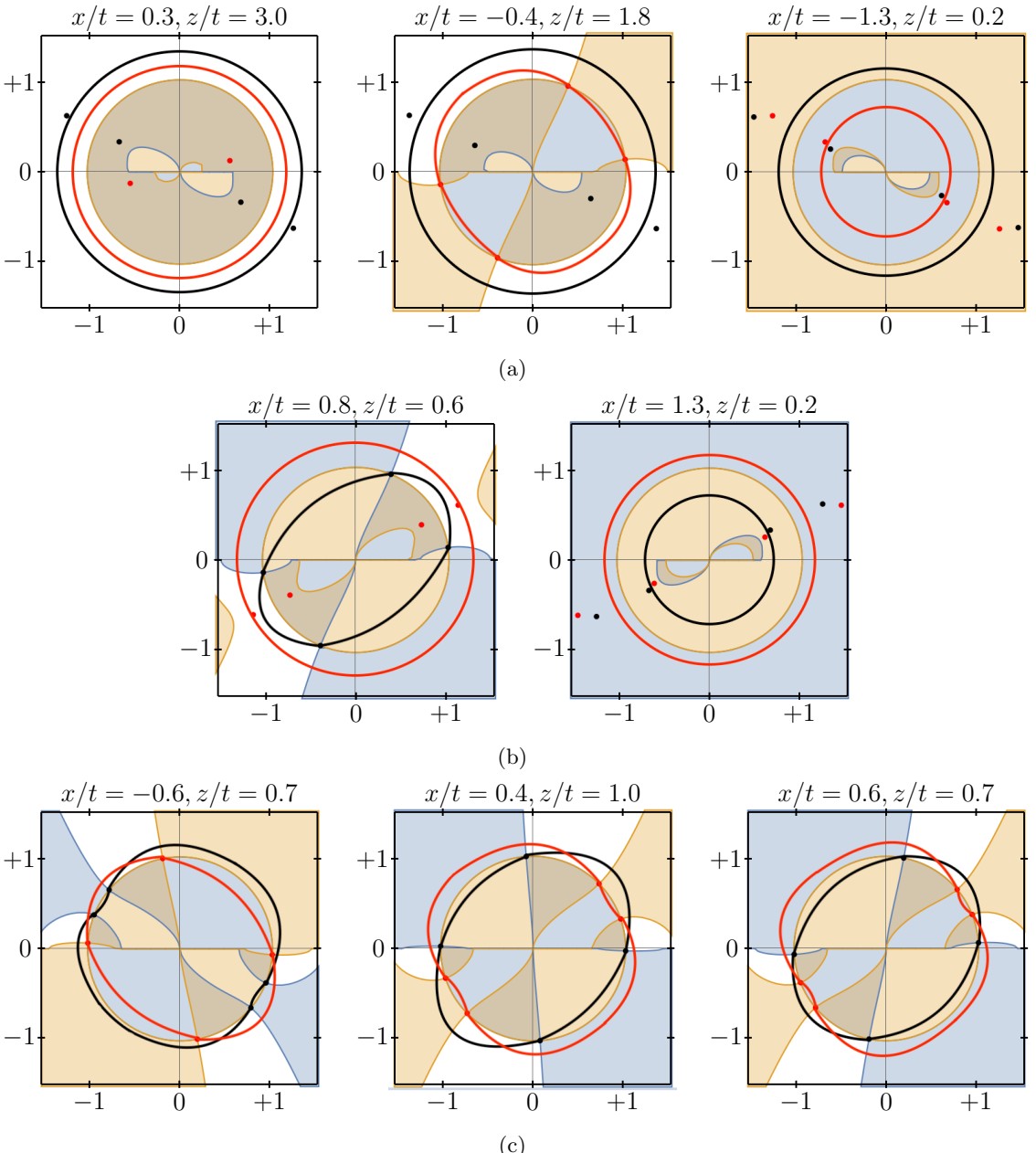

Figure 7: Regions in which $\mathrm{Re}(S_{(x-\frac{z}{2})/t}(a+\mathrm{i}b)) > 0$ (blue region) and $\mathrm{Re}(S_{(x+\frac{z}{2})/t}(a+\mathrm{i}b)) < 0$ (orange region) in function of $a$ (horizontal axis) and $b$ (vertical axis). The rest is the same of Fig. 2.

we get

$$
\mathcal{I}_2^\rho[f(w_1, w_2)](x, z, t) \sim \sum_{j=1}^{4} \int_{-\Delta}^{\Delta} \frac{\mathrm{d}s^+ \mathrm{d}s^-}{(2\pi)^2} \mathrm{e}^{\mathrm{i}\phi_j^+} \mathrm{e}^{\mathrm{i}\phi_j^-} f(\mathrm{e}^{\mathrm{i}\bar{p}_j}, \mathrm{e}^{\mathrm{i}\bar{p}_j}) \times
$$

$$
\times \frac{\mathrm{e}^{tS_{(x-\frac{z}{2})/t}(\mathrm{e}^{\mathrm{i}\bar{p}_j - \mathrm{i}\delta p_j}) - tS_{(x+\frac{z}{2})/t}(\mathrm{e}^{\mathrm{i}\bar{p}_j + \mathrm{i}\delta p_j}) - \frac{t}{2}|S''_{x/t}(\mathrm{e}^{\mathrm{i}\bar{p}_j + \mathrm{i}\delta p_j})|(s^-)^2 - \frac{t}{2}|S''_{x/t}(\mathrm{e}^{\mathrm{i}\bar{p}_j - \mathrm{i}\delta p_j})|(s^+)^2}}{-2\mathrm{i}\mathrm{e}^{\mathrm{i}\bar{p}_j}\sin(\delta p_j) + s^+ \mathrm{e}^{\mathrm{i}\phi_j^+} - s^- \mathrm{e}^{\mathrm{i}\phi_j^-}}, \quad (110)
$$

where we took $\Delta \sim t^\omega, \omega \in (-1/2, \min\{\alpha-1, -1/3\})$ and the two phases $\phi_j^\pm$ are approximately the same of the case discussed in the main text. Performing the Gaussian integrals we get

$$
\mathcal{I}_2^\rho[f(w_1, w_2)](x, z, t) \sim \sum_{j=1}^{4} \frac{\mathrm{e}^{-tS_{(x+\frac{z}{2})/t}(\mathrm{e}^{\mathrm{i}\bar{p}_j + \mathrm{i}\delta p_j}) + tS_{(x-\frac{z}{2})/t}(\mathrm{e}^{\mathrm{i}\bar{p}_j - \mathrm{i}\delta p_j})}}{-4\mathrm{i}\pi t |S''_{x/t}(\mathrm{e}^{\mathrm{i}\bar{p}_j})| \mathrm{e}^{\mathrm{i}\bar{p}_j} \delta p_j} \mathrm{e}^{\mathrm{i}\phi_j^+} \mathrm{e}^{\mathrm{i}\phi_j^-} f(\mathrm{e}^{\mathrm{i}\bar{p}_j}, \mathrm{e}^{\mathrm{i}\bar{p}_j}). \quad (111)
$$

This is combined in the text we the other contribution $\mathcal{I}_1^\rho$ to get the behavior of the correlation matrix elements.

# B Complement to the auxiliary-field partitioning protocol

In this section we compute the asymptotics of the integrals (96) in the scaling limit $t \to +\infty$, with $x = \zeta t, z = ct^\alpha > 0, \alpha \in [0, 1], \zeta \neq 0$. We can use the same arguments that we used in the root-density case. The saddle-point landscape for the first term of each integral is summarized in Fig. 8 and the situation is similar for the second term (see Section B.4). As can be inferred from the figure, it is always possible to deform $\mathcal{C}_1$ and $\mathcal{C}_2$ to $\mathcal{C}_1'$ and $\mathcal{C}_2'$ such that the real part of the phase is negative almost everywhere. If $x/t < 0$, we do not get any residue contribution after the deformation. If instead $x/t > 0$, the two contours have to be fully exchanged and we get a residue contribution in the form of an integral over a closed curve around the origin. If at least one ray is outside the lightcone, meaning that $|\zeta + \frac{z}{2t}| > v_{max}$ or $|\zeta - \frac{z}{2t}| > v_{max}$, then the double integral that one gets after the deformation is exponentially suppressed (the real part of the phase is negative everywhere) and we are left only with the residue contribution, if present. When both the rays are inside the lightcone, the function under the double-integral after the deformation is exponentially suppressed except in the saddle points, where the real part of the phase is zero. A standard application of the saddle point method shows that the contribution of the double integral in this case is of order $\sim t^{-1}$, independently from $\alpha$.

The residue contribution, whenever it is present, comes in the form

$$
\mathcal{R}_1^\psi(z, t) \equiv -\frac{\mathrm{i}}{2} \operatorname{Re} \int_\gamma \frac{\mathrm{d}w}{2\pi w} \psi^R(-2\mathrm{i}\ln(w)) \left( \mathrm{e}^{2tS_{\frac{z}{2t}}(w)} - \mathrm{e}^{2tS_{-\frac{z}{2t}}(w)} \right) \quad (112)
$$

for $\left(\Gamma_x^{z;\psi}\right)_{11}$, and

$$
\mathcal{R}_2^\psi(z, t) \equiv \frac{\mathrm{i}}{2} \operatorname{Im} \int_\gamma \frac{\mathrm{d}w}{2\pi w} \psi^R(-2\mathrm{i}\ln w) \left( \mathrm{e}^{2tS_{\frac{z}{2t}}(w)} \mathrm{e}^{\mathrm{i}\theta(-2\mathrm{i}\ln w)} - \mathrm{e}^{2tS_{-\frac{z}{2t}}(w)} \mathrm{e}^{-\mathrm{i}\theta(-2\mathrm{i}\ln w)} \right) \quad (113)
$$

for $\left(\Gamma_x^{z;\psi}\right)_{21}$. Here $\gamma$ is a closed curve in the annulus of analiticity around the unit circle. Note that for $|z/t| > 2v_{max}$ we can deform $\gamma$ (independently for each of the two terms in a

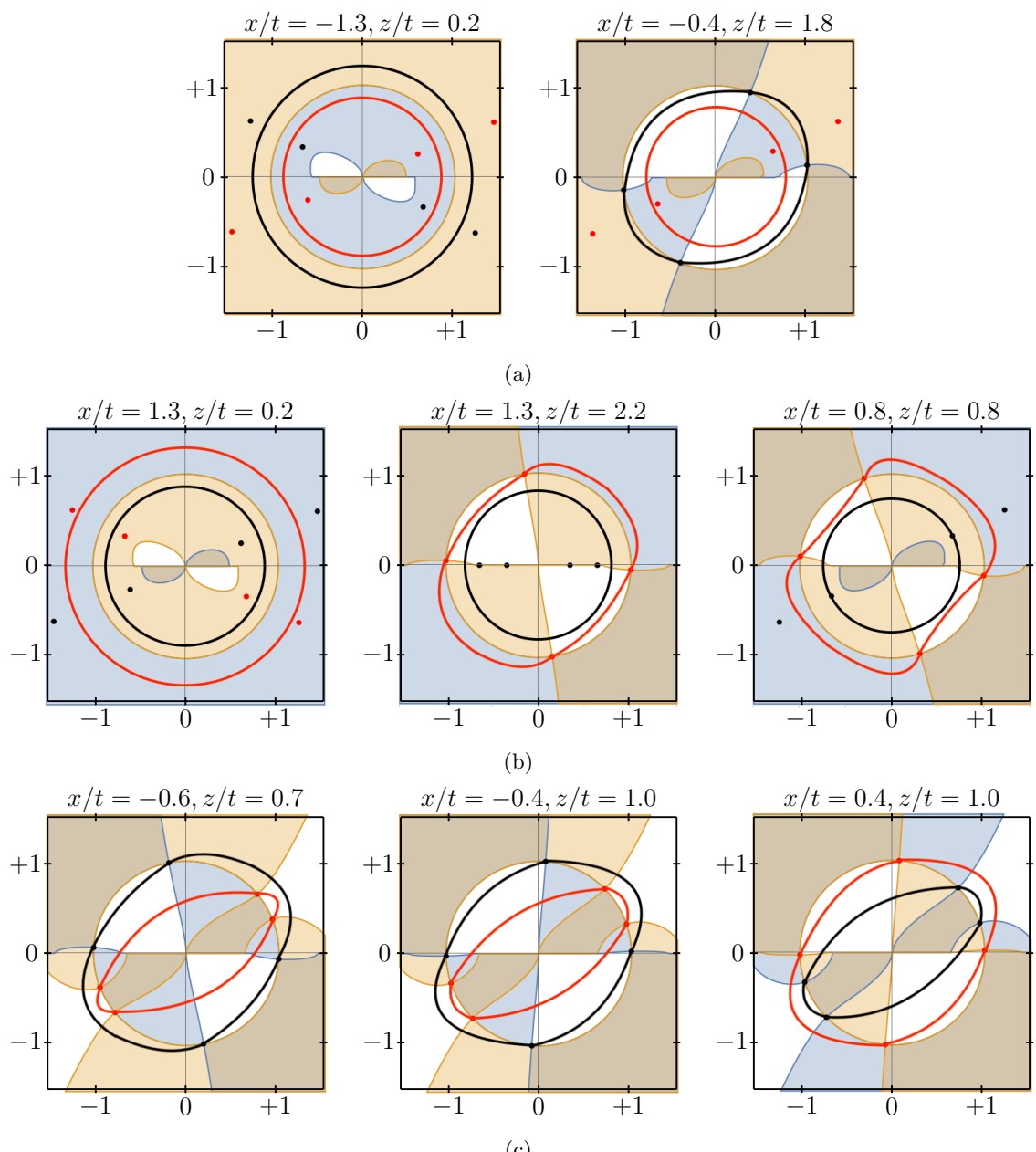

Figure 8: Regions in which $\mathrm{Re}(S_{(x+\frac{z}{2})/t}(a+\mathrm{i}b)) > 0$ (blue region) and $\mathrm{Re}(S_{(-x+\frac{z}{2})/t}(a+\mathrm{i}b)) > 0$ (orange region) in function of $a$ (horizontal axis) and $b$ (vertical axis). We consider a TFIC with $J = 1/2$, so that $v_{max} = 1$, and $h = 2$. The black and red lines represent respectively the curves $\mathcal{C}'_1$ and $\mathcal{C}'_2$ obtained by deforming $\mathcal{C}_1$ and $\mathcal{C}_2$ as detailed in the text. (a) Case in which the ray $(x - \frac{z}{2})/t, z > 0$, is on the left of the lightcone. (b) Case in which the ray $(x + \frac{z}{2})/t, z > 0$, is on the right of the lightcone and $(x - \frac{z}{2})/t$ is either inside or to the right of it. (c) Case in which the two rays $(x \pm \frac{z}{2})/t$ are both inside the lightcone.

given integral) in such a way that the function under integration is exponentially small; we conclude that in such a case the residue contribution is exponentially small, consistently with the Lieb-Robinson bound, since sites with $|z/t| > 2v_{max}$ are causally disconnected. In all the cases $|z/t| < 2v_{max}$, instead, we have four saddle points on the unit circle, in which the real part of the exponent is zero. We discuss this case distinguishing between the different scaling regimes.

## B.1  Linear scaling

Calling $e^{i\bar{p}_j^{\pm}}, \forall j \in \{1, 2, 3, 4\}$ the saddle points for the $S_{\pm \frac{z}{2t}}(w)$, the residue contributions, that are nonzero only for $\zeta > 0$, can be written as

$$
\mathcal{R}_1^{\psi}(z,t) \sim -\frac{i}{2} \operatorname{Re} \sum_{j=1}^{4} \int_{-\infty}^{\infty} \frac{dy}{2\pi}
$$

$$
\left( e^{2its\frac{z}{2t}(\bar{p}_j^+) - t|s''_{\frac{z}{2t}}(\bar{p}_j^+)|y^2} e^{i\phi_j^+} e^{-i\bar{p}_j^+} \psi^R(2\bar{p}_j^+) - e^{2its-\frac{z}{2t}(\bar{p}_j^-) - t|s''_{-\frac{z}{2t}}(\bar{p}_j^-)|y^2} e^{i\phi_j^-} e^{-i\bar{p}_j^-} \psi^R(2\bar{p}_j^-) \right),
$$
(114)

$$
\mathcal{R}_2^{\psi}(z,t) \sim \frac{i}{2} \operatorname{Im} \sum_{j=1}^{4} \int_{-\infty}^{\infty} \frac{dy}{2\pi} \left( e^{2its\frac{z}{2t}(\bar{p}_j^+) - t|s''_{\frac{z}{2t}}(\bar{p}_j^+)|y^2} e^{i\phi_j^+} e^{i\theta(2\bar{p}_j^+)} e^{-i\bar{p}_j^+} \psi^R(2\bar{p}_j^+) + \right.
$$

$$
\left. - e^{2its-\frac{z}{2t}(\bar{p}_j^-) - t|s''_{-\frac{z}{2t}}(\bar{p}_j^-)|y^2} e^{i\phi_j^-} e^{-i\theta(2\bar{p}_j^-)} e^{-i\bar{p}_j^-} \psi^R(2\bar{p}_j^-) \right), \quad (115)
$$

where

$$
e^{2i\phi_j^{\pm} - 2i\bar{p}_j^{\pm} - i\frac{\pi}{2} + i\pi\Theta(-s''_{\pm\frac{z}{2t}}(2\bar{p}_j^{\pm}))} = -1 \quad \Rightarrow \quad \phi_j^{\pm} = \bar{p}_j^{\pm} - \frac{\pi}{4} + \frac{\pi}{2}\Theta(\epsilon''(2\bar{p}_j^{\pm})) + \pi\kappa_j^{\pm}, \quad (116)
$$

with $\kappa_j^{\pm} \in \mathbb{Z}$ determined by the sign of circulation of the integration contour. Performing the Gaussian integration we get

$$
\mathcal{R}_1^{\psi}(z,t) \sim -\frac{i}{4\sqrt{\pi t}} \operatorname{Re} \sum_{j=1}^{4} \left( \frac{e^{2its\frac{z}{2t}(\bar{p}_j^+)} e^{i\phi_j^+} e^{-i\bar{p}_j^+} \psi^R(2\bar{p}_j^+)}{\sqrt{|s''_{\frac{z}{2t}}(\bar{p}_j^+)|}} - \frac{e^{2its\frac{z}{2t}(\bar{p}_j^-)} e^{i\phi_j^-} e^{-i\bar{p}_j^-} \psi^R(2\bar{p}_j^-)}{\sqrt{|s''_{\frac{z}{2t}}(\bar{p}_j^-)|}} \right),
$$
(117)

$$
\mathcal{R}_2^{\psi}(z,t) \sim \frac{i}{4\sqrt{\pi t}} \operatorname{Im} \sum_{j=1}^{4}
$$

$$
\left( \frac{e^{2its\frac{z}{2t}(\bar{p}_j^+)} e^{i\phi_j^+} e^{i\theta(2\bar{p}_j^+)} e^{-i\bar{p}_j^+} \psi^R(2\bar{p}_j^+)}{\sqrt{|s''_{\frac{z}{2t}}(\bar{p}_j^+)|}} - \frac{e^{2its-\frac{z}{2t}(\bar{p}_j^-)} e^{i\phi_j^-} e^{-i\theta(2\bar{p}_j^-)} e^{-i\bar{p}_j^-} \psi^R(2\bar{p}_j^-)}{\sqrt{|s''_{\frac{z}{2t}}(\bar{p}_j^-)|}} \right). \quad (118)
$$

At this point let us assign the numbers $\{1, 2, 3, 4\}$ to the saddle points that, for $z = 0$, are in $\{\pi/2, \pi, -\pi/2, 0\}$ respectively (since they change continuously with $z/t$, the definition is

not ambiguous). Then we have that $\bar{p}_1^\pm = \bar{p}_3^\pm + \pi$, $\bar{p}_2^\pm = \bar{p}_4^\pm + \pi$, $\epsilon''(2\bar{p}_1^\pm) < 0$, $\epsilon''(2\bar{p}_2^\pm) > 0$, $\kappa_1^\pm = 1 = \kappa_3^\pm$, $\kappa_2^\pm = 0 = \kappa_4^\pm$. Moreover, we have $\bar{p}_1^+ = \pi - \bar{p}_1^-$ and $\bar{p}_2^+ = -\bar{p}_2^-$, and $\sqrt{|s_\zeta''(p)|} = 2\sqrt{|\epsilon''(2p)|}$. Everything considered, the residue contribution gives the leading order for the correlation matrix elements for $\alpha = 1$ and $\zeta > 0$:

$$
\begin{aligned}
\left(\Gamma_x^{z;\psi}\right)_{11} &= -\left(\Gamma_x^{z;\psi}\right)_{22} \sim -\frac{i}{2\sqrt{\pi t}} \operatorname{Re} \sum_{j=1}^{2} e^{i\frac{\pi}{4}(5-2j)} \frac{e^{2its\frac{z}{2t}(\bar{p}_j^+)}\psi^R(2\bar{p}_j^+)}{\sqrt{|\epsilon''(2\bar{p}_j^+)|}}, \\
\left(\Gamma_x^{z;\psi}\right)_{21} &= -\left(\Gamma_x^{-z;\psi}\right)_{12} \sim \frac{i}{2\sqrt{\pi t}} \operatorname{Im} \sum_{j=1}^{2} e^{i\frac{\pi}{4}(5-2j)} \frac{e^{2its\frac{z}{2t}(\bar{p}_j^+)}e^{i\theta(2\bar{p}_j^+)}\psi^R(2\bar{p}_j^+)}{\sqrt{|\epsilon''(2\bar{p}_j^+)|}}.
\end{aligned}
\tag{119}
$$

Note that the existence of a non-exponentially-suppressed residue contribution depends only on the two conditions $x > 0$, $|z/t| < 2v_{max}$, which means that we have a residue contribution also outside (to the right) of the lightcone. For $-v_{max} < \zeta < 0$, instead, the residue contribution is zero and, as already pointed out, the leading order comes from the double integral over $\mathcal{C}_1'$ and $\mathcal{C}_2'$.

## B.2   Sub-linear scaling

Let us now look at the case in which $z \sim t^\alpha$, $\alpha \in (0,1)$, starting with the case $\zeta > 0$. The arguments above for $\alpha = 1$ still hold and the double integral that one gets after the deformation gives the same $t^{-1}$ contribution, but there is a subtlety in the application of the saddle-point-method to the residue contribution. The peculiarity is that the four saddle point $\bar{p}_j^+$ belong to $\{e^{i\delta p_4}, e^{i\frac{\pi}{2}+i\delta p_1}, e^{i\pi+i\delta p_2}, e^{-i\frac{\pi}{2}+i\delta p_3}\}$, with $\delta p_j \sim t^{\alpha-1}$, and the function $\psi^R(p)$ is zero in 0 and $\pi$. Importantly, this does not depend on the particular choice of the state that we made: $\psi^R(p)$ essentially represents the field $\psi_{x,t}(p)$ for a homogeneous case, for which it can be shown $\psi_{x,t}(p) \propto \langle b(p)b(-p)\rangle$, that equals zero for $p = 0$ and $p = \pi$ for any state. Therefore, when it is computed in the saddle points, it can be expanded and we get $\psi^R(2\bar{p}_j^+) \sim 2\delta p_j \psi'(0)$ for $j \in \{2,4\}$ and $\psi^R(2\bar{p}_j^+) \sim 2\delta p_j \psi'(\pi)$ for $j \in \{1,3\}$; combined with the $t^{-1/2}$ that one gets from the saddle point we can conclude that the residue contribution in this case is of order $t^{-3/2+\alpha}$. Note that for $\alpha < 1/2$, this becomes smaller than the double-integral contribution inside the lightcone, so that, for both rays inside the lightcone, it is actually the double-integral that gives the leading contribution.

We should point out that in this case the TFIC is special with respect to generic free chains. As a matter of fact, since in the generic case the dispersion relation have additional stationary points that are different from 0 and $\pi$, the symmetries of the auxiliary field do not imply that it is zero in such points, so that the generic decaying of the contribution under consideration is $t^{-1/2}$ for any observable whose support scales as $t^\alpha$, $\forall \alpha \in (0,1)$. That is the case e.g. of the XY quantum spin chain with transverse magnetic field when the dispersion relation has more than two stationary points.

## B.3   Constant distance

Let us finally discuss what happens for $\alpha = 0$. As the case $\alpha \in (0,1)$, the only thing that is affected is the residue contribution and only for spin chains whose dispersion relation has

stationary points in $\pi m$, with $m \in \mathbb{Z}$, like the TFIC. We have

$$\mathcal{R}_1^\psi(z,t) = \mathrm{i}\,\mathrm{Re}\int_{-\pi}^{\pi}\frac{\mathrm{d}p}{2\pi}\psi^R(p)\mathrm{e}^{-2\mathrm{i}t\epsilon(p)}\sin(zp)\,, \tag{120}$$

$$\mathcal{R}_2^\psi(z,t) = -\mathrm{i}\,\mathrm{Im}\int_{-\pi}^{\pi}\frac{\mathrm{d}p}{2\pi}\psi^R(p)\mathrm{e}^{-2\mathrm{i}t\epsilon(p)}\sin\left(zp+\theta(p)\right)\,. \tag{121}$$

Now, since the dispersion relation is stationary in $0$ and $\pi$, the expansion around those points gives

$$\begin{aligned}
\mathcal{R}_1^\psi(z,t) &\sim \mathrm{i}\,\mathrm{Re}\Bigg(\int_{-\infty}^{+\infty}\frac{\mathrm{d}p}{2\pi}\psi^{R\prime}(0)\mathrm{e}^{-2\mathrm{i}t\epsilon(0)}\mathrm{e}^{-\mathrm{i}t\epsilon''(0)p^2}zp^2+ \\
&\qquad\qquad + (-1)^z\int_{-\infty}^{+\infty}\frac{\mathrm{d}p}{2\pi}\psi^{R\prime}(\pi)\mathrm{e}^{-2\mathrm{i}t\epsilon(\pi)}\mathrm{e}^{-\mathrm{i}t\epsilon''(\pi)p^2}zp^2\Bigg) \\
&\sim \mathrm{i}\,\mathrm{Re}\Bigg(\frac{1}{4\sqrt{\pi}}t^{-3/2}z\mathrm{e}^{-\mathrm{i}\frac{\pi}{4}}\left(\psi^{R\prime}(0)\mathrm{e}^{-2\mathrm{i}t\epsilon(0)}(\epsilon''(0))^{-3/2}+(-1)^z\psi^{R\prime}(\pi)\mathrm{e}^{-2\mathrm{i}t\epsilon(\pi)}(\epsilon''(\pi))^{-3/2}\right)\Bigg)\,,
\end{aligned} \tag{122}$$

and similarly for $\mathcal{R}_2^\psi$. So the residue contribution in this case decays as $t^{-3/2}$ and it gives the leading order only to the right of the lightcone, since inside the lightcone the double integral over $\mathcal{C}_1'$ and $\mathcal{C}_2'$ scales as $t^{-1}$.

## B.4  Second contribution

In this section we show the deformations envisaged in the main text for the second term of the integrals (96) representing the contribution of the auxiliary field to the correlation matrix. They are reported in Fig. 9 and the same considerations done for the first contribution hold: it is always possible to deform $\mathcal{C}_1$ and $\mathcal{C}_2$ to $\mathcal{C}_1'$ and $\mathcal{C}_2'$ such that the real part of the phase is negative almost everywhere; we get a residue contribution from the deformation only for $x/t > 0$, when the two contours have to be fully exchanged.

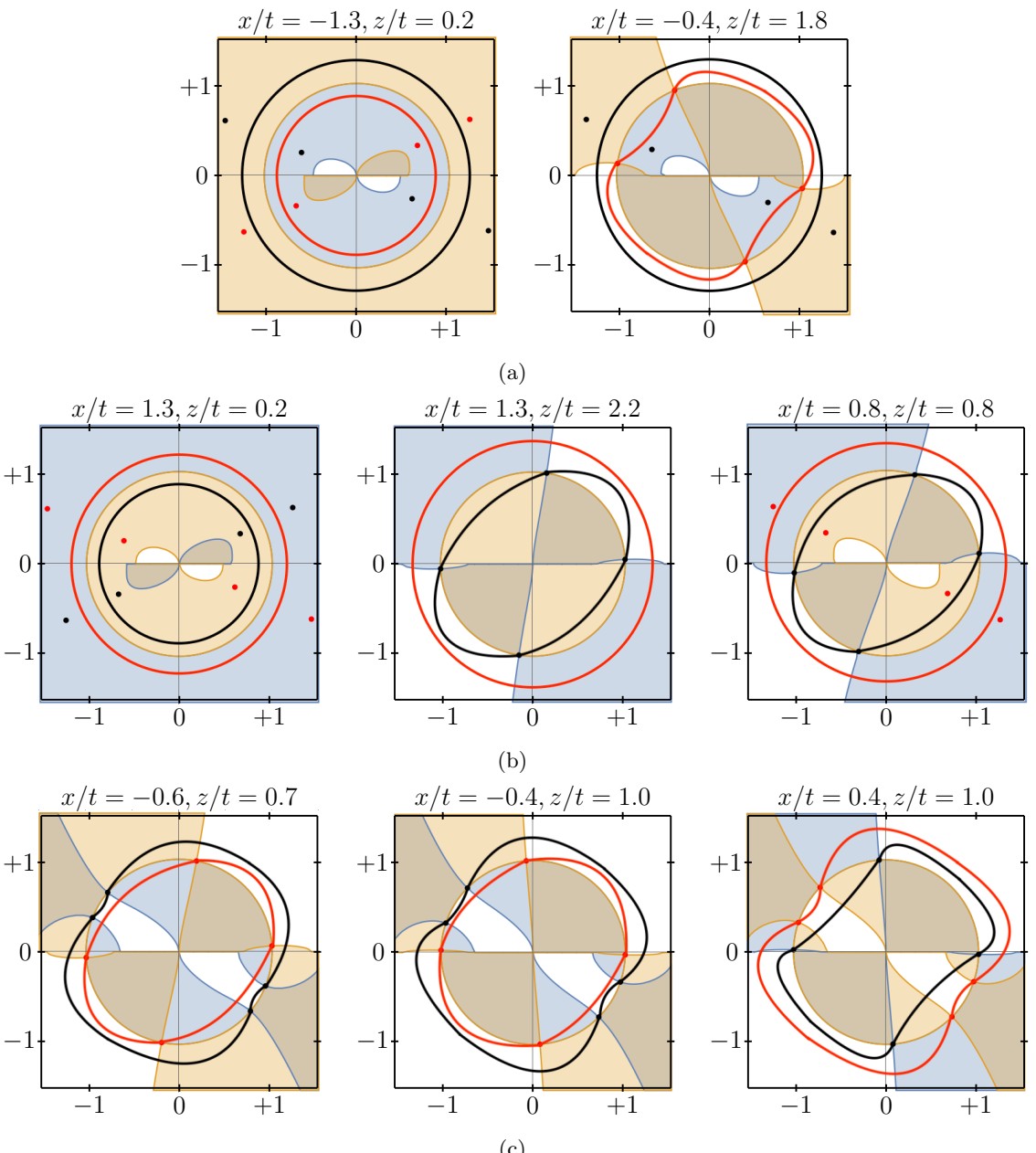

Figure 9: Regions in which $\mathrm{Re}(S_{(x-\frac{z}{2})/t}(a+\mathrm{i}b)) > 0$ (blue region) and $\mathrm{Re}(S_{(-x-\frac{z}{2})/t}(a+\mathrm{i}b)) > 0$ (orange region) in function of $a$ (horizontal axis) and $b$ (vertical axis). The rest is the same of Fig. 8.

# C Non-Gaussian partitioning protocol

In this appendix, we consider the partitioning protocol (53) between a RITS and an infinite-temperature state, described by the density matrix

$$\hat{\rho} = Z^{-1} e^{\beta \sum_{l=0}^{\infty} \sigma_l^z \sigma_{l+1}^z}, \tag{123}$$

with $\beta > 0$. By writing explicitly the correlation functions of Bogoliubov operators, we give an integral representation of the contribution from the field $\xi_{x_1,x_2,t}(p_1, p_2)$ to the spin-spin connected correlation function (47).

## C.1 General expression for fermionic 4-point correlations

We can use the Jordan-Wigner transformation (2) to express the 2- and 4-point correlation functions of Majorana fermions in terms of spin expectation values. We assume that the state under consideration has the same symmetries of our example, i.e. that it is invariant under rotations of $\pi$ around the $x$ and the $y$ axis, and it is invariant under $\sigma_\ell^\alpha \to -\sigma_\ell^\alpha, \forall \ell \in \mathbb{Z} \wedge \forall \alpha \in \{x, y\}$.

Observe first that, whenever we consider the expectation value of a string of Pauli matrices, it vanishes if any component of the string is a $\sigma^x$ or a $\sigma^y$, or if the number of $\sigma^z$'s is odd. For example, assuming $m \leq n$, the only way in which

$$\langle a_{2m} a_{2n} \rangle = \left\langle \sigma_m^y \left( \prod_{j=m}^{n-1} \sigma_j^z \right) \sigma_n^y \right\rangle \tag{124}$$

can satisfy all of the symmetry requirements above, is for $m = n$, while there is no way that $\langle a_{2m-1} a_{2n} \rangle$ can satisfy them, leading to

$$\langle a_i a_j \rangle_0 = \delta_{ij}. \tag{125}$$

This proves that the correlation matrix for this state is zero. The computation of the 4-point correlation in Majorana operators is slightly more complicated, but proceeds on the same lines. It can be shown that it is nonzero only if two of the indices are odd and the other two are even:

$$
\begin{aligned}
C_{2m_1+j_1, 2m_2+j_2, 2m_3+j_3, 2m_4+j_4} = \\
= \Big[ \delta_{m_1 m_2} \delta_{m_3, m_4} \langle \sigma_{m_1}^z \sigma_{m_3}^z \rangle_0 - \frac{1}{2} \left( \delta_{m_1 m_3} \delta_{m_2 m_4} + \delta_{m_1 m_4} \delta_{m_2 m_3} \right) \langle \sigma_{m_1}^z \sigma_{m_2}^z \rangle_0 \Big] \times \\
\times (\sigma^y \otimes \sigma^y)_{2(j_1-1)+j_3, 2(j_2-1)+j_4} + \\
+ \frac{1}{2} \left( \delta_{m_1 m_3} \delta_{m_2 m_4} - \delta_{m_1 m_4} \delta_{m_2 m_3} \right) \langle \sigma_{m_1}^z \sigma_{m_2}^z \rangle_0 \times \\
\times (\mathbb{I} \otimes \mathbb{I} - \sigma^z \otimes \sigma^z - \sigma^x \otimes \sigma^x)_{2(j_1-1)+j_3, 2(j_2-1)+j_4}, \tag{126}
\end{aligned}
$$

for any $m_1, m_2, m_3, m_4 \in \mathbb{Z}$ and $j_1, j_2, j_3, j_4 \in \{1, 2\}$.

At this point we can use the Bogoliubov transformation to write the Bogoliubov 4-point connected correlation function in terms of Majorana fermions using Eq. (13):

$$\langle b^\dagger(p_1)b(p_2)b^\dagger(p_3)b(p_4)\rangle_c = \frac{1}{4}\sum_{\ell_1,\ell_2,\ell_3,\ell_4\in\mathbb{Z}} w_{\ell_1}(p_1)w^*_{\ell_2}(p_2)w_{\ell_3}(p_3)w^*_{\ell_4}(p_4)\,\langle a_{\ell_1}a_{\ell_2}a_{\ell_3}a_{\ell_4}\rangle_c\,,$$

$$\langle b(p_1)b(p_2)b(p_3)b(p_4)\rangle_c = \frac{1}{4}\sum_{\ell_1,\ell_2,\ell_3,\ell_4\in\mathbb{Z}} w^*_{\ell_1}(p_1)w^*_{\ell_2}(p_2)w^*_{\ell_3}(p_3)w^*_{\ell_4}(p_4)\,\langle a_{\ell_1}a_{\ell_2}a_{\ell_3}a_{\ell_4}\rangle_c\,, \quad (127)$$

$$\langle b^\dagger(p_1)b(p_2)b(p_3)b(p_4)\rangle_c = \frac{1}{4}\sum_{\ell_1,\ell_2,\ell_3,\ell_4\in\mathbb{Z}} w_{\ell_1}(p_1)w^*_{\ell_2}(p_2)w^*_{\ell_3}(p_3)w^*_{\ell_4}(p_4)\,\langle a_{\ell_1}a_{\ell_2}a_{\ell_3}a_{\ell_4}\rangle_c\,.$$

Now the idea is to plug in the expression of $C_{\ell_1,\ell_2,\ell_3,\ell_4} = \langle a_{\ell_1}a_{\ell_2}a_{\ell_3}a_{\ell_4}\rangle_c$ that we obtained above. Before doing so, it is convenient to split the sum as

$$\sum_{\ell_1,\ell_2,\ell_3,\ell_4\in\mathbb{Z}} f_{\ell_1,\ell_2,\ell_3,\ell_4}C_{\ell_1,\ell_2,\ell_3,\ell_4} =$$

$$= \sum_{\ell_1,\ell_2,\ell_3,\ell_4\in\mathbb{Z}}\Bigg(f_{2\ell_1+1,2\ell_2+1,2\ell_3+2,2\ell_4+2}C_{2\ell_1+1,2\ell_2+1,2\ell_3+2,2\ell_4+2}+$$

$$+ f_{2\ell_1+1,2\ell_2+2,2\ell_3+1,2\ell_4+2}C_{2\ell_1+1,2\ell_2+2,2\ell_3+1,2\ell_4+2}$$

$$+ f_{2\ell_1+1,2\ell_2+2,2\ell_3+2,2\ell_4+1}C_{2\ell_1+1,2\ell_2+2,2\ell_3+2,2\ell_4+1}+$$

$$+ f_{2\ell_1+2,2\ell_2+1,2\ell_3+1,2\ell_4+2}C_{2\ell_1+2,2\ell_2+1,2\ell_3+1,2\ell_4+2}+$$

$$+ f_{2\ell_1+2,2\ell_2+1,2\ell_3+2,2\ell_4+1}C_{2\ell_1+2,2\ell_2+1,2\ell_3+2,2\ell_4+1}+$$

$$+ f_{2\ell_1+2,2\ell_2+2,2\ell_3+1,2\ell_4+1}C_{2\ell_1+2,2\ell_2+2,2\ell_3+1,2\ell_4+1}\Bigg) =$$

$$= \sum_{\ell_1,\ell_2\in\mathbb{Z}}\Bigg(f_{2\ell_1+1,2\ell_2+1,2\ell_1+2,2\ell_2+2} - f_{2\ell_1+1,2\ell_1+2,2\ell_2+1,2\ell_2+2} + f_{2\ell_1+1,2\ell_1+2,2\ell_2+2,2\ell_2+1}+$$

$$+ f_{2\ell_1+2,2\ell_1+1,2\ell_2+1,2\ell_2+2} - f_{2\ell_1+2,2\ell_1+1,2\ell_2+2,2\ell_2+1} + f_{2\ell_1+2,2\ell_2+2,2\ell_1+1,2\ell_2+1}+$$

$$- f_{2\ell_1+1,2\ell_2+1,2\ell_2+2,2\ell_1+2} + f_{2\ell_1+1,2\ell_2+2,2\ell_2+1,2\ell_1+2} - f_{2\ell_1+1,2\ell_2+2,2\ell_1+2,2\ell_2+1}+$$

$$- f_{2\ell_2+2,2\ell_1+1,2\ell_2+1,2\ell_1+2} + f_{2\ell_2+2,2\ell_1+1,2\ell_1+2,2\ell_2+1} - f_{2\ell_2+2,2\ell_1+2,2\ell_1+1,2\ell_2+1}\Bigg)\langle\sigma^z_{\ell_1}\sigma^z_{\ell_2}\rangle_0\,,$$

$$(128)$$

where we have used that, in order to have a nonzero result, $C$ needs to have two even and two odd indices, then we have used its explicit expression. Using the explicit expression of the eigenvectors $w(p)$ – see Eq. (11) – and introducing the (discrete) Fourier transform of the spin-spin correlation function

$$T(p,q) := \sum_{m,n\in\mathbb{Z}}\langle\sigma^z_m\sigma^z_n\rangle_0\,\mathrm{e}^{-\mathrm{i}pm}\mathrm{e}^{-\mathrm{i}qn}, \quad (129)$$

we finally have

$$\langle b^\dagger(p_1)b(p_2)b^\dagger(p_3)b(p_4)\rangle_c = \frac{1}{4}T(-p_1-p_3,p_2+p_4)\sin\left(\tfrac{\theta(p_1)-\theta(p_3)}{2}\right)\sin\left(\tfrac{\theta(p_2)-\theta(p_4)}{2}\right)$$

$$+ \frac{1}{4}T(-p_1+p_2,-p_3+p_4)\cos\left(\tfrac{\theta(p_1)+\theta(p_2)}{2}\right)\cos\left(\tfrac{\theta(p_3)+\theta(p_4)}{2}\right)$$

$$- \frac{1}{4}T(-p_1+p_4,p_2-p_3)\cos\left(\tfrac{\theta(p_1)+\theta(p_4)}{2}\right)\cos\left(\tfrac{\theta(p_2)+\theta(p_3)}{2}\right), \quad (130)$$

$$\langle b(p_1)b(p_2)b(p_3)b(p_4)\rangle_c = \frac{1}{4}T(p_1+p_3,p_2+p_4)\sin\left(\frac{\theta(p_1)-\theta(p_3)}{2}\right)\sin\left(\frac{\theta(p_2)-\theta(p_4)}{2}\right)$$
$$-\frac{1}{4}T(p_1+p_2,p_3+p_4)\sin\left(\frac{\theta(p_1)-\theta(p_2)}{2}\right)\sin\left(\frac{\theta(p_3)-\theta(p_4)}{2}\right)$$
$$-\frac{1}{4}T(p_1+p_4,p_2+p_3)\sin\left(\frac{\theta(p_1)-\theta(p_4)}{2}\right)\sin\left(\frac{\theta(p_2)-\theta(p_3)}{2}\right),\quad(131)$$

$$\langle b^\dagger(p_1)b(p_2)b(p_3)b(p_4)\rangle_c = \frac{i}{4}T(-p_1+p_2,p_3+p_4)\cos\left(\frac{\theta(p_1)+\theta(p_2)}{2}\right)\sin\left(\frac{\theta(p_3)-\theta(p_4)}{2}\right)$$
$$-\frac{i}{4}T(-p_1+p_3,p_2+p_4)\cos\left(\frac{\theta(p_1)+\theta(p_3)}{2}\right)\sin\left(\frac{\theta(p_2)-\theta(p_4)}{2}\right)$$
$$+\frac{i}{4}T(-p_1+p_4,p_2+p_3)\cos\left(\frac{\theta(p_1)+\theta(p_4)}{2}\right)\sin\left(\frac{\theta(p_2)-\theta(p_3)}{2}\right).\quad(132)$$

### C.2  Integral representation of the spin-spin correlation function

Let us now specialize to our case of interest. Using

$$\langle\sigma_m^z\sigma_n^z\rangle_0 = (\tanh\beta)^{|m-n|}\Theta_m\Theta_n\,,\qquad(133)$$

we can compute explicitly the function $T(p,q)$ of the previous section:

$$T(p,q) = \sum_{m,n\in\mathbb{Z}} e^{-ipm}e^{-iqn}(\tanh\beta)^{|m-n|}\Theta_m\Theta_n = \int_{-\pi}^{\pi}\frac{dk}{2\pi}\frac{T_\beta(k)}{(1-e^{-i(k+p-i0)})(1-e^{-i(q-k-i0)})},$$
$$(134)$$

where

$$T_\beta(k) := \sum_{j\in\mathbb{Z}}(\tanh\beta)^{|j|}e^{-ikj} = \frac{1}{\cosh^2(\beta)}\frac{1}{\tanh^2(\beta)-2\tanh(\beta)\cos(k)+1}\qquad(135)$$

is the (discrete) Fourier transform of the homogeneous spin-spin correlation function; note that a straightforward generalization of our problem to a different initial state is obtained simply by changing $T_\beta(k)$. Now we can compute the initial value of all the dynamical fields; for example:

$$\xi_{x_1,x_2,0}^{phys}(p_1,p_2) = \int_{-\pi}^{+\pi}\frac{d^2qdk}{(2\pi)^3}e^{2ix_1q_1}e^{2ix_2q_2}\frac{T_\beta(k)}{4}\times$$
$$\times\left(\frac{\sin\left(\frac{\theta(p_1-q_1)-\theta(p_2-q_2)}{2}\right)\sin\left(\frac{\theta(p_1+q_1)-\theta(p_2+q_2)}{2}\right)}{(1-e^{-i(k-p_1+q_1-p_2+q_2-i0)})(1-e^{-i(p_1+q_1+p_2+q_2-k-i0)})}\right.$$
$$+\frac{\cos\left(\frac{\theta(p_1-q_1)+\theta(p_1+q_1)}{2}\right)\cos\left(\frac{\theta(p_2-q_2)+\theta(p_2+q_2)}{2}\right)}{(1-e^{-i(k+2q_1-i0)})(1-e^{-i(2q_2-k-i0)})}+$$
$$\left.-\frac{\cos\left(\frac{\theta(p_1-q_1)+\theta(p_2+q_2)}{2}\right)\cos\left(\frac{\theta(p_1+q_1)+\theta(p_2-q_2)}{2}\right)}{(1-e^{-i(k-p_1+q_1+p_2+q_2-i0)})(1-e^{-i(p_1+q_1-p_2+q_2-k-i0)})}\right)$$
$$= \int_{-\pi}^{+\pi}\frac{d^2qdk}{(2\pi)^3}\frac{e^{2ix_1q_1}e^{2ix_2q_2}}{4(1-e^{-i(2q_1+k-i0)})(1-e^{-i(2q_2-k-i0)})}F_\beta(p_1,p_2,q_1,q_2,k),\quad(136)$$

with

$$F_\beta(p_1, p_2, q_1, q_2, k) :=$$

$$T_\beta(k + q_1 + p_1 + p_2 - q_2)\sin\left(\tfrac{\theta(p_1 - q_1) - \theta(p_2 - q_2)}{2}\right)\sin\left(\tfrac{\theta(p_1 + q_1) - \theta(p_2 + q_2)}{2}\right) +$$

$$+ T_\beta(k)\cos\left(\tfrac{\theta(p_1 - q_1) + \theta(p_1 + q_1)}{2}\right)\cos\left(\tfrac{\theta(p_2 - q_2) + \theta(p_2 + q_2)}{2}\right) +$$

$$- T_\beta(k + p_1 + q_1 - p_2 - q_2)\cos\left(\tfrac{\theta(p_1 - q_1) + \theta(p_2 + q_2)}{2}\right)\cos\left(\tfrac{\theta(p_1 + q_1) + \theta(p_2 - q_2)}{2}\right). \quad (137)$$

We can now use Eq. (46) to get the time-dependent expression of the field

$$\xi_{x_1, x_2, t}(p_1, p_2) =$$

$$\int_{-\pi}^{+\pi} \frac{\mathrm{d}^2 q \mathrm{d}k}{(2\pi)^3} \frac{\mathrm{e}^{2\mathrm{i}x_1 q_1}\mathrm{e}^{2\mathrm{i}x_2 q_2}\mathrm{e}^{-\mathrm{i}t(\epsilon(p_1 + q_1) - \epsilon(p_1 - q_1) + \epsilon(p_2 + q_2) - \epsilon(p_2 - q_2))}}{4(1 - \mathrm{e}^{-\mathrm{i}(2q_1 + k - \mathrm{i}0)})(1 - \mathrm{e}^{-\mathrm{i}(2q_2 - k - \mathrm{i}0)})} F_\beta(p_1, p_2, q_1, q_2, k), \quad (138)$$

and finally compute the contribution of the field $\xi$ to the spin-spin correlation using its expression (48), the parametrization of the symbol (39) and the inversion relation (38):

$$S_{mn}^{z,\xi}(t) = -4\int_{-\pi}^{\pi} \frac{\mathrm{d}^2 p \mathrm{d}^2 k}{(2\pi)^4} \sum_{y_1, y_2 \in \mathbb{Z}/2} \mathrm{e}^{2\mathrm{i}k_1(m-1-y_1)}\mathrm{e}^{2\mathrm{i}k_2(n-1-y_2)}\xi_{y_1, y_2, t; ee}(p_1, p_2) \times$$

$$\times \left(\mathrm{e}^{-\mathrm{i}\tfrac{\theta(p_1 + k_1)}{2}\sigma^z} \otimes \mathrm{e}^{-\mathrm{i}\tfrac{\theta(p_2 + k_2)}{2}\sigma^z}\right)(\sigma^y \otimes \sigma^y)\left(\mathrm{e}^{\mathrm{i}\tfrac{\theta(p_1 - k_1)}{2}\sigma^z} \otimes \mathrm{e}^{\mathrm{i}\tfrac{\theta(p_2 - k_2)}{2}\sigma^z}\right)|_{1,4}$$

$$= 4\int_{-\pi}^{\pi} \frac{\mathrm{d}^2 p \mathrm{d}^2 k}{(2\pi)^2 4} \sum_{y_1, y_2 \in \mathbb{Z}/2} \mathrm{e}^{2\mathrm{i}k_1(m-1-y_1)}\mathrm{e}^{2\mathrm{i}k_2(n-1-y_2)}\xi_{y_1, y_2, t; ee}(p_1, p_2) \times$$

$$\times \mathrm{e}^{-\mathrm{i}\tfrac{\theta(p_1 - k_1) + \theta(p_1 + k_1) + \theta(p_2 - k_2) + \theta(p_2 + k_2)}{2}}$$

$$= 4\int_{-\pi}^{\pi} \frac{\mathrm{d}^2 p \mathrm{d}^2 k}{(2\pi)^4} \sum_{y_1, y_2 \in \mathbb{Z}/2} \mathrm{e}^{2\mathrm{i}k_1(m-1-y_1)}\mathrm{e}^{2\mathrm{i}k_2(n-1-y_2)}\xi_{y_1, y_2, t}(p_1, p_2) \quad (139)$$

$$\cos\left(\frac{\theta(p_1 - k_1) + \theta(p_1 + k_1)}{2}\right)\cos\left(\frac{\theta(p_2 - k_2) + \theta(p_2 + k_2)}{2}\right)$$

$$= \int_{-\pi}^{\pi} \frac{\mathrm{d}^2 p \mathrm{d}^2 q \mathrm{d}k}{(2\pi)^5} \frac{\mathrm{e}^{2\mathrm{i}mq_1}\mathrm{e}^{2\mathrm{i}nq_2}\mathrm{e}^{-\mathrm{i}t(\epsilon(p_1 + q_1) - \epsilon(p_1 - q_1) + \epsilon(p_2 + q_2) - \epsilon(p_2 - q_2))}}{(1 - \mathrm{e}^{-\mathrm{i}(2q_1 + k - \mathrm{i}0)})(1 - \mathrm{e}^{-\mathrm{i}(2q_2 - k - \mathrm{i}0)})}\mathrm{e}^{-2\mathrm{i}q_1}\mathrm{e}^{-2\mathrm{i}q_2}$$

$$\cos\left(\tfrac{\theta(p_1 - q_1) + \theta(p_1 + q_1)}{2}\right)\cos\left(\tfrac{\theta(p_2 - q_2) + \theta(p_2 + q_2)}{2}\right) F_\beta(p_1, p_2, q_1, q_2, k),$$

where we used the Moyal product's integral representation (31) adapted to the symbol $\tilde{C}$. We then change variables as

$$p_1 \to p_1 + k/2, \qquad p_2 \to p_2 + q_2, \qquad q_1 \to q_1 + k/2, \qquad q_2 \to q_2 - q_1, \qquad k \to -2q_1, \quad (140)$$

obtaining

$$S_{mn}^{z,\xi}(t) =$$

$$\int_{-\pi}^{\pi} \frac{\mathrm{d}^2 p \mathrm{d}^2 q \mathrm{d}k}{(2\pi)^5} \frac{\mathrm{e}^{2\mathrm{i}q_1(m-n)}\mathrm{e}^{\mathrm{i}mk}\mathrm{e}^{2\mathrm{i}nq_2}\mathrm{e}^{-\mathrm{i}t(\epsilon(p_1 + q_1 + k) - \epsilon(p_1 - q_1) + \epsilon(p_2 - q_1 + 2q_2) - \epsilon(p_2 + q_1))}}{(1 - \mathrm{e}^{-\mathrm{i}(k - \mathrm{i}0)})(1 - \mathrm{e}^{-\mathrm{i}(2q_2 - \mathrm{i}0)})}\mathrm{e}^{-\mathrm{i}k} \quad (141)$$

$$\mathrm{e}^{-2\mathrm{i}q_2}\cos\left(\tfrac{\theta(p_1 - q_1) + \theta(p_1 + q_1 + k)}{2}\right)\cos\left(\tfrac{\theta(p_2 + q_1) + \theta(p_2 - q_1 + 2q_2)}{2}\right) G_\beta(p_1, p_2, q_1, 2q_2, k),$$

where

$$G_\beta(p_1, p_2, q_1, q_2, k) :=$$
$$T_\beta(k + p_1 + p_2) \sin\left(\tfrac{\theta(p_1-q_1)-\theta(p_2+q_1)}{2}\right) \sin\left(\tfrac{\theta(p_1+q_1+k)-\theta(p_2-q_1+q_2)}{2}\right) +$$
$$+ T_\beta(2q_1) \cos\left(\tfrac{\theta(p_1-q_1)+\theta(p_1+q_1+k)}{2}\right) \cos\left(\tfrac{\theta(p_2+q_1)+\theta(p_2-q_1+q_2)}{2}\right) +$$
$$- T_\beta(k + p_1 - p_2 - q_2) \cos\left(\tfrac{\theta(p_1-q_1)+\theta(p_2-q_1+q_2)}{2}\right) \cos\left(\tfrac{\theta(p_1+q_1+k)+\theta(p_2+q_1)}{2}\right). \quad (142)$$

Note that $G_\beta$ is periodic in all its entries with period $2\pi$. In the same way, we could derive an integral representation of the contribution of the other non-Gaussian fields. However, as discussed in the main text, those fields give a sub-leading contribution.

# D    Numerical simulations

Numerical simulations for Figures (3), (4) and (6) have been performed considering finite-size chains of length $2L$, in which $\mathcal{H}$ is a $2L \times 2L$ matrix. The part of state entering the expression of the spin-spin connected correlation (48) is described by the objects $\Gamma$ and $C$ defined in the main text, which have dimensions $2L \times 2L$ and $2L \times 2L \times 2L \times 2L$ respectively. Time-evolution has been done exactly using

$$a_i(t) = \sum_{j=1}^{2L} (\mathrm{e}^{-\mathrm{i}t\mathcal{H}})_{ij} a_j(0), \tag{143}$$

which implies

$$\Gamma_{m,n}(t) = \sum_{m',n'=1}^{2L} (\mathrm{e}^{-\mathrm{i}t\mathcal{H}})_{m,m'} (\mathrm{e}^{-\mathrm{i}t\mathcal{H}})_{n,n'} \Gamma_{m',n'}(0) \tag{144}$$

and

$$C_{m,n,r,s}(t) = \sum_{m',n',r',s'=1}^{2L} (\mathrm{e}^{-\mathrm{i}t\mathcal{H}})_{m,m'} (\mathrm{e}^{-\mathrm{i}t\mathcal{H}})_{n,n'} (\mathrm{e}^{-\mathrm{i}t\mathcal{H}})_{r,r'} (\mathrm{e}^{-\mathrm{i}t\mathcal{H}})_{s,s'} C_{m',n',r',s'}(0). \tag{145}$$

We always stop the simulation at a time such that the finite size effects have only exponentially-small consequences on the expectation values that we are looking at, which is always possible thanks to the Lieb-Robinson bounds. In the case of Gaussian protocols, the initial state has been actually considered sharp, and not smooth as described in Sections 5.2.1 and 5.2.2, since it is easier to construct. Therefore the state that we are simulating differs from the one described in the main text around the junction. However, the homogeneous states associated with the two partitioning protocols are the same and the difference arguably induces only sub-leading effects. That is why such a simulation can still be used to check the leading-order predictions.

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
