# Peer review of "Connected correlations in partitioning protocols: a case study and beyond"

_SciPost Physics_

## Round 1 · Referee Report · Anonymous · 2023-1-18

Strengths
1-An exact treatment of a many-body system to study local relaxation and "Gaussification".
2-Analytical predictions verified by numerical calculations.
3-A simple scaling result for what constitutes a "local" observable described by a (local) equilibrium state.
Weaknesses
1-Section 4 gives a brief review of Ref. 41, so the Equations 26,27 are introduced as a "convenient decomposition", even though they are crucial for what follows. Section 4.2 then considers an analogous decomposition of the four-point function, but it remains unclear how this is accomplished systematically or what motivates the definitions of the fields in Equation 41.
2 - Sections 6-8 on asymptotic analysis are hard to follow even after repeated attempts. The derivations might be much easier to visualize if the respective figures were placed in the text when they are discussed and the described contour deformations explicitly referred to. As is, the captions of these figures are insufficient to reconstruct the process.
3-Section 9 is somewhat vague and again not the easiest to follow. It appears that the predicted behavior is based on a generalization of the analysis of the two-point function, but the relevant part of Section 8 (page 36) appears to reference calculations that can be performed but are not shown? Is it possible to specify which predictions are conjectured and which follow directly from the results in the paper?
Report
The paper considers the question of how "local" must an observable be if its dynamics is to be described by a locally homogeneous ensemble, an assumption which underlies generalized hydrodynamics.
The author studies the equal-time spin two-point function in the transverse field Ising spin chain. The model is diagonalized by successive applications of the Jordan-Wigner and Bogoliubov transformations. The dynamics of the correlation matrix in a Gaussian state are reformulated in terms of two fields, one of which is demonstrated to reduce to the "root-density" in the homogeneous limit.
The dynamical contributions of the root-density, the auxiliary field and a particular non-Gaussian initial state are each studied in turn by considering different bipartitioning protocols. The rest of the paper is a detailed analysis of the asymptotics of the integral representations of the time-evolved two-point function.
The main outcome of the analysis is that the length of an equal-time observable can scale at most as the square root of time if it is to be descried solely by a local equilibrium state. Decay rates of sub-leading corrections are also extracted.
Questions:
1. Can a similar analysis be performed for an unequal-time observable? What would be the expected "locality scaling" there?
2. Is the $x \sim t^{1/2}$ threshold locality scaling expected generally in interacting integrable models or is it specific to free systems?
Requested changes
Address the above-mentioned weaknesses of the presentation.
Minor misprint:
-In Sections 6.1/7.1/8.1 the scaling should be $x/t = \zeta$ not $\zeta t$.
Author: Saverio Bocini on 2023-04-05 [id 3551]
(in reply to Report 1 on 2023-01-18)
1) A similar analysis can definitely be performed for unequal-time observables. In that case, the question of locality is shifted to the question “do the two observables lie along the same ray?”. If they do, GHD is expected to hold, while if they do not, one expects corrections similar to those addressed in the paper. However, I have not performed a detailed analysis of this case, and the study of the precise interplay between this particular scaling limit and locality is left as an interesting question for a future work 2) The exponent 1/2 is not expected to be the “locality threshold” for generic integrable systems, since, due to interactions, it is more likely to vary from one system to the other. But its existence is still an important result about generic integrable models: any theory that wants to go beyond GHD relying only on the root density is bound to fail at some point if we increase the scaling of the support of the observable.
Author: Saverio Bocini on 2023-04-05 [id 3552]
(in reply to Report 2 on 2023-03-18)1) The choice $\alpha=1$ corresponds indeed to the Euler-scale connected correlation function between two $\sigma^z$ operators. The correlation is sensitive to whether the initial condition is made by two GGE baths or not because the auxiliary field vanishes for a GGE. Therefore, in the GGE case the auxiliary field does not give any contribution outside the light cone. If instead the state is not a GGE, the contribution of the auxiliary field to the correlation decays as $t^{-1}$, regardless of smoothness of the state. In particular, it gives $t^{-1}$ even if the state is smooth. 2) The “straightforward” way to define $\psi$ in integrable models would be to link it to the expectation value of the product of two operators that correspond to the annihilation of quasi-particles in integrable models. However, I have not really worked out the details of this generalisation, nor have I studied to which extent the final quantity is manageable. As for a classical interpretation of $\psi$, it may help noting that for a translationally-invariant quench from a Gaussian state, the field represents completely all the component of the state that is evolving. In this sense, $\psi$ is the non-stationary part of the state. However, the dynamical equations satisfied by $\psi$ in the limit of low homogeneity maintain a dependence on $\hbar$ (unlike what happens for the field $\rho$), so the field can be regarded as purely quantum.

---

## Round 1 · Referee Report · Anonymous · 2023-3-18

Strengths
1- The paper evaluates dynamical spin-spin correlation functions for free fermionic chains at large space-time scale by exact computations.
2- The paper provides an extensive analysis on what fields are relevant dependent on how the distance between the two operators in the correlators scale.
3- The analytical results are numerically checked.
Weaknesses
1- Sometimes hard to read due to many definitions.
2- Results are limited to non-interacting systems.
Report
The article is concerned with the computations of dynamical correlation functions for the transverse field Ising chain at the Euler scale (i.e. spatial coordinates scale with time). Partitioning protocols are used as initial conditions, but unlike the usual case, correlations between spin operators across the origin are allowed to exist. The author then study the large scale behaviour of the correlators $S^z_{m,n}(t)=\langle\sigma^z_m(t)\sigma^z_n(t)\rangle^\mathrm{c}$ by varying the exponent $\alpha\in[0,1]$ that controls how fast the distance between two operators grows, i.e. $m-n\sim t^\alpha$. Three different types of contributions are identified, and interestingly, when $0\leq\alpha<1/2$, the root density is the only needed field to describe the leading behaviour of the correlators, which decay . This also implies that Gaussianization takes place for $\alpha$ in this parameter regime. All the results are obtained by tour de force computations, which are shown to agree with numerics pretty well.
As far as I can see, the calculations are carefully carried out and the results obtained in this paper provide an almost complete asymptotic analysis on $S^z_{m,n}(t)$. This is an important achievement. But I have one criticism about presentation and a few comments , which I would like the author to address before recommending for publication.
The first one is about presentation. I do feel that the pursuit of completeness with many definitions is reducing readability. I understand that, because of the nature of the work, it is to some extent inevitable to introduce different notions. Even so, there are simply too many functions that are related to the correlation functions $S^z_{m,n}(t)$ and it is hard to keep track of how they are related to each other. For instance, if it doesn't complicate the exposition, I think it is a good idea to write the results in Sect 6.2 in terms of the original correlators. Providing a table that displays the asymptotics of the correlation function with different regimes and the main contributing fields could be useful.
Concerning the results, I am curious about the nature of the auxiliary field. In general, the choice $\alpha=1$ corresponds to the Euler-scale correlation functions (see e.g. arXiv:2206.14167), which can be described by hydrodynamics provided that the initial condition is in a local equilibrium. If this condition is not satisfied, for example in a (GGE) partitioning protocol, the steep structure influences the initial microscopic physics, which amounts to long-range correlations that decay as $t^{-1}$. This general expectation is in agreement with the results presented in the paper, and the author attributes it to the auxiliary field. This raises a question: as we smoothen the initial condition, do we see that the contribution coming from the auxiliary field vanishes? If this can be somehow argued, it would make it clearer that the auxiliary field is indeed responsible for capturing the sharp structure at the Euler scale. It could be however sensitive to whether the initial condition is made of two GGE baths or not.
Another related question is whether we can generalize the notion of the auxiliary field to interacting integrable systems. I also wonder if there could be a classical analogue of it.
Requested changes
1- Change the format so that main results are easier to appreciate.

---

## Editorial Decision

unknown